# THE DIFFUSION DUALITY, CHAPTER II: Ψ-SAMPLERS

**Justin Deschenaux**[1*]  **Caglar Gulcehre**[1,2]  **Subham Sekhar Sahoo**[3*]

[1]EPFL, Lausanne, Switzerland  [2]Microsoft AI  [3]Cornell Tech, NY

## ABSTRACT

Uniform-state discrete diffusion models excel at few-step generation and guidance due to their ability to self-correct, making them preferred over autoregressive or Masked diffusion models in these settings. However, their sampling quality plateaus with ancestral samplers as the number of steps increases. We introduce a family of Predictor-Corrector (PC) samplers for discrete diffusion that generalize prior methods and apply to arbitrary noise processes. When paired with uniform-state diffusion, our samplers outperform ancestral sampling on both language and image modeling, achieving lower generative perplexity at matched unigram entropy on OpenWebText and better FID/IS scores on CIFAR10. Crucially, unlike conventional samplers, our PC methods continue to improve with more sampling steps. **Taken together, these findings call into question the assumption that Masked diffusion is the inevitable future of diffusion-based language modeling.** Beyond sampling, we develop a memory-efficient curriculum for the Gaussian relaxation training phase, reducing training time by 25% and memory by 33% compared to Duo while maintaining comparable perplexity on OpenWebText and LM1B and strong downstream performance. We release code, checkpoints, and a video-tutorial on:
https://s-sahoo.com/duo-ch2

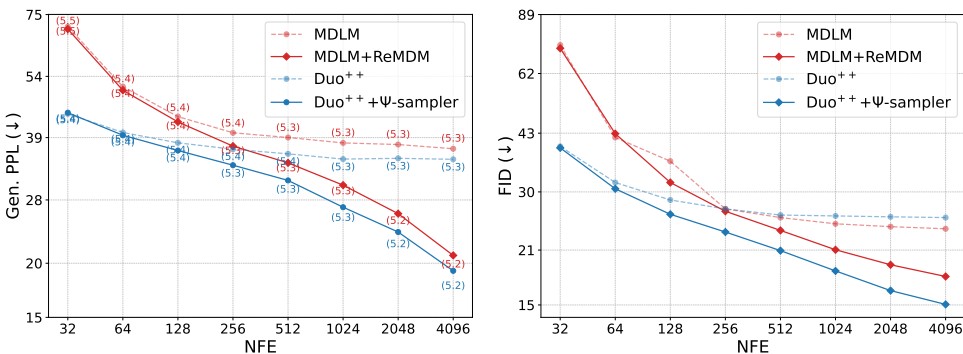

Figure 1: **Performance on Language Modeling and Image Modeling.** Ψ-samplers generalize ReMDM (Wang et al., 2025) to arbitrary noise distributions. **(Left):** Generative perplexity (Gen. PPL; ↓) as a function of NFEs, with nucleus sampling $p = 0.9$. Ψ-samplers consistently improve with more steps, unlike ancestral sampling which plateaus. Curves are annotated with the average unigram entropy per sequence as a proxy for diversity. **(Right):** On CIFAR-10, Ψ-samplers achieve better FID (↓) than MDLM (with ReMDM).

## 1 INTRODUCTION

Diffusion models are powerful generative algorithms that have achieved remarkable success in modeling continuous data domains, including images (Ho et al., 2020a; Rombach et al., 2022), audio (Kong et al., 2021; Liu et al., 2023b; Huang et al., 2023), and videos (Ho et al., 2022; Esser et al., 2023; Blattmann et al., 2023; Polyak et al., 2025). Recent advances have extended diffusion models to categorical data, demonstrating their potential for language modeling (Austin et al., 2023;

---

*Correspondence to justin.deschenaux@epfl.ch and ssahoo@cs.cornell.edu

Lou et al., 2024; Sahoo et al., 2024; Shi et al., 2025; Ou et al., 2025; Sahoo et al., 2025a;b), graphs (Liu et al., 2023a), molecules (Lee et al., 2025), and audio (Ku et al., 2025). Unlike autoregressive models that generate tokens sequentially from left to right, diffusion language models can decode tokens in parallel and in any order while leveraging bidirectional contextual information. This capability enables the design of language models that can be significantly faster than their autoregressive counterparts while maintaining strong downstream performance (Song et al., 2025; Labs et al., 2025).

Discrete diffusion models primarily employ one of two noise distributions: a uniform prior or a masked prior that concentrates all probability mass on a special [MASK] token. Unlike *Masked Diffusion Models* (MDMs), which update each token exactly once, *Uniform-State Diffusion Models* (USDMs) allow tokens to be revised multiple times during generation, enabling self-correction. This makes USDMs particularly effective for few-step (Sahoo et al., 2025a) and guided generation (Schiff et al., 2025). However, the generation quality of USDMs has not yet matched that of MDMs in high-sampling-step regimes, and USDMs' modeling capacity, as measured by likelihood, remains inferior to MDMs'. Although Sahoo et al. (2025a) proposed a curriculum learning strategy (Bengio et al., 2009) that narrows the likelihood gap, this curriculum is computationally expensive.

To address MDMs' inability to remask tokens, ReMDM (Wang et al., 2025) introduced "Predictor-Corrector" (PC) samplers that generalize and outperform earlier PC methods (Campbell et al., 2022; Gat et al., 2024). These samplers substantially improve the inference time scaling behavior of MDMs. However, PC methods for uniform-state diffusion remain underexplored. Campbell et al. (2022) proposed PC methods for samplers that take advantage of the rate change matrices of the continuous-time Markov chain (CTMC) formulation of discrete diffusion processes, but such samplers are known to perform worse than ancestral samplers (Lou et al., 2024; Schiff et al., 2025).

We propose Duo$^{++}$ to address these challenges, which expands the design space of USDMs using non-Markovian *superposition posteriors* (or as we refer to them in this paper, $\Psi$-posteriors). These posteriors align with the intermediate marginals of discrete diffusion processes and give rise to $\Psi$-samplers with predictor-corrector capabilities that are crucial for improving sample quality. In addition, Duo$^{++}$ introduces an efficient curriculum learning strategy that advances the approach of Sahoo et al. (2025a) by accelerating training and reducing memory usage.

In summary, our contributions are threefold: (1) we propose a family of non-Markovian posteriors ($\Psi$-posteriors) for discrete diffusion with arbitrary noise priors that share the same marginals as the Markovian discrete diffusion process (Sec. 3). (2) We demonstrate that the induced $\Psi$-samplers improve text and image generation quality and scale better than standard ancestral samplers in high NFE regimes, closing the performance gap with respect to MDMs coupled with remasking samplers in high NFE regimes for text generation (Sec. 5.1) and surpassing them on image generation tasks (Sec. 5.1.2). (3) We reformulate the curriculum learning strategy proposed in Sahoo et al. (2025a), achieving a $2\times$ speedup while reducing peak memory usage by 33% and end-to-end training time by 25%, while maintaining similar perplexity (Fig. 1, right, Table 5) and downstream task accuracy (Table 1).

## 2 BACKGROUND

**Notation** Let $\mathcal{V} := \{\mathbf{v} \in \{0,1\}^K : \sum_{i=1}^{K} \mathbf{v}_i = 1\}$ denote the set of one-hot encodings of discrete random variables over $K$ categories. Let $\mathbf{x} \in \mathcal{V}^L$ denote a sequence of $L$ discrete variables in $\mathcal{V}$ and $\mathbf{x}^\ell$ denote the entry $\ell^{\text{th}}$ in $\mathbf{x}$. We use boldface to denote both individual vectors and sequences; the context will make clear whether a symbol refers to a vector or a sequence. Let $\Delta$ denote the $K$ simplex. For $\mathbf{v} \in \Delta$, let $\text{Cat}(\cdot; \mathbf{v})$ denote a categorical distribution such that $\mathbb{P}(\mathbf{u}_i = 1) = \mathbf{v}_i$, for $\mathbf{u} \sim \text{Cat}(\cdot; \mathbf{v}), \mathbf{u} \in \mathcal{V}$. Let $\langle \mathbf{a}, \mathbf{b} \rangle$ and $\mathbf{a} \odot \mathbf{b}$ denote the dot and Hadamard products between two vectors respectively. Let $\mathbf{1} = \{1\}^K$ denote the all-ones vector. Let $\boldsymbol{\pi} \in \Delta$ be a designated categorical distribution referred to as the prior.

### 2.1 DISCRETE DIFFUSION MODELS

Consider the clean data sequence $\mathbf{x}$ of length $L$ drawn from the data distribution $q_{\text{data}}$. Discrete diffusion models (Sohl-Dickstein et al., 2015; Austin et al., 2023) define a sequence of increasingly noisy distributions $(q_t)_{t \in [0,1]}$, interpolating from $q_{\text{data}}$ to a factorized prior distribution, which is a product of $L$ independent $\text{Cat}(.; \boldsymbol{\pi})$ distributions, using Markovian transitions defined independently across input dimensions (Campbell et al., 2022; Sahoo et al., 2024; Shi et al., 2025; Ou et al., 2025;

Schiff et al., 2025; Sahoo et al., 2025a). Let $\mathbf{z}_t \sim \prod_{\ell=1}^L q_t(.|\mathbf{x}^\ell)$ denote the intermediate latents (sequence) at time step $t$. This work focuses on factorized, interpolating noise processes (Sahoo et al., 2024), whose conditional marginal distribution takes the form:

$$\mathbf{z}_t^\ell \sim q_t(.|\mathbf{x}^\ell; \alpha_t) = \text{Cat}(.; \alpha_t \mathbf{x}^\ell + (1 - \alpha_t)\boldsymbol{\pi}), \tag{1}$$

where $\alpha_t \in [0, 1]$ is monotonically decreasing with $t$, and is known as the *noise schedule*. (1) defines the *forward process*, which progressively corrupts the data. The goal is to learn a *generative process* $p_\theta$, parameterized by a neural network with parameters $\theta$, that reverses this forward process to map from the noise prior back to $q_{\text{data}}$. The model is typically trained by minimizing the "Negative Evidence Lower Bound" (NELBO). The choice of token prior $\boldsymbol{\pi}$ gives rise to two popular variants: Masked Diffusion Models (MDMs) and Uniform-state Diffusion Models (USDMs), which we discuss in the following.

### 2.1.1 MASKED DIFFUSION PROCESSES

MDMs (Sahoo et al., 2024; Shi et al., 2025; Ou et al., 2025) use a masked prior, where $\boldsymbol{\pi} = \mathbf{m} \in \mathcal{V}$ is the one-hot representation of a special [MASK] token (Devlin et al., 2019). During the forward process (1), tokens either remain unchanged or transition to the masked state $\mathbf{m}$, after which they stay masked. This behavior carries over to the reverse process. The posterior of the reverse process $q_{s|t}^{\text{MDM}}$ for $0 \le s < t < 1$ can be derived using Bayes' Rule, and is given by:

$$q_{s|t}^{\text{MDM}}(.|\mathbf{z}_t^\ell, \mathbf{x}^\ell) = \begin{cases} \text{Cat}\left(.; \frac{\alpha_s - \alpha_t}{1 - \alpha_t}\mathbf{x}^\ell + \frac{1 - \alpha_s}{1 - \alpha_t}\mathbf{z}_t^\ell\right) & \text{if } \mathbf{z}_t^\ell = \mathbf{m}, \\ \text{Cat}(.; \mathbf{x}^\ell) & \text{otherwise.} \end{cases} \tag{2}$$

The approximate reverse posterior is $p_{s|t}^\theta = \prod_\ell q_{s|t}^{\text{MDM}}(.|\mathbf{z}_t^\ell, \mathbf{x}^\ell = \mathbf{x}_\theta^\ell(\mathbf{z}_t^{1:L}, t))$ where $\mathbf{x}_\theta : \mathcal{V}^L \times [0, 1] \to \Delta^L$ is the denoising model. A key limitation is that once unmasked, tokens cannot be remasked (2). This can create compounding errors during inference, as the denoising model $\mathbf{x}_\theta$ imperfectly models the clean data.

**Predictor-Corrector Methods** Wang et al. (2025) propose posteriors, and associated samplers (ReMDM) that maintain the same marginals as (2) during the generation process, while allowing remasking and generalizing previous training-free predictor-corrector methods such as Campbell et al. (2022); Gat et al. (2024).

### 2.1.2 UNIFORM-STATE DIFFUSION PROCESSES

Alternatively, discrete diffusion models can use a uniform prior $\boldsymbol{\pi} = \mathbf{1}/K$ (Schiff et al., 2025; Sahoo et al., 2025a). This choice allows tokens to change values multiple times throughout the generative process, in contrast to Masked diffusion. This property allows USDMs to excel in few-step generation (Sahoo et al., 2025a) and guidance applications (Schiff et al., 2025).

USDMs admit the following posterior distribution $q_{s|t}^{\text{USDM}}$ (for brevity, we simply write $q_{s|t}$ for $q_{s|t}^{\text{USDM}}$):

$$q_{s|t}(. \mid \mathbf{z}_t^\ell, \mathbf{x}^\ell) = \text{Cat}\left(.; \frac{K\alpha_t \mathbf{z}_t^\ell \odot \mathbf{x}^\ell + (\alpha_{t|s} - \alpha_t)\mathbf{z}_t^\ell + (\alpha_s - \alpha_t)\mathbf{x}^\ell + (1 - \alpha_{t|s})(1 - \alpha_s)\mathbf{1}/K}{K\alpha_t \langle \mathbf{z}_t^\ell, \mathbf{x}^\ell \rangle + 1 - \alpha_t}\right). \tag{3}$$

This posterior induces the following NELBO (Sahoo et al., 2025a):

$$\text{NELBO}(q, p_\theta; \mathbf{x}) = -\mathbb{E}_{t \sim \mathcal{U}[0,1], \, q_t(\mathbf{z}_t^\ell|\mathbf{x}^\ell; \alpha_t)} \sum_{\ell \in [L]} f(\mathbf{z}_t^\ell, \mathbf{x}_\theta^\ell(\mathbf{z}_t^\ell, t), \alpha_t; \mathbf{x}^\ell), \tag{4}$$

where

$$f(\mathbf{z}_t^\ell, \mathbf{x}_\theta^\ell(\mathbf{z}_t^\ell, t), \alpha_t; \mathbf{x}^\ell) = \frac{\alpha_t'}{K\alpha_t}\left[\frac{K}{\bar{\mathbf{x}}_r^\ell} - \frac{K}{(\bar{\mathbf{x}}_\theta^\ell)_r} - \left(\zeta_t \mathbb{1}_{\mathbf{z}_t^\ell = \mathbf{x}^\ell} + \mathbb{1}_{\mathbf{z}_t^\ell \neq \mathbf{x}^\ell}\right)\sum_j \log \frac{(\bar{\mathbf{x}}_\theta^\ell)_r}{(\bar{\mathbf{x}}_\theta^\ell)_j}\right.$$

$$\left. - K\frac{\alpha_t}{1 - \alpha_t}\log\frac{(\bar{\mathbf{x}}_\theta^\ell)_r}{(\bar{\mathbf{x}}_\theta^\ell)_i}\mathbb{1}_{\mathbf{z}_t^\ell \neq \mathbf{x}^\ell} - \left((K-1)\zeta_t\mathbb{1}_{\mathbf{z}_t^\ell = \mathbf{x}^\ell} - \frac{1}{\zeta_t}\mathbb{1}_{\mathbf{z}_t^\ell \neq \mathbf{x}^\ell}\right)\log\zeta_t\right]. \tag{5}$$

Here, $\bar{\mathbf{x}}^\ell = K\alpha_t\mathbf{x}^\ell + (1 - \alpha_t)\mathbf{1}$, $\bar{\mathbf{x}}_\theta^\ell = K\alpha_t\mathbf{x}_\theta^\ell(\mathbf{z}_t, t) + (1 - \alpha_t)\mathbf{1}$, $\alpha_t'$ denotes the time derivative of $\alpha_t$, $r = \arg\max_{j \in [K]}(\mathbf{z}_t^\ell)_j$ is the nonzero entry of $\mathbf{z}_t$, $\zeta_t = \frac{1 - \alpha_t}{K\alpha_t + 1 - \alpha_t}$, and $i$ denotes the index in $\mathbf{x}$ corresponding to 1, that is, $\mathbf{x}_i = 1$.

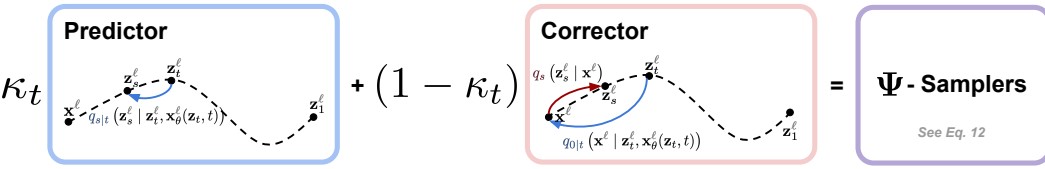

$\Psi$ - **Samplers simplify and generalize previous Predictor-Corrector methods to arbitrary prior** $\pi$

Figure 2: $\Psi$-samplers combine predictor and corrector steps. The predictor transitions from $\mathbf{z}_t$ to $\mathbf{z}_s$ via $q_{s|t}$, but fails to remask tokens in MDMs. The corrector steps inject noise via $q_s$, to revise earlier predictions. For $\kappa_t < 1$, noise injection enables error correction while preserving the forward process marginals. Our framework extends prior PC methods (Campbell et al., 2022; Gat et al., 2024; Wang et al., 2025) to arbitrary priors $\pi$.

**The Diffusion Duality**   Sahoo et al. (2025a) show that USDMs emerge from an underlying Gaussian diffusion process (Sohl-Dickstein et al., 2015; Ho et al., 2020b; Song et al., 2021; Kingma et al., 2023) on the one-hot representation $\mathbf{x}^\ell \in \mathcal{V}$. The Gaussian diffusion begins with $\mathbf{x}^\ell$ and progressively adds Gaussian noise leading to a sequence of noisy latents $\mathbf{w}_t^\ell \in \mathbb{R}^K \sim \tilde{q}_t(.|\mathbf{x}^\ell)$ for $t \in [0, 1]$ with the marginals:

$$\tilde{q}_t(.|\mathbf{x}^\ell; \tilde{\alpha}_t) = \mathcal{N}(.; \tilde{\alpha}_t \mathbf{x}^\ell, (1 - \tilde{\alpha}_t^2)\mathbf{I}_K),$$

where $(\tilde{\alpha}_t)_{t\in[0,1]}$ is a monotonically decreasing noise schedule. Let $\arg\max : \mathbb{R}^K \to \mathcal{V}$ map a continuous vector $\mathbf{v} \in \mathbb{R}^K$ to the one-hot vector corresponding to the index of its largest entry in $\mathbf{v}$, that is, $\arg\max(\mathbf{v}) = \arg\max_{\mathbf{z}\in\mathcal{V}} \mathbf{z}^\top \mathbf{v}$. When applied to a sequence of Gaussian latents $\mathbf{w}$, $\arg\max$ transforms them to the discrete latents $\mathbf{z}_t$ whose marginals take the form: $\mathbf{z}_t^\ell \sim q_t(.|\mathbf{x}^\ell; \alpha_t := \mathcal{T}(\tilde{\alpha}_t))$, where the function $\mathcal{T} : [0, 1] \to [0, 1]$ is the *Diffusion Transformation Operator*:

$$\mathcal{T}(\tilde{\alpha}_t) = \frac{K}{K-1}\left[\int_{-\infty}^{\infty} \phi\left(z - \frac{\tilde{\alpha}_t}{\sqrt{1-\tilde{\alpha}_t^2}}\right)\Phi^{K-1}(z)\mathrm{d}z - \frac{1}{K}\right], \quad (6)$$

where $\phi(z) = \exp(-z^2/2)/\sqrt{2\pi}$ and $\Phi(z) = \int_{-\infty}^{z} \phi(t)\mathrm{d}t$ are the standard Normal PDF and CDF, respectively. More formally, this relationship is expressed as:

$$q_t(\mathbf{z}_t^\ell|\mathbf{x}^\ell; \mathcal{T}(\tilde{\alpha}_t)) = [\arg\max]_\star \tilde{q}_t(\mathbf{w}_t^\ell|\mathbf{x}^\ell; \tilde{\alpha}_t) \quad (7)$$

where the $\star$ operator denotes the *pushforward* of the $K$-dimensional Gaussian density under the $\arg\max$ map, yielding a categorical distribution with $K$ classes. Note that while the marginal distribution $q_t(\mathbf{z}_t|\mathbf{x}; \mathcal{T}(\tilde{\alpha}_t))$ matches the discrete-space marginal in (1), **this does not imply that the full trajectory** $\{\mathbf{z}_t := \arg\max(\mathbf{w}_t)\}_{t\in[0,1]}$ **follows a (Markovian) discrete diffusion process** (Sahoo et al., 2025a). An interesting outcome of (7) is that the discrete NELBO (4) can be written in terms of Gaussian latents in the following manner, where the second $\arg\max$ is applied to each token independently:

$$\mathrm{NELBO}\,(q, p_\theta; \mathbf{x})$$
$$= \mathbb{E}_{\mathbf{x}, t\sim\mathcal{U}[0,1], \tilde{q}_t} \sum_{\ell\in[L]} f\Big(\mathbf{z}_t^\ell := \arg\max(\mathbf{w}_t^\ell), \mathbf{x}_\theta^\ell(\arg\max(\mathbf{w}_t), t), \alpha_t := \mathcal{T}(\tilde{\alpha}_t); \mathbf{x}^\ell\Big). \quad (8)$$

**Curriculum Learning**   Curriculum learning (Bengio et al., 2009) trains models by gradually increasing task difficulty. Building on this idea, Sahoo et al. (2025a) accelerate early training by using a biased but low-variance NELBO estimator for (8). Concretely, during the first 50% of training steps, the hard $\arg\max$ used to convert Gaussian latents into discrete tokens in the transformer's input is replaced by a low-temperature softmax relaxation. This relaxation interfaces naturally with the transformer input layer: if the latent at position $\ell$ is a probability vector $\mathbf{y}^\ell \in \Delta^K$, then the token representation is computed as a matrix product $\mathbf{V}^\top \mathbf{y}^\ell$, where $\mathbf{V} \in \mathbb{R}^{K\times m}$ is the vocabulary embedding matrix. For one-hot $\mathbf{y}^\ell$ (as produced by $\arg\max$), this reduces to a standard embedding lookup; for softmax-relaxed latents, it becomes a linear combination of vocabulary embeddings. As a result, the model is no longer asked to denoise from a fully corrupted discrete token embedding, but instead receives an embedding that is a superposition of clean and noisy token embedding. This "partially clean" input provides a direct signal about the underlying token, making denoising easier

than relying solely on the surrounding context. Fig. 3 (top) illustrates this curriculum. More formally, during the curriculum phase Sahoo et al. (2025a) optimize the following loss, where the softmax is applied independently at each position:

$$\mathcal{L}^{\text{train}} = \mathbb{E}_{\mathbf{x},t\sim\mathcal{U}[\beta,\gamma],\bar{q}_t} \sum_{\ell\in[L]} f\Big(\mathbf{z}_t^{\ell} := \arg\max(\mathbf{w}_t^{\ell}), \mathbf{x}_{\theta}^{\ell}(\text{softmax}(\mathbf{w}_t/\tau),t), \alpha_t := \mathcal{T}(\tilde{\alpha}_t); \mathbf{x}^{\ell}\Big). \quad (9)$$

Notice that $\mathcal{L}^{\text{train}}$ in (9) reduces to the NELBO (8) in the limit $\lim_{\tau\to 0}$, for $\beta = 0$ and $\gamma = 1$, since $\lim_{\tau\to 0} \text{softmax}(\mathbf{v}/\tau) = \arg\max(\mathbf{v})$, as shown by Jang et al. (2017); Maddison et al. (2017). However, explicitly materializing the high-dimensional latents $\mathbf{w}_t$ is memory-intensive, an issue we address in Sec. 4.

## 2.2 Diffusion Guidance

For continuous data, diffusion models have achieved state-of-the-art controllable generation through both classifier-based guidance (Sohl-Dickstein et al., 2015; Dhariwal & Nichol, 2021) and Classifier-Free Guidance (CFG; Nichol & Dhariwal (2021); Ho & Salimans (2022)). These approaches have since been extended to discrete data (Gruver et al., 2023). Let $y \in \{1,\dots,C\}$ denote one of $C$ possible classes. For CFG, the sampling posterior $p_{\theta}^{(\gamma)}$, which modulates the strength of the guidance term via the temperature parameter $\gamma$, is defined as (Nisonoff et al., 2024; Schiff et al., 2025):

$$\log p_{\theta}^{(\gamma)}(\mathbf{z}_s^{\ell} \mid y, \mathbf{z}_t) = \gamma \log p_{\theta}(\mathbf{z}_s^{\ell} \mid y, \mathbf{z}_t) + (1-\gamma)\log p_{\theta}(\mathbf{z}_s^{\ell} \mid \emptyset, \mathbf{z}_t); \ \forall\ell\in[L], \quad (10)$$

where $\emptyset$ denotes no class conditioning, and $p_{\theta}$ is the generative posterior (Sec. 2.1).

## 3 THE $\Psi$-POSTERIORS

Multiple joint distributions can give rise to the same marginals as the discrete diffusion process defined in (1). In this work, we introduce a family of posteriors, denoted $\Psi$, that share the same marginals as in (1); see Suppl. A.2 for details. **These alternative generative processes are non-Markovian and apply both to the Masked diffusion processes and to the Uniform-state diffusion processes.** Specifically, we define the posteriors for the generative process as:

$$\Psi_{s|t}(.\mid\mathbf{x}^{\ell},\mathbf{z}_t^{\ell}) = \kappa_t q_{s|t}(.\mid\mathbf{z}_t^{\ell},\mathbf{x}^{\ell}) + (1-\kappa_t)q_s(.\mid\mathbf{x}^{\ell}); \ \forall\ell\in[L] \quad (11)$$

where $\kappa_t \in [0,1]$ and $\Psi_1(.\mid\mathbf{x}^{\ell}) = \text{Cat}(.\mid\boldsymbol{\pi})$, with $\boldsymbol{\pi} = \mathbf{m}$ for MDMs and $\boldsymbol{\pi} = \mathbf{1}/K$ for USDMs. (11) is thus a linear combination of the forward process (1) and the reverse posteriors (2, 3) of standard discrete diffusion models. We therefore refer to these as *superposition posteriors*, or simply $\Psi$-posteriors.

**$\Psi$-Forward Processes**  Consider the interpolating diffusion process in (1) discretized into $T$ steps. Let $\mathbf{z}_{t(i)}$ denote the latent variables at times $t(i) = i/T$ for $0 \le i \le T$. The distribution of a trajectory $\mathbf{z}_{0:1}$ factorizes independently over tokens as: $\Psi(\mathbf{z}_{0:1}|\mathbf{x}) = \prod_{\ell} \Psi(\mathbf{z}_{0:1}^{\ell}|\mathbf{x}^{\ell})$ where $\Psi(\mathbf{z}_{0:1}^{\ell}|\mathbf{x}^{\ell}) = \Psi_1(\mathbf{z}_1^{\ell}|\mathbf{x}^{\ell})\prod_{i=1}^{T} \Psi_{s|t}(\mathbf{z}_{s(i)}^{\ell}|\mathbf{z}_{t(i)}^{\ell},\mathbf{x}^{\ell})$. In what follows, we use $s,t$ as shorthand for $s(i),t(i)$, respectively. The forward process can be derived from Bayes' rule: $\Psi(\mathbf{z}_t^{\ell}|\mathbf{z}_s^{\ell},\mathbf{x}^{\ell}) = \Psi(\mathbf{z}_s^{\ell}|\mathbf{z}_t^{\ell},\mathbf{x}^{\ell})\Psi(\mathbf{z}_t^{\ell}|\mathbf{x}^{\ell})/\Psi(\mathbf{z}_s^{\ell}|\mathbf{x}^{\ell})$. Unlike the Markovian interpolating process in (1), this forward process is generally not Markovian, since each $\mathbf{z}_t^{\ell}$ may depend on both $\mathbf{z}_s^{\ell}$ and $\mathbf{x}^{\ell}$.

**$\Psi$-Reverse Processes**  In Suppl. A.1, we show that the approximate reverse posterior takes the form:

$$[\Psi_{s|t}^{\theta}(.\mid\mathbf{z}_t)]^{\ell} = \kappa_t q_{s|t}(.\mid\mathbf{z}_t^{\ell},\mathbf{x}_{\theta}^{\ell}(\mathbf{z}_t,t)) + (1-\kappa_t)\left[\alpha_s q_{0|t}(.\mid\mathbf{z}_t^{\ell},\mathbf{x}_{\theta}^{\ell}(\mathbf{z}_t,t)) + (1-\alpha_s)\boldsymbol{\pi}\right]. \quad (12)$$

where $\mathbf{x}_{\theta}$ denotes the denoising model. We dub (12) as $\Psi$-sampler. For $(\kappa_t = 1)_{t\in[0,1]}$, we recover the standard ancestral sampler defined in (2) for MDMs and (3) for USDMs. Notice that for $\kappa_t < 1$, $\Psi_{s|t}$ corresponds to a noisier version of the ancestral sampler marginal $q_{s|t}$. This is analogous to Predictor-Corrector methods in Gaussian diffusion (Song et al., 2021), where the corrector introduces additional Gaussian noise. In our case, $q_t$ plays the role of the corrector, while $q_{s|t}$ acts as the predictor. The $\Psi$-posteriors also admit a principled NELBO formulation (see Suppl. A.3), though this is not directly relevant for sampling.

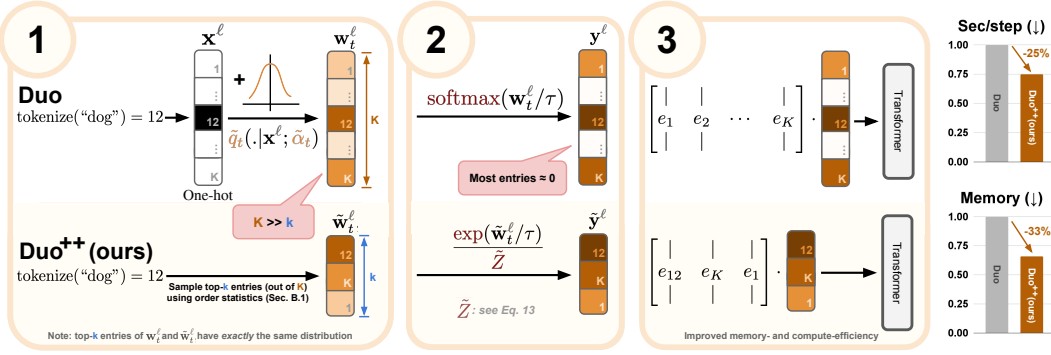

Figure 3: **Efficient Curriculum for USDMs.** Duo (Sahoo et al., 2025a) replaces discrete lookups with linear combinations of all $K$ embeddings: (1) Gaussian diffusion on one-hot representations, (2) Low-temperature softmax, (3) weighted sum. **Duo$^{++}$** exploits the sparsity of the tempered softmax (most weights are effectively zero), and simulate the k largest entries (out of K) using ordered statistics. The approximate normalizer $\tilde{Z}$ admits a closed form expression (14). **Duo$^{++}$ has a 33% lower memory and 25% faster training than Duo.**

**Corollary** For $\boldsymbol{\pi} = \mathbf{m}$, different choices of $\{\kappa_t\}_{t \in [0,1]}$ recover previous Predictor-Corrector formulations in the literature (Campbell et al., 2022; Gat et al., 2024; Wang et al., 2025) (see Suppl. A.4 for the proof). The $\Psi$ framework thus subsumes these samplers as special cases, extending these predictor-corrector methods for discrete diffusion with any prior $\boldsymbol{\pi}$.

> **Takeaway 1: $\Psi$-samplers generalize prior Predictor-Corrector methods to arbitrary noise priors**, subsuming ReMDM (Wang et al., 2025) and prior work (Campbell et al., 2022; Gat et al., 2024) as special cases.

**Intuitive Explanation** In practice, the denoiser $\mathbf{x}_\theta$ imperfectly models the clean data $\mathbf{x}$. The key to the effectiveness of $\Psi$-sampler is the offset term $(1 - \kappa_t)(1 - \alpha_s)\boldsymbol{\pi}$ in (12), which enables error correction during generation. For MDMs ($\boldsymbol{\pi} = \mathbf{m}$), this offset allows previously denoised tokens to return to the masked state, unlike the ancestral sampler, which prevents remasking (see Sec. 2.1.1). Incorrect tokens can thus be replaced with better ones. For USDMs ($\boldsymbol{\pi} = \mathbf{1}/K$), the offset ensures every token has non-zero sampling probability. Even if the denoiser assigns near-zero probability to the correct token, the $\Psi$-sampler gives it a chance to appear, whereas ancestral sampling would not. While this offset may occasionally introduce incorrect tokens, the marginals of the $\Psi$-samplers (11) match those of the Markovian forward process (1), hence we converge to the correct distribution given sufficient samples.

## 4 SCALABLE CURRICULUM FOR FASTER TRAINING

Recall from Sec. 2.1.2 that the curriculum of Sahoo et al. (2025a) accelerates training by replacing discrete tokens as inputs to the transformer with softmaxed Gaussian latents. Naively, however, this requires materializing a $K$-dimensional weight vector for every token at every training step, which is infeasible for modern LLM vocabularies with $K > 100{,}000$ (Touvron et al., 2023; OpenAI, 2024). Our key observation is that Sahoo et al. (2025a) use a very low temperature $\tau = 10^{-3}$ in (8). At such temperatures, the softmax concentrates almost all its mass on only a few entries, making most of the $K$ weights negligible. We exploit this induced sparsity by approximating the full linear combination using only $k \ll K$ embeddings (Fig. 3, bottom). We illustrate the process hereby:

**Step 1: Generating top-$k$ entries in $\mathbf{w}_t^\ell$ w/o materializing it** Let $o \in [K]$ be the nonzero coordinate of the one-hot vector $\mathbf{x}^\ell$, i.e. $(\mathbf{x}^\ell)_o = 1$. Recall that $\mathbf{w}_t^\ell = \tilde{\alpha}_t \mathbf{x}^\ell + \tilde{\sigma}_t \boldsymbol{\epsilon}$, with $\tilde{\sigma}_t = \sqrt{1 - \tilde{\alpha}_t^2}$ and $\boldsymbol{\epsilon} \sim \mathcal{N}(\mathbf{0}, \mathbf{I}_K)$. Therefore, $(\mathbf{w}_t^\ell)_o \sim \mathcal{N}(\tilde{\alpha}_t, \tilde{\sigma}_t^2)$ and $(\mathbf{w}_t^\ell)_{i \neq o} \sim \mathcal{N}(0, \tilde{\sigma}_t^2)$. All coordinates in $[K] \setminus o$ are exchangeable (i.i.d. with mean 0), while the coordinate $o$ is the only one with a shifted mean. As

a result, the top-$k$ set falls into one of two cases: Case 1: $o$ is not among the top-$k$ , so all $k$ winners lie in $[K] \setminus o$. Case 2: $o$ is among the top-$k$ , so the winners are $o$ plus $k-1$ indices from $[K] \setminus o$. Next, we describe how to sample the top-$k$ values (and later their indices) without ever forming the full $K$-dimensional vector.

Let $m = K-1$ and consider $m$ i.i.d. samples $\tilde{w}_1, \ldots, \tilde{w}_m \sim \mathcal{N}(0, \tilde{\sigma}_t^2)$. Rather than drawing all $m$ values, we exploit the fact that order statistics of i.i.d. *uniform* random variables can be sampled recursively: the maximum of $m$ i.i.d. $\mathcal{U}[0,1]$ variables has CDF $u^m$, so it can be sampled directly. Conditioned on the maximum, the remaining values are also i.i.d. uniforms, so the next-largest can be sampled the same way, and so on. Applying the inverse normal CDF $\Phi^{-1}(\cdot) \cdot \tilde{\sigma}_t$ converts the top-$k$ uniform order statistics into the top-$k$ Gaussian values (see Suppl. B.1.4 for details). We denote the result $\mathcal{K} = (\mathcal{K}_1 \geq \cdots \geq \mathcal{K}_k)$ , where $\mathcal{K}_j$ is the $j$-th largest among the $K-1$ zero-mean coordinates.

We draw independently the "special" entry $\tilde{w} \sim \mathcal{N}(\tilde{\alpha}_t, \tilde{\sigma}_t^2)$, which matches the distribution of $(\mathbf{w}_t^\ell)_o$. Now compare $\tilde{w}$ to the current $k$-th largest value among the zero-mean coordinates: If $\mathcal{K}_k > \tilde{w}$, then $o$ cannot enter the top-$k$. We are in Case 1, and $\mathcal{K}$ already contains the correct top-$k$ values. If $\tilde{w} > \mathcal{K}_k$, then $o$ must be in the top-$k$. We are in Case 2. Let $r = \left| \{ j \in [k] : \mathcal{K}_j > \tilde{w} \} \right|$ be the number of values in $\mathcal{K}$ that are larger than $\tilde{w}$. We insert $\tilde{w}$ at rank $r+1$ and drop the previous smallest value: $\mathcal{K} \leftarrow \mathcal{K}_{1:r} \| \tilde{w} \| \mathcal{K}_{r+1:k-1}$ , where $\|$ denotes concatenation.

For clarity, let $\binom{S}{m}$ denote the set of all possible tuples with $m$ distinct elements in the set $S$. We generate the index tuple $\mathcal{I}$ corresponding to the top-$k$ values $\mathcal{K}$ as follows: Case 1: all top-$k$ indices lie in $[K] \setminus o$; hence, $\mathcal{I} \sim \binom{[K]\setminus\{o\}}{k}$. Case 2: the index $o$ appears at position $r+1$ in $\mathcal{K}$. Sample the indices for the remaining $k-1$ values from $[K] \setminus o$, split into those above and below $o$: $\mathcal{I} = L \| (o) \| R$ , where $L \sim \binom{[K]\setminus\{o\}}{r}, R \sim \binom{[K]\setminus\{o\}\setminus L}{k-r-1}$. This produces both the top-$k$ values and their matching indices while sampling only $O(k)$ random variables, without constructing the full $K$-dimensional vector.

**Step 2: Approximating the softmax**  Given the top-$k$ values and indices $(\mathcal{K}, \mathcal{I})$ from Step 1, we approximate the softmax-weighted embedding vector by retaining only the $k$ selected rows of the `embeddings` $\in \mathbb{R}^{K \times d}$ ($d$ denotes the embedding size):

$$\text{softmax}(\mathbf{w}_t^\ell)^\top \texttt{embeddings} \approx \sum_{i=1}^{k} \frac{\exp(\mathcal{K}_i/\tau)}{\tilde{Z}} \texttt{embeddings}[\mathcal{I}_i], \qquad (13)$$

where `embeddings`$[j]$ denotes the $j$-th row. The normalizer $\tilde{Z}$ includes both sampled (top-$k$) and unsampled terms. While each unsampled term is small, their sum may be non-negligible, hence we approximate it as (Suppl. B.2):

$$\tilde{Z} \approx \underbrace{\sum_{i=1}^{k} \exp\left(\frac{\mathcal{K}_i}{\tau}\right)}_{\text{top-}k\text{ terms}} + \underbrace{\delta \exp\left(\frac{\tilde{w}}{\tau}\right)}_{\text{clean token}} + \underbrace{(K - k - \delta) \exp\left(\frac{\tilde{\sigma}_t^2}{2\tau^2} - \log \Phi\left(\frac{\mathcal{K}_k}{\tilde{\sigma}_t}\right) + \log \Phi\left(\frac{\mathcal{K}_k - \tilde{\sigma}_t^2/\tau}{\tilde{\sigma}_t}\right)\right)}_{\text{unsampled zero-mean terms}},$$

$$(14)$$

where $\delta = 1$ if $\tilde{w} \in \mathcal{K}$ ( case 2 ) and 0 otherwise. We provide the full derivation in Suppl. B.2 and pseudocode in Algo. 3.

Lastly, the curriculum objective in (9) requires evaluating the diffusion transformation operator $\mathcal{T}(\cdot)$. Directly computing $\mathcal{T}$ via (6) is prohibitively expensive during training; Sahoo et al. (2025a) therefore precompute and cache many $(\alpha_t, \mathcal{T}(\tilde{\alpha}_t))$ pairs, which is cumbersome. Instead, we compute $\mathcal{T}(\cdot)$ on the fly using its Taylor expansion; see Suppl. B.3.1.

## 5 EXPERIMENTS

We evaluate Duo$^{++}$ with $\Psi$-samplers on language modeling (Sec. 5.1.1) and image generation (Sec. 5.1.2), showing that $\Psi$-samplers markedly improve text and image quality, making USDMs

Table 1: **Accuracy on multiple-choice question answering datasets.** Abbreviations: Arc-e (ARC-Easy), Arc-c (ARC-Challenge), HSwag (HellaSwag), WinoG (Winogrande), PIQA (Physical Intelligence Question Answering), OQA (OpenBookQA). [†]Results from Deschenaux et al. (2025). Duo[++] ($k = 2$) achieves slightly higher accuracy than Duo on 4 out of 6 tasks. Overall, **Duo[++] matches Duo's performance while using 25% fewer flops**. The highest accuracy among USDMs is **bolded**. The absolute best per column is underlined.

|  | Arc-e | Arc-c | HSwag | WinoG | PIQA | MathQA | OQA |
|---|---|---|---|---|---|---|---|
| AR Transformer | 38.55 | 23.04 | 30.55 | 53.19 | 63.60 | 22.41 | 32.4 |
| MDLM[†] | 34.26 | 24.66 | 31.54 | 51.93 | 57.89 | 20.70 | 28.60 |
| Duo | 28.11 | 25.43 | 26.46 | 47.20 | 51.14 | 20.00 | 23.40 |
| Duo[++] ($k = 2$) | 27.32 | **26.11** | 26.26 | 49.64 | **52.12** | 20.40 | **27.80** |
| Duo[++] ($k = 3$) | **28.28** | 25.00 | 25.89 | 47.36 | 50.65 | **21.01** | 23.00 |
| Duo[++] ($k = 5$) | 28.03 | 25.77 | **26.90** | **50.12** | 51.25 | 20.20 | 25.40 |

outperform MDMs in sample quality. In Sec. 5.2, we show that Duo[++] matches Duo (Sahoo et al., 2025a) while using 33% less memory and training 25% faster, enabled by our efficient curriculum strategy (Sec. 4).

## 5.1 $\Psi$-SAMPLERS

### 5.1.1 LANGUAGE MODELING

> **Takeaway 2: $\Psi$-samplers substantially improve Generative Perplexity** for USDMs, with gains especially pronounced when NFEs exceed the sequence length.
>
> **Takeaway 3:** Unlike ancestral sampling, which plateaus, **$\Psi$-samplers continue to improve with more sampling steps**, closing the gap with Masked diffusion models.

**Experimental Settings** We compare MDLM (Sahoo et al., 2024) and ReMDM (Wang et al., 2025) with Duo[++] and $\Psi$-samplers. We use the original checkpoints of Sahoo et al. (2024), trained for 1M steps with a batch size of 512 on OpenWebText (OWT; Gokaslan & Cohen (2019)) and context length $L = 1024$. Duo[++] is trained with the same context length, batch size and number of steps, but with the efficient curriculum. Refer to the original works for more details. We measure the sample quality using the Gen. PPL ($\downarrow$) computed with GPT-2 Large (Radford et al., 2019) and the diversity using the unigram entropy ($\uparrow$) (Dieleman et al., 2022; Sahoo et al., 2024; 2025a). We cast logits to 64-bit precision for sampling (Zheng et al., 2025). See Suppl. C.1 for more details.

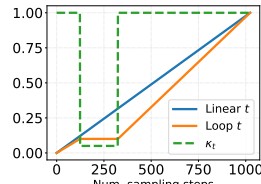

Figure 4: Illustration of the possible evolution of $t$ and the associated $\kappa_t$. In practice, we use $\kappa_t$ close to 1 during the PC phase.

**Results and Ablation** Fig. 1 (left) shows the Gen. PPL and entropy as a function of the NFE. Duo[++] with $\Psi$-samplers outperforms MDLM with ReMDM and ancestral sampling across the entire range of NFEs. As the number of NFEs increases beyond the sequence length, ReMDM and $\Psi$-samplers further improve sample quality while ancestral sampling plateaus. We ablate on the $\kappa_t$ schedule type (cap, rescale, loop, see Suppl. A.5), the step-size parameter $\eta$, the nucleus sampling threshold $p \in \{0.9, 0.95, 1.0\}$ (Suppl. D.1). The rescale schedule with $\eta = 0.05$ yields the best Gen. PPL while preserving the unigram entropy. Nucleus sampling ($p$=0.9) consistently improves Gen. PPL for both MDLM and Duo, as observed in Wang et al. (2025).

**How to Pick $\kappa_t$?** We recommend the ReMDM-equivalent rescale schedule with $\eta = 0.05$ and nucleus sampling ($p = 0.9$), using the log-linear noise schedule with linearly decreasing $t$, which outperforms the "loop" strategy (Suppl. A.5).

### 5.1.2 IMAGE MODELING

> **Takeaway 4: On CIFAR-10, Duo$^{++}$ with $\Psi$-samplers achieves better FID and IS than** MDLM with both ancestral sampling and ReMDM.

**Experimental Setup** We train the same 35M-parameter U-Net as Austin et al. (2023) on CIFAR-10 for 1.5M steps. Following Schiff et al. (2025), the U-Net is made class conditional and we sample with Discrete Classifier-free Guidance (CFG; Ho & Salimans (2022); Schiff et al. (2025)). See Suppl. C.1 for full training details. We report the *Fréchet Inception Distance* (FID; Heusel et al. (2018)) and *Inception Score* (IS; Salimans et al. (2016)) between the training set and generated samples.

**Results and Ablation** Fig. 1 *(right)* and Fig. 6 show that $\Psi$-samplers and ReMDM substantially improve FID and IS compared to ancestral sampling, with Duo$^{++}$ reaching the best scores overall. We ablate on the sampling noise schedule (cosine vs. log-linear), $\kappa_t$, the activation range $[t_{\text{off}}, t_{\text{on}}]$, nucleus sampling, and the $\kappa_t$ schedules (including the ReMDM variants; see Suppl. A.5). Full results are in Suppl. D.1. Using $\kappa_t$ close to 1 (light noise injection) with a cosine schedule achieves the best FID and IS, and Duo$^{++}$ tolerates stronger noise injection than MDLM. With ancestral sampling, nucleus sampling improves the FID in the low NFE regime for MDLM, and always helps for Duo. Since it is detrimental to MDLM at high NFE, we do not use nucleus sampling when using the $\Psi$-samplers, for both MDLM and Duo.

**How to Pick $\kappa_t$?** We recommend a cosine sampling schedule with $\kappa_t = 0.95$, $t_{\text{on}} \in \{0.5, 0.6\}$, $t_{\text{off}} = 0.1$ for Duo$^{++}$, and $\kappa_t = 0.99$, $t_{\text{on}} = 1.0$, $t_{\text{off}} = 0.1$ for MDLM. We suggest using a piecewise constant $\kappa_t$ with linearly decreasing $t$, rather than the ReMDM loop schedule (Suppl. A.5), setting $\kappa_t < 1$ when $t \in [t_{\text{off}}, t_{\text{on}}]$, since it outperforms the ReMDM schedules.

## 5.2 FAST CURRICULUM

> **Takeaway 5: The efficient curriculum reduces peak memory by 33% and training time by** **25%**, while matching the performance of Duo on likelihood benchmarks and downstream tasks.

**Experimental Settings** We train Duo$^{++}$ with the scalable curriculum (Sec. 4) on OpenWebText (OWT; Gokaslan & Cohen (2019)) and LM1B (Chelba et al., 2014). We train all models for 1M steps, using a batch size of 512. For LM1B, we use the `bert-base-uncased` tokenizer with a context length of 128, padding shorter sequences. This setup follows previous work (Sahoo et al., 2024; Lou et al., 2024; He et al., 2022). For OWT, we use the GPT-2 tokenizer (Radford et al., 2019), and reserve the last 100k documents for validation, following Sahoo et al. (2025a; 2024). We follow Lou et al. (2024) and use a modified diffusion transformer (DiT) (Peebles & Xie, 2023) with rotary positional encoding (Su et al., 2023). We evaluate the impact of $k = \{2, 3, 5\}$ during the efficient curriculum. All models are trained on 16 H100 GPUs with bfloat16 precision. Training uses the loss in (9), with $\tau = 0.001$ and $(\beta, \gamma) = (0.03, 0.15)$ for the first 500K steps (Sahoo et al., 2025a).

**Likelihood Results** Table 2 shows that on both LM1B and OWT, our efficient curriculum Duo$^{++}$ matches the performance of Duo with its expensive curriculum. The lowest validation perplexity is achieved with $k = 2$, although $k \in \{2, 3, 5\}$ performs similarly. We also compare

Table 2: Test perplexity (PPL) on LM1B and OWT. Lower is better. [†]Results from Sahoo et al. (2025a). Best Uniform-state diffusion numbers are **bolded**. Duo and Duo$^{++}$ achieve comparable performance across both datasets while requiring 25% fewer GPU-hours (Table 4), demonstrating the effectiveness of our memory-efficient curriculum.

|  | LM1B | OWT |
|---|---|---|
| *Autoregressive* | | |
| Transformer[†] | 22.3 | 17.5 |
| *Masked Diffusion* | | |
| SEDD Absorb[†] | 32.7 | 24.1 |
| MDLM[†] | 27.0 | 23.2 |
| *Uniform-state Diffusion* | | |
| SEDD Uniform[†] | 40.3 | 29.7 |
| UDLM[†] | 31.3 | 27.4 |
| Duo[†] | **29.9** | **25.2** |
| Duo$^{++}$ (Ours), $k = 2$ | 30.0 | **25.2** |
| Duo$^{++}$ (Ours), $k = 3$ | 30.1 | 25.3 |
| Duo$^{++}$ (Ours), $k = 5$ | 30.2 | 25.4 |

the models trained on OWT in Zero-Shot perplexity, and find that Duo$^{++}$ achieves a performance comparable to Duo. That is, we evaluate on the validation splits of the Penn Treebank (Marcus et al., 1993), WikiText (Merity et al., 2016), LM1B (Chelba et al., 2014), LAMBADA (Paperno et al., 2016), AG News (Zhang et al., 2016) and scientific articles from ArXiv and PubMed (Cohan et al., 2018). Table 5 shows that Duo$^{++}$ reaches a zero-shot probability similar to that of Duo *while requiring 25% less training GPU-hours.*

**Likelihood-based Downstream Tasks**    In Table 1, we compare the multiple-choice question (MCQ) accuracy of Duo, Duo$^{++}$, MDLM (Sahoo et al., 2024), and an autoregressive transformer (1M training steps with a batch size of 512 on OWT, same hyperparameters as MDLM) using the `lm-eval-harness` suite (Gao et al., 2024) ; details in Suppl. C.3. We find that Duo$^{++}$ achieves an accuracy similar to that of Duo, despite requiring 25% less training GPU-hours. However, it trails MDLM on most tasks, consistent with its higher perplexity.

**Throughput and Peak Memory Usage**    Table 4 reports the throughput and peak memory usage for Duo and Duo$^{++}$. Duo$^{++}$ reduces the peak memory usage by about 33% and doubles the speed of the Curriculum Learning phase. When applying Curriculum Learning for half of the training steps, Duo$^{++}$ trains 25% faster than Duo on the 138M-parameter scale. Notably, both peak memory usage and throughput remain stable over the full training run when $k \in \{2, 3, 5\}$.

# 6    RELATED WORK AND DISCUSSION

**Compatibility with General Discrete Diffusion Processes**    This work focuses on discrete diffusion with uniform or masked noise. However, our approach extends to more general discrete diffusion processes (Shaul et al., 2024; von Rütte et al., 2025; Holderrieth et al., 2025) featuring a combination of masked and uniform prior, since we provide a general predictor–corrector algorithm for discrete diffusion with arbitrary noise.

**Predictor-Corrector Samplers**    In the context of Masked diffusion, ReMDM (Wang et al., 2025) generalizes previous predictor-corrector methods (Campbell et al., 2022; 2024; Gat et al., 2024) that were based on Continuous Time Markov Chain formulation of discrete diffusion processes. Our approach further generalizes ReMDM to support arbitrary diffusion processes. Unlike Lezama et al. (2023); Zhao et al. (2025); Liu et al. (2025); Kim et al. (2025), who train an additional corrector module, our method does not introduce additional learned components.

**Comparison to Other Discrete Diffusion Samplers**    Park et al. (2024) uses noise-adaptive step sizes; while we use uniform steps, $\Psi$-samplers support any step-size schedule. Ren et al. (2025) develops higher-order samplers; we use only first-order information, though the posterior in (11) could be approximated with higher-order methods. Thus, $\Psi$-samplers complement both lines of work.

# 7    CONCLUSION

We introduced a unified and practical framework for predictor-corrector sampling in discrete diffusion language models through $\Psi$-posteriors. By linearly superposing the forward and reverse diffusion processes (11), the $\Psi$-*posteriors* preserve the marginals of standard diffusion models. Importantly, the $\Psi$-*posteriors* and associated $\Psi$-samplers subsume prior masked-diffusion PC samplers (Campbell et al., 2022; Gat et al., 2024; Wang et al., 2025) as special cases, and naturally extend to discrete diffusion models with uniform prior. Empirically, Duo$^{++}$ with $\Psi$-samplers matches the performance of MDMs on natural language generation and achieves stronger FID and IS scores on CIFAR-10. Moreover, they exhibit superior scaling: performance continues to improve with NFEs, unlike ancestral samplers, which plateau. Finally, we propose a scalable training curriculum (Sahoo et al., 2025a) that reduces the peak memory usage by 33% and shortens the training time by 25%. Concurrently, Sahoo et al. (2026) show that Duo surpasses an autoregressive model at the 1.7B scale on the math and reasoning benchmark (GSM8K). Taken together, these results challenge the view that Masked diffusion is categorically the future of diffusion language modeling.

## 8 ACKNOWLEDGEMENTS

This work has received funding from the Swiss State Secretariat for Education, Research and Innovation (SERI). We are grateful to Ricky T. Q. Chen and Zhihan Yang for insightful discussions and suggestions. We acknowledge the SCITAS team at EPFL for providing access to their cluster, and the Swiss National Supercomputing Centre for the Alps platform. We are grateful to Karin Gétaz for her administrative assistance.

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

## CONTENTS

# A  $\Psi$-POSTERIORS

## A.1  APPROXIMATE REVERSE MARGINALS

We parameterize the (generative) $\Psi$-reverse marginals to have a similar form as the true posterior (11). Therefore, the generative reverse marginals also factorize over the sequence length. Because $\mathbf{x}^{1:L}$ is not available during sampling, there are two terms in (11) that are intractable. First, we choose to replace the posterior $q_{s|t}(.|\mathbf{z}_t^\ell, \mathbf{x}^\ell)$ by $q_{s|t}(.|\mathbf{z}_t^\ell, \mathbf{x}^\ell = \mathbf{x}_\theta^\ell)$. Additionally, as we cannot sample from $q_s(.|\mathbf{x}^\ell)$ without $\mathbf{x}^\ell$, we replace $\mathbf{x}^\ell$ by $q_{0|t}(.|\mathbf{z}_t, \mathbf{x}^\ell = \mathbf{x}_\theta^\ell), \forall \ell \in [L]$. Replacing these two intractable terms yields our generative reverse marginals:

$$\Psi_{s|t}^\theta(.|\mathbf{z}_t) = \kappa_t q_{s|t}(.|\mathbf{z}_t, \mathbf{x} = \mathbf{x}_\theta(\mathbf{z}_t, t)) + (1 - \kappa_t)\left[\alpha_s q_{0|t}(.|\mathbf{z}_t, \mathbf{x} = \mathbf{x}_\theta(\mathbf{z}_t, t)) + (1 - \alpha_s)\pi\right].$$
(15)

Note that for the masked posterior (2), $q_{0|t}(.|\mathbf{z}_t, \mathbf{x} = \mathbf{x}_\theta(\mathbf{z}_t, t)) = \mathbf{x}_\theta(\mathbf{z}_t, t)$.

## A.2  PROOF THAT THE $\Psi$-POSTERIORS HAVE THE CORRECT MARGINALS

Let $\Psi_{s|t}(.|\mathbf{x}^\ell, \mathbf{z}_t^\ell)$ denote the $\Psi$-posteriors defined in (11). Let $s$ denote $s(k) = t(k - 1)$ and $t$ denote $t(k)$. To prove that the $\Psi$-posteriors have the correct marginals, we proceed by (downwards) induction, similar to Song et al. (2022). First, note that $\Psi_s(\mathbf{z}_s^\ell|\mathbf{x}^\ell)$ can be written as a marginalization over $\tilde{\mathbf{z}}_t^\ell$, for $s < t$:

$$\Psi_s(\mathbf{z}_s^\ell|\mathbf{x}^\ell) = \sum_{\tilde{\mathbf{z}}_t^\ell} \Psi_t(\tilde{\mathbf{z}}_t^\ell|\mathbf{x}^\ell)\Psi_{s|t}(\mathbf{z}_s^\ell|\tilde{\mathbf{z}}_t^\ell, \mathbf{x})$$
(16)

**Base Case**  Let $\Psi_1(\mathbf{z}_1^\ell|\mathbf{x}^\ell)$ denote the marginal at time $t = 1$. By definition in (11), $\Psi_1(\mathbf{z}_1^\ell|\mathbf{x}^\ell) = \text{Cat}(.|\boldsymbol{\pi})$. Therefore, the $\Psi$-posteriors have the correct marginal for $t = 1$.

**Induction hypothesis**  Suppose that the $\Psi$-posteriors have the correct marginal for a certain $t \leq 1$, that is, $\Psi_t(.|\mathbf{x}^\ell) = q_t(.|\mathbf{x}^\ell)$.

**Inductive step**  Based on the induction hypothesis, we now show that $\Psi_s(.|\mathbf{x}^\ell) = q_s(.|\mathbf{x}^\ell)$, for $s(k) = t(k - 1)$. Indeed

$$\Psi_s(.|\mathbf{x}^\ell) \overset{(1)}{=} \sum_{\tilde{\mathbf{z}}_t^\ell} \Psi_t(\tilde{\mathbf{z}}_t^\ell|\mathbf{x}^\ell)\Psi_{s|t}(\mathbf{z}_s^\ell|\tilde{\mathbf{z}}_t^\ell, \mathbf{x}^\ell)$$

$$\overset{(2)}{=} \sum_{\tilde{\mathbf{z}}_t^\ell} q_t(\tilde{\mathbf{z}}_t^\ell|\mathbf{x}^\ell)\Psi_{s|t}(\mathbf{z}_s^\ell|\tilde{\mathbf{z}}_t^\ell, \mathbf{x}^\ell)$$

$$\overset{(3)}{=} \sum_{\tilde{\mathbf{z}}_t} q_t(\tilde{\mathbf{z}}_t^\ell|\mathbf{x}^\ell)\left[\kappa_t q_{s|t}(\mathbf{z}_s^\ell|\mathbf{x}^\ell, \tilde{\mathbf{z}}_t^\ell) + (1 - \kappa_t)q_s(\mathbf{z}_s^\ell|\mathbf{x}^\ell)\right]$$

$$\overset{(4)}{=} \kappa_t \sum_{\tilde{\mathbf{z}}_t^\ell} q_t(\tilde{\mathbf{z}}_t^\ell|\mathbf{x}^\ell)q_{s|t}(\mathbf{z}_s^\ell|\mathbf{x}^\ell, \tilde{\mathbf{z}}_t^\ell) + (1 - \kappa_t)q_s(\mathbf{z}_s^\ell|\mathbf{x}^\ell)\sum_{\tilde{\mathbf{z}}_t^\ell} q_t(\tilde{\mathbf{z}}_t^\ell|\mathbf{x}^\ell)$$

$$\overset{(5)}{=} \kappa_t q_s(\mathbf{z}_s^\ell|\mathbf{x}^\ell) + (1 - \kappa_t)q_s(\mathbf{z}_s^\ell|\mathbf{x}^\ell) = q_s(\mathbf{z}_s^\ell|\mathbf{x}^\ell).$$

Specifically, (1) hold by (16), (2) by the induction hypothesis, (3) by definition of the $\Psi$-posteriors, (4) by distributing $q_t(\tilde{\mathbf{z}}_t^\ell|\mathbf{x}^\ell)$, (5) by definition of marginal probability (first term), and by observing that $\sum_{\tilde{\mathbf{z}}^\ell} q_t(\tilde{\mathbf{z}}_t^\ell|\mathbf{x}^\ell) = 1$ since $q_t$ is normalized. This concludes the inductive step, and shows that the $\Psi$-posteriors have the correct marginal.

## A.3 NEGATIVE EVIDENCE LOWER BOUND

Let $\mathbf{z}_{0:1}^\ell$ denote a reverse trajectory with time indices $\{0, \frac{1}{T}, \frac{2}{T}, \ldots, 1\}$ for token $\ell$. The joint distribution of $(\mathbf{x}^\ell, \mathbf{z}_{0:1}^\ell)$ under the generative model factorizes as

$$p^\theta(\mathbf{x}^\ell, \mathbf{z}_{0:1}^\ell) = p(\mathbf{x}^\ell \mid \mathbf{z}_0^\ell)\Psi_1(\mathbf{z}_1^\ell)\prod_{i=1}^{T}\Psi_{s|t}^\theta(\mathbf{z}_{s(i)}^\ell \mid \mathbf{z}_{t(i)}^\ell), \tag{17}$$

where each pair $(s(i), t(i))$ denotes one reverse transition with $s(i) < t(i)$. The marginal likelihood is

$$p^\theta(\mathbf{x}^\ell) = \sum_{\mathbf{z}_{0:1}^\ell} p^\theta(\mathbf{x}^\ell, \mathbf{z}_{0:1}^\ell). \tag{18}$$

Introducing the variational distribution $\Psi(\mathbf{z}_{0:1}^\ell \mid \mathbf{x}^\ell) = \Psi_1(\mathbf{z}_1^\ell \mid \mathbf{x}^\ell)\prod_{i=1}^{T}\Psi_{s|t}(\mathbf{z}_{s(i)}^\ell \mid \mathbf{z}_{t(i)}^\ell, \mathbf{x}^\ell)$, Jensen's inequality results in:

$$-\log p^\theta(\mathbf{x}^\ell) \le \mathbb{E}_{\Psi(\mathbf{z}_{0:1}^\ell \mid \mathbf{x}^\ell)}\left[-\log p(\mathbf{x}^\ell \mid \mathbf{z}_0^\ell)\right] + \mathrm{KL}\left(\Psi_1(\cdot \mid \mathbf{x}^\ell)\|\Psi_1\right) \tag{19}$$

$$+ \sum_{i=1}^{T}\mathbb{E}_{\Psi(\mathbf{z}_{t(i)}^\ell \mid \mathbf{x}^\ell)}\left[D_{\mathrm{KL}}\left(\Psi_{s|t}(\cdot \mid \mathbf{z}_{t(i)}^\ell, \mathbf{x}^\ell)\,\|\,\Psi_{s|t}^\theta(\cdot \mid \mathbf{z}_{t(i)}^\ell)\right)\right]. \tag{20}$$

This expression is similar to the standard diffusion NELBO, with a reconstruction term, a prior term at $t=1$, and a sum of KL divergences. As $T \to \infty$, $p(\mathbf{x}^\ell \mid \mathbf{z}_0^\ell)$ concentrates around $\mathbf{x}^\ell$, hence $-\log p(\mathbf{x}^\ell \mid \mathbf{z}_0^\ell) \to 0$. Furthermore, the prior term is zero by definition of the $\Psi$-posteriors in (11).

## A.4 RECOVERING PREDICTOR-CORRECTOR METHODS FOR MASKED DIFFUSION

Wang et al. (2025) introduce the ReMDM posterior, which generalizes the MDM posterior (2) by allowing previously decoded tokens to be remasked. For a given position $\ell$, the ReMDM posterior is

$$q_\sigma(\mathbf{z}_s^\ell \mid \mathbf{z}_t^\ell, \mathbf{x}^\ell) = \begin{cases} \mathrm{Cat}\big(.; (1 - \sigma_t)\,\mathbf{x}^\ell + \sigma_t\,\mathbf{m}\big), & \mathbf{z}_t^\ell \ne \mathbf{m}, \\ \mathrm{Cat}\left(.; \frac{\alpha_s - (1-\sigma_t)\alpha_t}{1-\alpha_t}\,\mathbf{x}^\ell + \frac{1-\alpha_s - \sigma_t\alpha_t}{1-\alpha_t}\,\mathbf{m}\right), & \mathbf{z}_t^\ell = \mathbf{m}, \end{cases} \tag{21}$$

where $\sigma_t \in [0, \sigma_t^{\max}]$ is a free parameter that controls the remasking probability. The upper bound $\sigma_t^{\max} := \min\{1, (1 - \alpha_s)/\alpha_t\}$ ensures that (21) defines a valid distribution. When $\sigma_t = 0$, the ReMDM posterior reduces to the standard MDM posterior. Below, we show that the $\Psi$-posteriors recover the ReMDM posterior with the substitution $\kappa_t = 1 - \sigma_t/(1 - \alpha_s)$. Suppose that we work with Masked diffusion, hence $\pi = \mathbf{m}$. The $\Psi$-posteriors can be expanded as

$$\Psi_{s|t}(.|\mathbf{z}_t^\ell)$$

$$= \kappa_t q_{s|t}(.|\mathbf{z}_t^\ell, \mathbf{x}^\ell) + (1 - \kappa_t)\left[\alpha_s q_{0|t}(.|\mathbf{z}_t^\ell, \mathbf{x}^\ell) + (1 - \alpha_s)\pi\right] \tag{22}$$

$$= \kappa_t \begin{cases} \mathrm{Cat}(.; \mathbf{z}_t^\ell), & \mathbf{z}_t^\ell \ne \mathbf{m}, \\ \mathrm{Cat}\left(.; \dfrac{(1 - \alpha_s)\mathbf{m} + (\alpha_s - \alpha_t)\mathbf{x}^\ell}{1 - \alpha_t}\right), & \mathbf{z}_t^\ell = \mathbf{m} \end{cases} + (1 - \kappa_t)\left[\alpha_s\mathbf{x}^\ell + (1 - \alpha_s)\mathbf{m}\right] \tag{23}$$

$$\overset{(1)}{=} \kappa_t \begin{cases} \mathrm{Cat}(.; \mathbf{x}^\ell), & \mathbf{z}_t^\ell \ne \mathbf{m}, \\ \mathrm{Cat}\left(.; \dfrac{(1 - \alpha_s)\mathbf{m} + (\alpha_s - \alpha_t)\mathbf{x}^\ell}{1 - \alpha_t}\right), & \mathbf{z}_t^\ell = \mathbf{m} \end{cases} + (1 - \kappa_t)\left[\alpha_s\mathbf{x}^\ell + (1 - \alpha_s)\mathbf{m}\right] \tag{24}$$

$$= \begin{cases} \mathrm{Cat}(.; \kappa_t\mathbf{x}^\ell + (1 - \kappa_t)[\alpha_s\mathbf{x}^\ell + (1 - \alpha_s)\mathbf{m}]), & \mathbf{z}_t^\ell \ne \mathbf{m} \\ \mathrm{Cat}\left(.; \kappa_t\frac{(1-\alpha_s)\mathbf{m}+(\alpha_s-\alpha_t)\mathbf{x}^\ell}{1-\alpha_t} + (1 - \kappa_t)[\alpha_s\mathbf{x}^\ell + (1 - \alpha_s)\mathbf{m}]\right), & \mathbf{z}_t^\ell = \mathbf{m} \end{cases} \tag{25}$$

$$= \begin{cases} \mathrm{Cat}(.; [\kappa_t + (1 - \kappa_t)\alpha_s]\mathbf{x}^\ell + (1 - \kappa_t)(1 - \alpha_s)\mathbf{m}), & \mathbf{z}_t^\ell \ne \mathbf{m} \\ \mathrm{Cat}\left(.; \left[\kappa_t\frac{\alpha_s-\alpha_t}{1-\alpha_t} + (1 - \kappa_t)\alpha_s\right]\mathbf{x}^\ell + \left[\kappa_t\frac{1-\alpha_s}{1-\alpha_t} + (1 - \kappa_t)(1 - \alpha_s)\right]\mathbf{m}\right), & \mathbf{z}_t^\ell = \mathbf{m} \end{cases}, \tag{26}$$

where (1) holds since $\mathbf{z}_t^\ell \ne \mathbf{m}$ implies that $\mathbf{z}_t^\ell = \mathbf{x}^\ell$, since in Masked diffusion, the latents $\mathbf{z}_t^\ell$ are either a clean token or the masked token.

To conclude, if we pick $\kappa_t = 1 - \frac{\sigma_t}{1 - \alpha_s}$, where $\sigma_t$ is the free parameter in the ReMDM sampler, then the equation reduces to the ReMDM posterior. Therefore, the $\Psi$-posteriors generalize ReMDM, which itself generalized the FB (Campbell et al., 2022) and DFM (Gat et al., 2024) posteriors. Additionally, the $\Psi$-posteriors are not limited to Masked diffusion, as we showed in this work. In Table 11, we sample from the ReMDM-equivalent $\Psi$-samplers, comparing the official ReMDM implementation and our version, and find obtain similar performance.

### A.5 ReMDM Sampling Schedules

Wang et al. (2025) introduce three schedules for the remasking parameter $\sigma_t$, which controls how aggressively previously generated tokens are remasked. As shown in Suppl. A.4, the $\Psi$-samplers recover ReMDM when $\kappa_t = 1 - \sigma_t/(1 - \alpha_s)$, and $(\sigma_t)_{t=0}^1$ denotes the ReMDM noise injection schedule. Therefore, each ReMDM $\sigma_t$ schedule has a direct $\Psi$-samplers equivalent. Below, we present the three $\sigma_t$ schedules studied in Wang et al. (2025). The schedules must satisfy $0 \leq \sigma_t \leq \sigma_t^{\max} := \min\{1, (1 - \alpha_s)/\alpha_t\}$ to ensure that the reverse posterior remains a valid distribution, and are generally defined in terms of $\sigma_t^{\max}$ and a parameter $\eta \in [0, 1]$.

**Cap Schedule** With the "Cap" schedule, the remasking probability is capped at a constant $\eta \in [0, 1]$, as long as it remains in the valid bounds:

$$\sigma_t = \min\{\eta, \sigma_t^{\max}\}. \tag{27}$$

**Rescale Schedule** With the "Rescale" schedule, $\sigma_t$ is obtained by scaling the upper bound $\sigma_t^{\max}$ by a constant $\eta \in [0, 1]$:

$$\sigma_t = \eta \cdot \sigma_t^{\max}. \tag{28}$$

**Loop Schedule** Unlike the cap and rescale schedules, which only modulate $\sigma_t$, the "Loop" schedule also changes the evolution of $t$ during sampling (see Fig. 4 for an illustration). It is controlled by the time boundaries $t_{\mathrm{on}} > t_{\mathrm{off}}$. The noise schedule $\alpha_t$ is reparametrized to be piecewise linear. First, $\alpha_t$ increases linearly from 0 to $\alpha_{t_{\mathrm{on}}}$ when $t > t_{\mathrm{on}}$. Secondly, when $t \in [t_{\mathrm{off}}, t_{\mathrm{on}}]$, it is held constant to $\alpha_{t_{\mathrm{on}}}$, and finally increases from $\alpha_{t_{\mathrm{on}}}$ to 1 when $t < t_{\mathrm{off}}$. When $t \notin [t_{\mathrm{off}}, t_{\mathrm{on}}]$, ReMDM samples from the MDM posterior (2). When $t \in [t_{\mathrm{off}}, t_{\mathrm{on}}]$, since $\alpha_t = \alpha_s = \alpha_{t_{\mathrm{on}}}$, the model samples from the ReMDM posterior (21) with a constant $\sigma_t = \eta$. In practice, $t_{\mathrm{on}}$ is chosen so that $\alpha_{t_{\mathrm{on}}}$ is close to 1, where most tokens have already been decoded.

## B Fast Curriculum

In this section, we expand on the implementation of the efficient curriculum. In Sec. B.4, we present the overall design, pseudocode, and the three main implementation challenges. Our approach relies on several mathematical results, which we present in order of dependency. We first recall inverse transform sampling (Sec. B.1.1), then derive the distributions of the largest (Sec. B.1.2) and second largest (Sec. B.1.3) uniform order statistics. These results enable generating the $k$ largest Gaussian random variables out of $K$ without materializing the full vector (Sec. B.1.4). We also derive a closed-form expression for the conditional mean of the exponential of a Gaussian random variable (Sec. B.2), used to estimate the softmax normalizer.

Furthermore, although the efficient curriculum could be implemented using the original definition of the Diffusion Transformation Operator $\mathcal{T}$, we show that $\mathcal{T}$ admits a convenient series expansion in Sec. B.3.1. This avoids the need to precompute 100k function values, and simplifies the implementation. Finally, in Sec. B.3.2, we show that $\mathcal{T}$ can be well approximated by a degree-9 polynomial, which removes the need to store a large number of coefficients during training.

### B.1 Sampling the top-$k$ values in $\mathbf{w}_t^\ell$ Without Materializing The Full Vector

Computing our curriculum step requires access to the $k$ largest entries of the diffused weight vector $\mathbf{w}_t^\ell \in \mathbb{R}^K$, but explicitly materializing $\mathbf{w}_t^\ell$ (and then running a full top-$k$) is prohibitively expensive when $K$ is large. In this subsection, we show how to obtain the top-$k$ values and their associated indices while using only $\mathcal{O}(k)$ memory and without simulating all $K$ random variables. The key

idea is to decouple *values* from *locations*: we first sample the top-$k$ Gaussian order statistics directly via inverse transform sampling (Sec. B.1.1), leveraging closed-form expressions for uniform order statistics (Sec. B.1.2, Sec. B.1.3) and a numerically stable log-space implementation (Algo. 1). We then assign these top-$k$ to their corresponding indices (Sec. B.1.5, Algo. 2). This yields an efficient routine whose cost scales with $k$ rather than $K$, enabling top-$k$ truncation of $\mathbf{w}_t^l$ without ever materializing it.

### B.1.1  INVERSE TRANSFORM SAMPLING

The Inverse Transform Sampling method (Devroye, 1986) is an algorithm for generating a continuous random variables $X$ with a known Cumulative Distribution Function (CDF) $F_X$. Implementing Inverse Transform Sampling requires access to the inverse CDF $F_X^{-1}$, and a source of $i.i.d$ uniform random variables. If $X = F_X^{-1}(U)$, where $U \sim \mathcal{U}[0,1]$, then $X \sim F_X$. Indeed, for $x \in \mathbb{R}$,

$$\mathbb{P}(X \leq x) = \mathbb{P}(F_X^{-1}(U) \leq x) = \mathbb{P}(U \leq F_X(x)) = F_X(x), \tag{29}$$

since for $a \in [0,1]$, $\mathbb{P}(U \leq a) = a$. This shows that $X$ has the correct distribution.

### B.1.2  DISTRIBUTION OF THE LARGEST UNIFORM RANDOM VARIABLE OUT OF $K$

The distribution of the largest uniform random variable out of $K$ admits a simple closed-form expression:

**Proposition B.1** (Distribution of the largest uniform random variable out of $K$). $U^{(1)} \geq U^{(2)} \geq ... \geq U^{(K)}$ *denote an order statistic over $K$ i.i.d uniform random variables $\mathcal{U}([0,\theta])$ with Cumulative Density Function (CDF) $F_U$. Suppose that $u \in [0,1]$, then $F_U(u) = \frac{u}{\theta}$. Then, the CDF $F_{U^{(1)}}$ and probability density function (PDF) $f_{U^{(1)}}$ of the largest random variable $U^{(1)}$ are as follows:*

$$\begin{aligned} F_{U^{(1)}}(u) &= F_U^K(u) = u^K \theta^{-K} \\ f_{U^{(1)}}(u) &= K F_U^{K-1}(u) f_U(u) = K u^{K-1} \theta^{-K} \end{aligned} \tag{30}$$

*Proof.*

$$F_{U^{(1)}}(u) = \mathbb{P}(U^{(1)} \leq u) = \mathbb{P}(U_i \leq u)_{\forall i \in [K]} = P(U \leq u)^K = F_U^K(u). \tag{31}$$

The PDF is obtained by differentiation:

$$f_{U^{(1)}}(u) = \frac{d}{du} F_{U^{(1)}}(u) = K F_U^{K-1}(u) f_U(u), \tag{32}$$

$\square$

### B.1.3  DISTRIBUTION OF THE $k$ LARGEST UNIFORM RANDOM VARIABLE OUT OF $K$

In this part, we show how to sample the $k$ largest uniform random variables out of $K$. Importantly, we do not need to draw all $K$ values and sort them, which would be impractical for large $K$. Let $U^{(1)} \geq \cdots \geq U^{(K)}$ denote the order statistics of $K$ i.i.d. $\mathcal{U}[0,1]$ random variables. We argue that for $1 \leq i < K$, conditioned on $U^{(i)} = u^{(i)}$, $U^{(i+1)}$ is distributed as the largest out of $K - i$ uniform random variables on $[0, u^{(i)}]$. This enables an iterative scheme to generate the $k$ largest variables in decreasing order. The argument relies on two standard results: the conditional density formula (Prop. B.2) and the joint density of a pair of order statistics (Prop. B.3). The conditional distribution of $U^{(i+1)} \mid U^{(i)} = u^{(i)}$ is given in Prop. B.4.

**Proposition B.2** (Conditional Density (Berger & Casella, 2001)). *Let $X, Y$ be two random variables with joint density $f_{X,Y}$ and marginals $f_X, f_Y$. Then, the conditional density of $X$ given $Y = y$ is*

$$f_{X|Y=y}(x|y) = \frac{f_{X,Y}(x,y)}{f_Y(y)}. \tag{33}$$

**Proposition B.3** (Joint Density of Order Statistics (Berger & Casella (2001); proof in Border (2021))). *Let $X^{(1)} \geq \cdots \geq X^{(K)}$ denote the order statistics of $K$ random variables with CDF $F$ and PDF $f$, arranged in descending order. Then, the joint density of $X^{(n)}$ and $X^{(m)}$, where $n < m$ (so that $X^{(n)} \geq X^{(m)}$), is given by*

$$f_{X^{(n)}, X^{(m)}}(u, v) = \frac{K!}{(n-1)!(m-n-1)!(K-m)!} \left(1 - F(u)\right)^{n-1} \left(F(u) - F(v)\right)^{m-n-1} F(v)^{K-m} f(u) f(v).$$
(34)

**Proposition B.4** (Conditional Distribution of $U^{(i+1)}$ given $U^{(i)}$). *Let $U^{(1)} \geq \cdots \geq U^{(K)}$ denote the order statistics of $K$ independent and uniformly distributed random variables on $[0, 1]$, arranged in descending order. For any $1 \leq i < K$, conditioned on $U^{(i)} = u^{(i)}$, $U^{(i+1)}$ is distributed as the largest of $K - i$ i.i.d. uniform random variables on $[0, u^{(i)}]$.*

*Proof.* From Prop. B.3 with $k = i$ and $l = i + 1$, the joint density of $(U^{(i)}, U^{(i+1)})$ for $\mathcal{U}[0, 1]$ variables (where $F(u) = u$ and $f(u) = 1$) is

$$f_{U^{(i)}, U^{(i+1)}}(u^{(i)}, u^{(i+1)}) = \frac{K!}{(i-1)! \, (K-i-1)!} \left(1 - u^{(i)}\right)^{i-1} \left(u^{(i+1)}\right)^{K-i-1},$$
(35)

since $(F(u^{(i)}) - F(u^{(i+1)}))^{l-k-1} = 1$ as $l - k = 1$.

To apply Prop. B.2, we need the marginal density of $U^{(i)}$. The event $\{u^{(i)} \leq U^{(i)} \leq u^{(i)} + du\}$ requires that, among the $K$ i.i.d. draws, exactly $i - 1$ fall in $(u^{(i)} + du, 1]$, exactly one falls in $[u^{(i)}, u^{(i)} + du)$, and the remaining $K - i$ fall in $[0, u^{(i)})$. The number of such assignments is the multinomial coefficient $\frac{K!}{(i-1)! \, 1! \, (K-i)!}$, and the probability of each assignment is $\left(1 - u^{(i)} - du\right)^{i-1} \cdot du \cdot \left(u^{(i)}\right)^{K-i}$. Multiplying these terms gives

$$P(u^{(i)} \leq U^{(i)} \leq u^{(i)} + du) = \frac{K!}{(i-1)! \, (K-i)!} \left(1 - u^{(i)} - du\right)^{i-1} \cdot du \cdot (u^{(i)})^{K-i}.$$
(36)

By definition, $f_{U^{(i)}}(u^{(i)}) = \lim_{du \to 0} \frac{P(u^{(i)} \leq U^{(i)} \leq u^{(i)} + du)}{du}$. Since $(1 - u^{(i)} - du)^{i-1} \to (1 - u^{(i)})^{i-1}$ as $du \to 0$, we obtain

$$f_{U^{(i)}}(u^{(i)}) = \frac{K!}{(i-1)! \, (K-i)!} \left(1 - u^{(i)}\right)^{i-1} (u^{(i)})^{K-i}.$$
(37)

Applying Prop. B.2:

$$
\begin{aligned}
f_{U^{(i+1)}|U^{(i)}}(u^{(i+1)} \mid u^{(i)}) &= \frac{f_{U^{(i)}, U^{(i+1)}}(u^{(i)}, u^{(i+1)})}{f_{U^{(i)}}(u^{(i)})} \\
&= \frac{\frac{K!}{(i-1)! \, (K-i-1)!} \left(1 - u^{(i)}\right)^{i-1} \left(u^{(i+1)}\right)^{K-i-1}}{\frac{K!}{(i-1)! \, (K-i)!} \left(1 - u^{(i)}\right)^{i-1} (u^{(i)})^{K-i}} \\
&= (K - i) \frac{\left(u^{(i+1)}\right)^{K-i-1}}{(u^{(i)})^{K-i}},
\end{aligned}
$$
(38)

since the factors $K!/(i-1)!$ and $(1 - u^{(i)})^{i-1}$ cancel. This is precisely the density of the largest of $K - i$ i.i.d. $\mathcal{U}[0, u^{(i)}]$ random variables. $\square$

### B.1.4 GENERATING THE $k$ LARGEST GAUSSIAN RANDOM VARIABLES OUT OF $K$

We now show that it is possible to generate the $k$ largest Gaussian random variables out of $K$ via inverse transform sampling (Sec. B.1.1) as follows.

Given a single uniform random variable $U \sim \mathcal{U}[0, 1]$, one can obtain a standard Gaussian random variable $W = \Phi^{-1}(U)$, where $\Phi$ is the Gaussian CDF, via inverse transform sampling. Now

assume we have a sorted list of $K$ uniform random variables $U_1 \geq U_2 \geq ... \geq U_K$. Since $\Phi$ is a monotonically increasing function, the largest uniform random variable, $U_1$, is mapped to the largest Gaussian random variable, i.e. $\Phi^{-1}(U_1)$ is distributed as the largest Gaussian random variable out of $K$.

**Sampling The Largest**  As shown in Prop. B.1, the CDF of the largest uniform random variable out of $K$ has an analytical solution. For $u \in [0, 1]$, $P(U_1 \leq u) = u^K$, hence it can be generated via inverse transform sampling.

**Sampling The Second Largest**  Furthermore, the distribution of the second largest, conditioned on $U_1 = u_1$, also admits a closed-form solution (Sec. B.1.3): for $u_2 \in [0, u_1]$, it is given by $P(U_2 \leq u_2|U_1 = u_1) = u_2^{K-1} u_1^{-(K-1)}$, i.e. it is distributed as the largest uniform variable out of $K - 1$, supported on $[0, u_1]$.

**Sampling The $k^{\text{th}}$-largest**  More generally, the same argument shows that conditioned on $U_i = u_i$, the random variable $U_{i+1}$ is distributed as the largest uniform variable on $[0, u_i]$ out of $K - i + 1$. This shows that we can sample $U_1, ..., U_k$ in decreasing order and without simulating all the $K$ variables. Finally, the $k$ largest $U_i$ can be transformed into the $k$ largest standard Gaussians out of $K$ as $\{\Phi^{-1}(U_i)\}_{i=1}^{k}$. In practice, a naive implementation of inverse transform sampling is numerically unstable when $K$ is large. For stability, operations should be implemented in log-space. Algo. 1 shows the pseudocode of the log-space implementation.

---

**Algorithm 1** Reverse Sampling from Order Statistics of Gaussian Random Variables. Here $N$ corresponds to $K-1$ (the number of zero-mean entries) and $\sigma$ to $\tilde{\sigma}_t = \sqrt{1 - \tilde{\alpha}_t^2}$.

---

**Input** Number of variables $N$, standard deviation $\sigma$, number of top values $k$
Sample $U_\ell \sim \mathcal{U}(0, 1)$, for $N \geq \ell \geq N - k + 1$
Compute the random variables: $R_\ell = \frac{\log U_\ell}{\ell}$
Compute the cumulative sums: $P_\ell = \sum_{m=\ell}^{N} R_m$
Let $V_\ell = \exp(P_\ell)$, the $\ell$-th sample from the (uniform) order statistic.
Apply inverse normal CDF: $X^{(\ell)} = \Phi^{-1}(V_\ell) \cdot \sigma$
**return** $\{X^{(\ell)}\}_{\ell=N}^{N-k+1}$

---

### B.1.5 Sampling Integers Without Repetitions and Without Shuffling

Suppose that $\mathbf{x}$ denotes the one-hot vector of category $o$. By symmetry, after applying Gaussian diffusion to $\mathbf{x}$, all entries $(\mathbf{x}_j)_{j \neq o}$ such that follow the exact same distribution. Therefore, they have the same probability of being one of the top $k$ largest random variable.

To implement the curriculum, we must not only approximate the weights of the embedding combination but also select which embeddings to include. As described in Sec. 4, we sample $k$ random indices *without repetition* excluding $i$. If the random variable at position $o$, corresponding to the clean token, belongs to the top-$k$, we replace one of the sampled indices with $o$. Otherwise, we use the $k$ sampled indices directly.

A simple way to sample $k$ random indices without repetition is to shuffle a list of $K$ integers and take the first $k$. However, this defeats the purpose of our efficient curriculum, as it requires materializing large tensors. Instead, Floyd's algorithm (Bentley, 1999), given in Algo. 2, samples without repetition while avoiding shuffling. Although sequential with $k$ iterations, it is much faster than shuffling when $k \ll K$.

---

**Algorithm 2** Floyd's Algorithm for Sampling Without Repetition

---

**Input** Number of possible values $N$, number of samples $k$.
Initialize array $S$ of size $k$ to store samples
**for** $t = 0$ to $k - 1$ **do**
    Sample $j \sim \text{Randint}(0, N - k + t)$
    **if** $t > 0$ and $j$ appears in $S[0:t]$ **then**
        $S[t] \leftarrow N - k + t$ {Use largest remaining value}
    **else**
        $S[t] \leftarrow j$
    **end if**
**end for**
**return** $S$

---

### B.2 APPROXIMATING THE WEIGHTED SUM OF THE EMBEDDINGS

After extracting the top-$k$ values and indices $(\mathcal{K}, \mathcal{I})$ from $\mathbf{w}_t^\ell$, we approximate the softmax-weighted embedding by retaining only the $k$ selected rows of the `embeddings` $\in \mathbb{R}^{K \times d}$ ($d$ denotes the embedding size):

$$\text{softmax}(\mathbf{w}_t^\ell)^\top \texttt{embeddings} \approx \sum_{i=1}^k \frac{\exp(\mathcal{K}_i/\tau)}{\tilde{Z}} \texttt{embeddings}[\mathcal{I}_i], \tag{39}$$

where `embeddings`$[j]$ denotes the $j$-th row.

The normalizer $\tilde{Z}$ includes both sampled (top-$k$) and unsampled terms. To account for this contribution, let

$$\mu = \mathbb{E}[\exp(X/\tau) \mid X < \mathcal{K}_k] \tag{40}$$

denote the expectation that a r.v. $X$ with pmf $\mathcal{N}(0, \tilde{\sigma}_t^2)$ is less than $\mathcal{K}_k$.

We approximate $\tilde{Z}$ via two cases. Let $o$ be the index corresponding to the clean-token category in $\mathbf{x}^\ell$, and let $\tilde{w} \sim \mathcal{N}(\tilde{\alpha}_t, \tilde{\sigma}_t^2)$.

Case 1. If $o$ is not among the top-$k$ indices (i.e., $o \notin \mathcal{I}$, and consequently $\tilde{w} \notin \mathcal{K}$), then $\tilde{Z}$ includes: (i) the $k$ terms in $\mathcal{K}$, (ii) the explicit contribution from index $o$, and (iii) the remaining $K - k - 1$ unsampled terms, each approximated by $\mu$. Thus,

$$\tilde{Z} \approx \underbrace{\sum_{i=1}^k \exp\left(\frac{\mathcal{K}_i}{\tau}\right)}_{\text{top-}k\text{ terms}} + \underbrace{\exp\left(\frac{\tilde{w}}{\tau}\right)}_{\text{index } o} + \underbrace{(K - k - 1)\mu}_{\text{unsampled terms}}. \tag{41}$$

Case 2. If $o$ is among the top-$k$ indices (i.e., $o \in \mathcal{I}$, and hence, $\tilde{w} \in \mathcal{K}$), its contribution is already included in the top-$k$ sum, leaving $K - k$ unsampled terms. Hence,

$$\tilde{Z} \approx \underbrace{\sum_{i=1}^k \exp\left(\frac{\mathcal{K}_i}{\tau}\right)}_{\text{top-}k\text{ terms}} + \underbrace{(K - k)\mu}_{\text{unsampled terms}}. \tag{42}$$

Next, we derive a closed-form expression for $\mu$ in (40).

$$\mu = \log \mathbb{E}[\exp(X) \mid X < \mathcal{K}_k] = \frac{\tilde{\sigma}_t^2}{2\tau^2} - \log \Phi\left(\frac{\mathcal{K}_k}{\tilde{\sigma}_t}\right) + \log \Phi\left(\frac{\mathcal{K}_k - \tilde{\sigma}_t^2/\tau}{\tilde{\sigma}_t}\right). \tag{43}$$

*Proof.*

$$\mu = \log \mathbb{E}\left[\exp\left(\frac{X}{\tau}\right) \mid X < \mathcal{K}_k\right]$$

Applying change of variables $\bar{X} = X/\tau$; we get,

$$= \log \mathbb{E}[\exp(\bar{X}) \mid \bar{X} < \mathcal{K}_k/\tau]$$

$$= \log \int_{-\infty}^{\mathcal{K}_k/\tau} \exp(x) \frac{f_{\bar{X}}(x)}{\mathbb{P}(\bar{X} < \mathcal{K}_k/\tau)} dx$$

Substituting $\sigma := \tilde{\sigma}_t/\tau$, we get:

$$= \log\left[\frac{1}{\Phi(\mathcal{K}_k/\tilde{\sigma}_t)} \int_{-\infty}^{\mathcal{K}_k/\tau} \exp(x) \frac{1}{\sqrt{2\pi\sigma^2}} \exp\left(-\frac{x^2}{2\sigma^2}\right) dx\right]$$

$$= \log\left[\frac{1}{\Phi(\mathcal{K}_k/\tilde{\sigma}_t)} \frac{1}{\sqrt{2\pi\sigma^2}} \int_{-\infty}^{\mathcal{K}_k/\tau} \exp\left(-\frac{x^2}{2\sigma^2} + x\right) dx\right]$$

$$= \log\left[\frac{1}{\Phi(\mathcal{K}_k/\tilde{\sigma}_t)} \frac{1}{\sqrt{2\pi\sigma^2}} \int_{-\infty}^{\mathcal{K}_k/\tau} \exp\left(-\frac{1}{2\sigma^2}(x^2 - 2\sigma^2 x + \sigma^4 - \sigma^4)\right) dx\right]$$

$$= \log\left[\frac{\exp(\sigma^2/2)}{\Phi(\mathcal{K}_k/\tilde{\sigma}_t)} \frac{1}{\sqrt{2\pi\sigma^2}} \int_{-\infty}^{\mathcal{K}_k/\tau} \exp\left(-\frac{1}{2\sigma^2}(x - \sigma^2)^2\right) dx\right]$$

$$= \log\left[\frac{\exp(\sigma^2/2)}{\Phi(\mathcal{K}_k/\tilde{\sigma}_t)} \Phi\left(\frac{\mathcal{K}_k/\tau - \sigma^2}{\sigma}\right)\right]$$

$$= \frac{\sigma^2}{2} - \log \Phi\left(\frac{\mathcal{K}_k}{\tilde{\sigma}_t}\right) + \log \Phi\left(\frac{\mathcal{K}_k/\tau - \sigma^2}{\sigma}\right)$$

Substituting back $\sigma := \tilde{\sigma}_t/\tau$, we get:

$$= \frac{\tilde{\sigma}_t^2}{2\tau^2} - \log \Phi\left(\frac{\mathcal{K}_k}{\tilde{\sigma}_t}\right) + \log \Phi\left(\frac{\mathcal{K}_k - \tilde{\sigma}_t^2/\tau}{\tilde{\sigma}_t}\right) \tag{44}$$

This concludes our proof. $\qquad\square$

Finally, substituting the value of $\mu$ from (44) into (41) and (42), we obtain:

$$\tilde{Z} \approx \underbrace{\sum_{i=1}^{k} \exp\left(\frac{\mathcal{K}_i}{\tau}\right)}_{\text{top-}k\text{ terms}} + \underbrace{\delta \exp\left(\frac{\tilde{w}}{\tau}\right)}_{\text{clean token}} + \underbrace{(K - k - \delta) \exp\left(\frac{\tilde{\sigma}_t^2}{2\tau^2} - \log \Phi\left(\frac{\mathcal{K}_k}{\tilde{\sigma}_t}\right) + \log \Phi\left(\frac{\mathcal{K}_k - \tilde{\sigma}_t^2/\tau}{\tilde{\sigma}_t}\right)\right)}_{\text{unsampled zero-mean terms}},$$

$$\tag{45}$$

## B.3 Efficient computation of $\mathcal{T}$ during training

The curriculum objective in (9) requires evaluating the diffusion transformation operator $\mathcal{T}(\cdot)$. Directly computing $\mathcal{T}$ via (6) is prohibitively expensive during training; Sahoo et al. (2025a) therefore precompute and cache many $(\alpha_t, \mathcal{T}(\tilde{\alpha}_t))$ pairs, which is cumbersome. Instead, we compute $\mathcal{T}(\cdot)$ on the fly using its Taylor expansion, detailed below. This derivation relies on two propositions that justify swapping the order of (i) summation and integration (Prop. B.5) and (ii) differentiation and integration (Prop. B.6).

**Proposition B.5** (First Corollary of the Dominated Convergence Theorem (Folland (1999), Theorem 2.25))**.** *If the sum $\sum_{n=0}^{\infty} f_n(x)$ exists for all $x$ and there exists an integrable function $g(x)$ such that*

$$\left|\sum_{n=0}^{k} f_n(x)\right| \leq g(x) \tag{46}$$

*for all $k$, then*

$$\int_{-\infty}^{\infty} \sum_{n=0}^{\infty} f_n(x) dx = \sum_{n=0}^{\infty} \int_{-\infty}^{\infty} f_n(x) dx. \tag{47}$$

**Proposition B.6** (Second Corollary of the Dominated Convergence Theorem (Folland (1999), Theorem 2.27)). *Let $f(x,t)$ be differentiable in $t$ and suppose there exists a function $g(x,t)$ such that:*

1. $\left| \frac{\partial f(x,t)}{\partial t} \right| \leq g(x, t_0)$ *for all $x$ and $t$ in some neighborhood $|t - t_0| \leq \delta_0$*

2. $\int_{-\infty}^{\infty} g(x,t) dx < \infty$ *for all $t$*

*Then*

$$\frac{d}{dt} \int_{-\infty}^{\infty} f(x,t) dx = \int_{-\infty}^{\infty} \frac{\partial f(x,t)}{\partial t} dx \tag{48}$$

### B.3.1 SERIES REPRESENTATION OF $\mathcal{T}$ AND $\partial_t \mathcal{T}$

Evaluating the series expansion on the fly is faster than precomputing and caching $\mathcal{T}$ for many $(\alpha_t, \mathcal{T}(\tilde{\alpha}_t))$ pairs as done by Sahoo et al. (2025a). We can further speed the evaluation up by fitting a low-degree polynomial to the series, which we use in practice (see Suppl. B.3.2 for the approximation error analysis). Below, we derive the series expansion for $\mathcal{T}$ (Prop. B.7) and its time-derivative $\partial_t \mathcal{T}$ (Prop. B.8):

**Proposition B.7** (Series Expansion of the Diffusion Transformation Operator). *The diffusion transformation operator $\mathcal{T}$ can be expressed as:*

$$\mathcal{T}(\tilde{\alpha}_t) = \frac{K}{K-1} \left[ e^{-\nu_t^2/2} \sum_{n=0}^{\infty} \frac{\nu_t^n}{n!} M_n - \frac{1}{K} \right] \tag{49}$$

$\nu_t = \frac{\tilde{\alpha}_t}{\sqrt{1-\tilde{\alpha}_t^2}}$ *and* $M_n = \int_{-\infty}^{\infty} z^n \phi(z) \Phi^{K-1}(z) dz$.

*Proof.* Recall that the standard Gaussian PDF is given by

$$\phi(x) = \frac{1}{\sqrt{2\pi}} e^{-x^2/2}. \tag{50}$$

For notational convenience, let $\nu_t = \frac{\tilde{\alpha}_t}{\sqrt{1-\tilde{\alpha}_t^2}}$. We can rewrite $\phi(x - \nu_t)$ in terms of $\phi(x)$:

$$\phi(x - \nu_t) = \frac{1}{\sqrt{2\pi}} e^{-(x-\nu_t)^2/2} = \frac{1}{\sqrt{2\pi}} e^{-(x^2 - 2\nu_t x + \nu_t^2)/2} = \phi(x) e^{\nu_t x} e^{-\nu_t^2/2}. \tag{51}$$

Using the definition of the infinite series of $e^x$, we can expand $e^{\nu_t x}$:

$$\phi(x - \nu_t) = \phi(x) e^{-\nu_t^2/2} \sum_{n=0}^{\infty} \frac{\nu_t^n x^n}{n!}. \tag{52}$$

Substituting this into our original integral:

$$\int_{-\infty}^{\infty} \phi(z - \nu_t) \Phi^{K-1}(z) dz = \int_{-\infty}^{\infty} \phi(z) e^{-\nu_t^2/2} \sum_{n=0}^{\infty} \frac{\nu_t^n z^n}{n!} \Phi^{K-1}(z) dz \tag{53}$$

Since Prop. B.5 is satisfied, as the sum is the Taylor series of the exponential function, we can exchange the order of integration and summation. This leads to our final result:

$$\int_{-\infty}^{\infty} \phi(z - \nu_t) \Phi^{K-1}(z) dz = e^{-\nu_t^2/2} \sum_{n=0}^{\infty} \frac{\nu_t^n}{n!} \int_{-\infty}^{\infty} z^n \phi(z) \Phi^{K-1}(z) dz$$
$$= e^{-\nu_t^2/2} \sum_{n=0}^{\infty} \frac{\nu_t^n}{n!} M_n. \tag{54}$$

$\square$

**Advantages**   At this point, one might ask what is gained by expressing $\mathcal{T}$ as a series expansion. There are two key advantages. First, since $\mathcal{T}$ is intractable, Sahoo et al. (2024) resort to precomputing 100k evaluations, which can take up to two hours with the GPT-2 tokenizer. Second, they approximate the time derivative using finite differences. Crucially, observe that $M_n$ and $I_n$ in Prop. B.7 and B.8 are the only intractable components of the series expansion, and they are independent of the input $\tilde{\alpha}_t$. We find that the terms of the series decay to zero after roughly 150 terms (with slower decay as $t \to 1$). Thus, instead of pre-computing 100k evaluations of $\mathcal{T}$, it suffices to cache $M_n$ and $I_n$ for $n < 150$. In practice, this takes only a few seconds and can be performed at the start of training.

**Proposition B.8** (Time-Derivative of the Diffusion Transformation Operator). *The time-derivative of the diffusion transformation operator $\mathcal{T}$ can be expressed as:*

$$\frac{d}{dt}\mathcal{T}(\tilde{\alpha}_t) = \frac{K \cdot e^{-\nu_t^2/2}}{K-1} \frac{\tilde{\alpha}_t'}{(1-\tilde{\alpha}_t^2)^{3/2}} \sum_{n=0}^{\infty} \frac{\nu_t^n}{n!} \left[ I_n - \nu_t M_n \right] \tag{55}$$

*where $\nu_t$ and $M_n$ are defined as in Prop. B.7. Finally, $I_n = \int_{-\infty}^{\infty} z^{n+1}\phi(z)\Phi^{K-1}(z)dz$, and $\tilde{\alpha}_t'$ denotes the time-derivative of the Gaussian noise schedule $\tilde{\alpha}_t$.*

*Proof.*   We want to compute

$$\frac{d}{d\nu_t}\mathcal{T}(\tilde{\alpha}_t) = \frac{K}{K-1} \frac{d}{d\nu_t} \int \phi(z - \nu_t)\Phi^{K-1}(z)dz. \tag{56}$$

To justify passing the derivative under the integral, we verify the conditions of Prop. B.6. Define

$$f(z,t) = \phi\left(z - \frac{\tilde{\alpha}_t}{\sqrt{1-\tilde{\alpha}_t^2}}\right)\Phi^{K-1}(z) = \phi(z - \nu_t)\,\Phi^{K-1}(z), \tag{57}$$

which has time derivative

$$\frac{\partial f(z,t)}{\partial t} = \frac{(z-\nu_t)\phi(z-\nu_t)}{(1-\tilde{\alpha}_t^2)^{3/2}}\Phi^{K-1}(z). \tag{58}$$

We need to find a suitable dominating function $g$. Let $1 > \delta_0 > 0$ and choose $t_0 = \frac{1-\delta_0}{2}$. When $|t - t_0| \le \delta_0$, we have $t \in [t_0 - \delta_0, t_0 + \delta_0]$. Since $t_0 - \delta_0 < t_0 < 1$ and $t_0 + \delta_0 = \frac{1-\delta_0}{2} + \delta_0 < 1$, we are guaranteed that $t < 1$. This ensures that $\nu_t$ is finite. Because $\tilde{\alpha}_t \in [0,1)$ when $t < 1$, there exists a constant $C$, such that

$$C := \max_{|t-t_0|\le\delta_0} \frac{1}{(1-\tilde{\alpha}_t^2)^{3/2}} < \infty. \tag{59}$$

For $z \in \mathbb{R}$ and $|t - t_0| \le \delta_0$, we can bound the absolute value of the time derivative of $f$ as follows:

$$\left|\frac{\partial f(z,t)}{\partial t}\right| = \frac{|z-\nu_t|}{(1-\tilde{\alpha}_t^2)^{3/2}}\phi(z-\nu_t)\,\Phi^{K-1}(z)$$
$$\le C|z-\nu_t|\phi(z-\nu_t) = g(z,t).$$

Finally, for all $t \in [0,1)$:

$$\int_{-\infty}^{\infty} g(z,t)dz = C\int_{-\infty}^{\infty} |z-\nu_t|\phi(z-\nu_t)dz = C\int_{-\infty}^{\infty} |z|\phi(z)dz$$
$$= C\int_{-\infty}^{\infty} |z|\phi(z)dz = 2C\int_0^{\infty} z\phi(z)dz$$
$$= 2C\int_0^{\infty} z \cdot \frac{1}{\sqrt{2\pi}}e^{-z^2/2}dz \tag{60}$$
$$= \frac{2C}{\sqrt{2\pi}}\int_0^{\infty} ze^{-z^2/2}dz$$
$$= \frac{2C}{\sqrt{2\pi}} \cdot 1 = C\sqrt{\frac{2}{\pi}} < \infty,$$

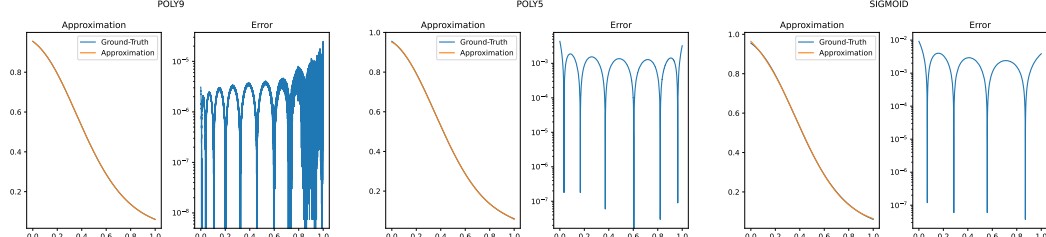

Figure 5: Polynomial approximation and approximation error, compared to the series approximation, truncated at 150 terms. The degree-9 polynomial (left) achieves orders of magnitude lower error than the degree-5 polynomial (center) and sigmoid (right) approximations.

where we used the substitution $u = z^2/2$ in the integral $\int_0^\infty ze^{-z^2/2}dz$ to obtain $\int_0^\infty e^{-u}du = 1$. The conditions of Prop. B.6 are satisfied, so we can pass the derivative under the integral.

Applying the derivative under the integral sign and using the identity $\phi(z - \nu_t) = \phi(z)e^{\nu_t z}e^{-\nu_t^2/2}$, we have:

$$
\begin{aligned}
\frac{d}{d\nu_t}\phi(z - \nu_t) &= \phi(z)\frac{d}{d\nu_t}[e^{\nu_t z - \nu_t^2/2}] \\
&= \phi(z)e^{\nu_t z - \nu_t^2/2}(z - \nu_t) \\
&= (z - \nu_t)\phi(z - \nu_t)
\end{aligned}
\tag{61}
$$

Therefore:

$$
\frac{d}{d\nu_t}\mathcal{T}(\tilde{\alpha}_t) = \frac{K}{K-1}\int_{-\infty}^\infty (z - \nu_t)\phi(z - \nu_t)\Phi^{K-1}(z)dz
\tag{62}
$$

Now using the Taylor series of $\phi(z - \nu_t)$, found earlier, and inverting the sum and integral as before, we find

$$
\begin{aligned}
\frac{d}{d\nu_t}\mathcal{T}(\tilde{\alpha}_t) &= \frac{K}{K-1}\int_{-\infty}^\infty (z - \nu_t)\phi(z)e^{\nu_t z}e^{-\nu_t^2/2}\Phi^{K-1}(z)dz \\
&= \frac{K \cdot e^{-\nu_t^2/2}}{K-1}\sum_{n=0}^\infty \frac{\nu_t^n}{n!}\left[\int_{-\infty}^\infty z^{n+1}\phi(z)\Phi^{K-1}(z)dz - \nu_t\int_{-\infty}^\infty z^n\phi(z)\Phi^{K-1}(z)dz\right] \\
&= \frac{K \cdot e^{-\nu_t^2/2}}{K-1}\sum_{n=0}^\infty \frac{\nu_t^n}{n!}\left[I_n - \nu_t M_n\right].
\end{aligned}
\tag{63}
$$

where $I_n = \int_{-\infty}^\infty z^{n+1}\phi(z)\Phi^{K-1}(z)dz$ and $M_n = \int_{-\infty}^\infty z^n\phi(z)\Phi^{K-1}(z)dz$. □

### B.3.2 POLYNOMIAL APPROXIMATION OF $\mathcal{T}$

Because the Diffusion Transformation Operator $\mathcal{T}$ has a sigmoid-like shape, we approximate it with S-shaped functions that require only a handful of coefficients. This allows us to store fewer parameters during training, instead of the 100k values required by the original curriculum or the 300 coefficients from the series approximation. Concretely, we test several functional forms with fewer than 10 parameters and fit them using non-linear least squares, via `scipy.optimize.curve_fit`.

As shown in Fig. 5, approximations tend to be less accurate at the boundaries, when $t \approx 0$ or $t \approx 1$. We find that the degree-9 polynomial works better than a sigmoid function of the form $a\sigma(bt + c) + d$, especially at the boundaries.

### B.4 IMPLEMENTATION OF THE FAST CURRICULUM

In Sec. 4, we described our efficient curriculum. Here we provide the pseudocode (Algo. 3) and elaborate on the three main implementation challenges:

- First, we need to sample the $k$ largest zero-mean Gaussian random variables out of $K$, to emulate the Gaussian Diffusion over the one-hot data samples $\mathbf{x}$ (Sec. B.1.4).

---

**Algorithm 3** Scalable Top-$k$ Curriculum Weights (Sec. 4).

Recall that $\mathbf{w}_t^\ell = \tilde{\alpha}_t \mathbf{x}^\ell + \tilde{\sigma}_t \boldsymbol{\epsilon}$, $\boldsymbol{\epsilon} \sim \mathcal{N}(\mathbf{0}, \mathbf{I}_K)$, where $\mathbf{x}^\ell$ is the one-hot vector representation of category $o$. Hence, $(\mathbf{w}_t^\ell)_o \sim \mathcal{N}(\tilde{\alpha}_t, \tilde{\sigma}_t^2)$ and the remaining $K{-}1$ entries are i.i.d. $\mathcal{N}(0, \tilde{\sigma}_t^2)$. The goal is to approximate $\mathrm{softmax}(\mathbf{w}_t^\ell / \tau)$ using only the $k$ largest entries ($k \ll K$).

---

**Input:** Clean token index $o \in [K]$, vocabulary size $K$, top-$k$ count $k$, temperature $\tau$, Gaussian schedule $(\tilde{\alpha}_t, \tilde{\sigma}_t)$

**Output:** Approximate softmax weights $\boldsymbol{\lambda} \in [0, 1]^k$ and corresponding token indices $\mathcal{I} \in [K]^k$

---

         $\triangleright$ Step 1: Sample the $k$ largest values of $\mathbf{w}_t^\ell$ without materializing all $K$ entries

$\mathcal{K}_1 \geq \cdots \geq \mathcal{K}_k \leftarrow$ top-$k$ order statistics of $(K{-}1)$ i.i.d. $\mathcal{N}(0, \tilde{\sigma}_t^2)$ draws      $\triangleright$ Algo. 1

$\tilde{w} \sim \mathcal{N}(\tilde{\alpha}_t, \tilde{\sigma}_t^2)$          $\triangleright$ Clean-token value: sample of $(\mathbf{w}_t^\ell)_o$

         $\triangleright$ Step 2: Build the top-$k$ set and approximate the softmax normalizer $\tilde{Z}$

**if** $\tilde{w} > \mathcal{K}_k$ **then**

         $\triangleright$ Case 2 : clean token $o$ belongs to the top-$k$

     $\mu \leftarrow \mathbb{E}[\exp(X/\tau) \mid X < \mathcal{K}_{k-1}], \quad X \sim \mathcal{N}(0, \tilde{\sigma}_t^2)$      $\triangleright$ Mean contribution unsimulated entry

     (Suppl. B.2)

     $r \leftarrow \big|\{j \in [k] : \mathcal{K}_j > \tilde{w}\}\big|$          $\triangleright$ Rank of $\tilde{w}$ in $\mathcal{K}$

     $\mathcal{K} \leftarrow (\mathcal{K}_{1:r}, \ \tilde{w}, \ \mathcal{K}_{r+1:k-1})$          $\triangleright$ Insert $\tilde{w}$, drop the smallest $\mathcal{K}_k$

     Sample $k{-}1$ indices $\mathcal{J}$ uniformly w/o replacement from $[K] \setminus \{o\}$      $\triangleright$ Algo. 2

     $\mathcal{I} \leftarrow (\mathcal{J}_{1:r}, \ o, \ \mathcal{J}_{r+1:k-1})$

     $\tilde{Z} \leftarrow \sum_{i=1}^k \exp(\mathcal{K}_i/\tau) + (K - k)\,\mu$      $\triangleright$ $K{-}k$ unsimulated entries, each contributing $\mu$

**else**

         $\triangleright$ Case 1 : clean token $o$ is *not* in the top-$k$

     $\mu \leftarrow \mathbb{E}[\exp(X/\tau) \mid X < \mathcal{K}_k], \quad X \sim \mathcal{N}(0, \tilde{\sigma}_t^2)$ $\triangleright$ Mean contribution unsimulated entry (Suppl. B.2)

     Sample $k$ indices $\mathcal{I}$ uniformly w/o replacement from $[K] \setminus \{o\}$      $\triangleright$ Algo. 2

     $\tilde{Z} \leftarrow \sum_{i=1}^k \exp(\mathcal{K}_i/\tau) + (K{-}k{-}1)\,\mu + \exp(\tilde{w}/\tau)$      $\triangleright$ $\tilde{w}$ counted exactly, $K{-}k{-}1$ via $\mu$

**end if**

         $\triangleright$ Step 3: Normalized softmax weights over the $k$ selected entries

$\boldsymbol{\lambda}_i \leftarrow \exp(\mathcal{K}_i/\tau) \,/\, \tilde{Z}, \quad i = 1, \ldots, k$

**return** $\boldsymbol{\lambda}$, $\mathcal{I}$

---

- Secondly, we must estimate the normalization constant of the softmax, without actually sampling the $K$ random variables (Sec. B.2).

- Third, we require an efficient method to sample $k$ distinct integers from $K$ without replacement (Sec. B.1.5).

Algo. 3 shows the pseudocode of the complete algorithm.

## C  EXPERIMENTAL DETAILS

### C.1  $\Psi$-SAMPLERS

#### C.1.1  OPENWEBTEXT

To evaluate the samplers, we use the pre-trained MDLM (Sahoo et al., 2024) and Duo (Sahoo et al., 2025a) checkpoints, as well as their distilled variants (using SDTT (Deschenaux & Gulcehre, 2025) and discrete consistency distillation, respectively, after 5 rounds of 10k steps). We re-state the training hyperparameters of both models in Suppl. C.2.1. For ReMDM, we use both the official implementation of Wang et al. (2025) and our re-implementation, which matches the original results while supporting additional sampling schedules beyond the log-linear one. See Suppl. D.1 for details on selecting $\kappa_t$.

### C.1.2   CIFAR10 (D3PM-LIKE ARCHITECTURE)

We train a U-Net backbone (Ronneberger et al., 2015) for 1.5M steps with a batch size of 128, using class conditioning with a class-dropout rate of 0.1 (as in Schiff et al. (2025)), and the default hyperparameters of Austin et al. (2023) (Table 3). For both MDLM and Duo, we experiment with time-conditional and unconditional variants, and train models using either cosine or log-linear noise schedules. See Table 6 for the ancestral-sampling evaluation of all variants after pre-training. See Suppl. D.1 for details on selecting $\kappa_t$.

Table 3: Model architecture on CIFAR10

| Component | Value |
|---|---|
| Vocab size | 256 |
| Number of ResNet blocks per scale | 2 |
| Base channels | 128 |
| Channel multiplier per scale | (1,2,2,2) |
| Attention resolutions | 16 |
| Conditional embedding dimension | 128 |
| Number of parameters | 35.8M |

### C.2   IMPROVED CURRICULUM

#### C.2.1   LANGUAGE MODELING

We adopt the same setup as prior work on discrete diffusion (Lou et al., 2024; Sahoo et al., 2024; 2025a), and restate it for completeness.

**LM1B**   We detokenize the One Billion Words (Chelba et al., 2014) as in Lou et al. (2024); Sahoo et al. (2024)[1], and tokenize it using the `bert-base-uncased` tokenizer (Devlin et al., 2019), as He et al. (2022). We use a context length of 128 and pad shorter documents.

**OpenWebText**   We tokenize OpenWebText (Gokaslan & Cohen, 2019) with the `GPT-2` tokenizer, concatenate sequences to a length of 1024, and insert an `eos` token between documents. Since the dataset lacks an official validation split, we reserve the last 100k documents for validation.

**Backbone**   We parameterize all models using the modified diffusion transformer architecture of Peebles & Xie (2023), following Lou et al. (2024); Sahoo et al. (2024). Our models use 12 layers, a hidden dimension of 768, 12 attention heads, and a timestep embedding of size 128 for the uniform-state diffusion variants. Word embeddings are not tied between input and output.

**Curriculum Lookup**   For the Duo baseline, we train models using the original code. To implement the efficient curriculum, we replace the full linear combination of embeddings by a sparse lookup, implemented using `torch.nn.functional.embedding_bag` to avoid materializing intermediate tensors. The curriculum phase lasts for the first 500k steps, after which we perform regular embedding table lookups, just like Sahoo et al. (2025a).

**Optimization**   We train all models with the AdamW optimizer (Loshchilov & Hutter, 2019) using a batch size of 512. The learning rate is linearly warmed up from 0 to $3 \times 10^{-4}$ over 2,500 steps, then kept constant for the remainder of training. We apply a dropout rate of 0.1 throughout.

### C.3   DOWNSTREAM EVALUATION PROTOCOL

We evaluate downstream performance using the `lm-eval-harness` library (Gao et al., 2024), following the protocol of Deschenaux et al. (2025). We focus on multiple choice tasks, where the log-likelihood of each candidate answer, given a prompt, is computed and the answer with the highest score is selected. For diffusion language models, which optimize a variational bound on the log-likelihood of the full sequence, we adapt the evaluation by using Bayes' rule:

$$\log p(\mathbf{y}_i | \mathbf{x}) = \log p(\mathbf{x}, \mathbf{y}_i) - \log p(\mathbf{x}) \propto \log p(\mathbf{x}, \mathbf{y}_i), \qquad (64)$$

Since $\log p(\mathbf{x})$ does not depend on the candidate $\mathbf{y}_i$, we simply select the answer that maximizes $\log p(\mathbf{x}, \mathbf{y}_i)$. In practice, we use the log-likelihood ELBO (4), estimated via Monte Carlo with 1024 samples, and choose the continuation $\mathbf{y}_i$ with the highest estimated likelihood.

---

[1]`https://github.com/louaaron/Score-Entropy-Discrete-Diffusion/blob/main/data.py`

### C.4 ZERO-SHOT LIKELIHOOD

Our setting is the same as used by Sahoo et al. (2025a). Specifically, we measure the likelihood of the models trained on OpenWebText using the validation splits of seven diverse datasets: Penn Tree Bank (PTB; Marcus et al. (1993)), Wikitext (Merity et al., 2016), One Billion Words (LM1B; Chelba et al. (2014)), Lambada (Paperno et al., 2016), AG News (Zhang et al., 2016), and Scientific Papers (Pubmed and Arxiv subsets; Cohan et al. (2018)). The datasets are detokenized following the protocol of Lou et al. (2024); Sahoo et al. (2025a). We wrap all sequences to a maximum length of 1024 tokens and do not insert `eos` tokens between them. Table 5 shows that we reach similar performance as Duo.

## D  ADDITIONAL EXPERIMENTAL RESULTS

In Suppl. D.1, we elaborate on the impact of $\kappa_t$ on the performance of the $\Psi$-samplers. In Suppl. D.2, we show that our efficient curriculum produces weights with the same marginal distributions as Sahoo et al. (2025a).

### D.1  TUNING $\kappa_t$ FOR THE $\Psi$-SAMPLERS

As discussed in Sec. 5.1, the choice of $\kappa_t$ is critical for strong performance. With a poor choice of $\kappa_t$, $\Psi$-samplers can underperform ancestral sampling. Below, we report all of our hyperparameter sweeps across datasets.

- We perform image modeling on CIFAR-10 using the U-Net architecture of Austin et al. (2023); Schiff et al. (2025), and use horizontal flipping as the sole data augmentation.
- We evaluate $\Psi$-samplers on OpenWebText (Gokaslan & Cohen, 2019) using the original checkpoint of MDLM (Sahoo et al., 2024) and Duo (Sahoo et al., 2025a).

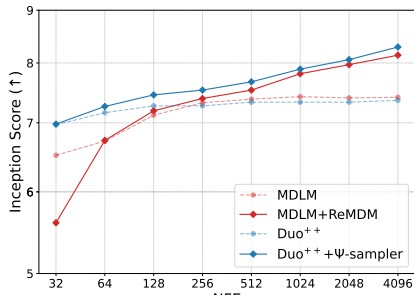

Figure 6: $\Psi$-samplers, which generalize ReMDM, significantly improve the Inception Score on CIFAR-10, compared to ancestral sampling.

#### D.1.1  CIFAR-10

We report FID (Heusel et al., 2018), computed between 50k generated samples and the training set. Before evaluating $\Psi$-samplers, we ablate on the training hyperparameters. Specifically, we train models with cosine and log-linear noise schedule, optionally with time-conditioning. We sample with both cosine and log-linear schedules. Finally, we check whether nucleus sampling (Holtzman et al., 2020) and greedy decoding on the final step can help, compared to vanilly ancestral sampling. Since nucleus sampling helps Duo but not MDLM, we compare the two models without nucleus sampling. Table 6 shows the validation perplexity and FID for a few number of sampling steps. Table 7 reports FID for ancestral sampling using step counts that are powers of two, from 32 up to 4096. Table 8 shows the results with ReMDM. Table 9 reports FID scores for $\Psi$-samplers using a stepwise-constant $\kappa$ schedule. Table 11 shows the performance of $\Psi$-samplers using the $\kappa$ schedule equivalent to ReMDM. We obtain similar results, which supports our theoretical claims.

- **MDLM (Ancestral).** Training with cosine noise schedule and time conditioning yields the best validation perplexity and FID.
- **MDLM (ReMDM).** We find that ReMDM improves the best FID over ancestral sampling, from 24.73 to 23.71 using 4096 sampling steps. Nucleus sampling can help at very low step counts, but the best performance is obtained with ancestral sampling. As the number of steps increases, nucleus sampling *worsens* the FID.
- **Duo (Ancestral).** Cosine training without time conditioning yields the lowest perplexity, while log-linear training without time conditioning gives the best FID. We use the latter in downstream experiments. Nucleus sampling improves FID, and greedy decoding slightly worsens it.

Table 4: Training efficiency comparison between Duo and Duo$^{++}$ on 138M parameter models. All measurements are conducted on a training job on 8 NVIDIA GH200-120GB GPU with batch size 32. We report the average throughput in sequence per second. The row "Duo (after CL)" denotes the resources consumption of Duo after the Curriculum phase. The impact of $k$ is minimal when $k \in \{2, 3, 5\}$, and Duo$^{++}$ uses similar resources.

| Method | Throughput (samples/s) $\uparrow$ | Peak Memory (GiB) $\downarrow$ |
|---|---|---|
| Duo | 81.8 | 94.3 |
| Duo (after the CL) | 122.4 | 63.3 |
| Duo$^{++}$ ($k \in \{2, 3, 5\}$) | 121.9 | 63.4 |

- **Duo ($\Psi$-samplers).** $\Psi$-samplers further improve performance beyond ReMDM. With the log-linear sampling schedule (as used by ReMDM), $\Psi$-samplers reduce the FID from 23.71 to 20.71. Using a cosine sampling schedule further improves the FID. Overall, Duo improves *from an FID of 25.63 (ancestral) to 15.05* with $\Psi$-samplers, and MDLM improves from *24.73 (ancestral) to 17.86* with $\Psi$-samplers.

### D.1.2 OPENWEBTEXT

We report the generative perplexity using GPT-2 Large, following standard practice (Sahoo et al., 2024; 2025a). Because language models can artificially lower the generative perplexity by producing repetitive text, we also report unigram entropy (Dieleman et al., 2022), as a proxy.

Some $\Psi$-samplers schedules reduce the unigram entropy more than others. Therefore, for figures, we select the $\kappa$ schedule whose unigram entropy matches (or is closest to) the entropy of samples generated with ancestral sampling. If multiple schedules achieve the same entropy, we choose the one with the lowest generative perplexity. We indicate which schedule is used for plots by highlighting the corresponding row in blue in the tables. Overall, the $\Psi$-samplers can reduce the Gen. PPL of *all* models while retaining the unigram entropy. Best results are achieved using the rescale schedule with $\eta \in \{0.01, 0.02\}$, for both MDLM and Duo.

Table 12 shows the generative perplexity of MDLM and Duo after pre-training and after distillation with SDTT (Deschenaux & Gulcehre, 2025) or DCD (Sahoo et al., 2025a) respectively, with and without nucleus sampling, using ancestral sampling. Table 13 shows the results when sampling with $\Psi$-samplers that are equivalent to ReMDM (Wang et al., 2025), with the *non-distilled* models, while Table 14 shows the result for the distilled models.

### D.2 DISTRIBUTION OF THE TOP $k$ ENTRIES OF THE SOFTMAX

To verify that our sparse implementation accurately approximates the curriculum weights of Sahoo et al. (2025a), we compare the empirical distributions of the top-$k$ largest entries between the original and our efficient implementation. While matching marginal distributions does not guarantee matching joint distributions, matching marginals are necessary for matching joints, and are easier to visualize. Recall that experimentally, our efficient implementation is sufficient to achieve strong performance (Sec. 5.2). Specifically, we show histograms using a tokenizer with $100k$ tokens in Figures 7, 8, 9, 10, and with the `GPT-2` tokenizer in Figures 11, 12, 13, 14, with varying temperature and log signal-to-noise ratios. In all cases, the top $k$ variables have matching distributions.

### D.3 TRAINING EFFICIENCY OF OUR FAST CURRICULUM

As shown in Table 4, our sparse curriculum achieves a 33% reduction in peak memory usage and reaches an average throughput 25% higher than Duo, at a context length of 1024.

Table 5: Zero-shot perplexity (PPL) on seven datasets. Lower is better. [†]Results taken from Sahoo et al. (2025a). Duo$^{++}$ ($k = 2$) achieves a slightly lower zero-shot perplexity than Duo on 6 of 7 datasets.

|  | PTB | Wiki | LM1B | LBD | AG News | PubMed | ArXiv |
|---|---|---|---|---|---|---|---|
| *Autoregressive* | | | | | | | |
| Transformer[†] | 82.05 | 25.75 | 51.25 | 51.28 | 52.09 | 49.01 | 41.73 |
| *Diffusion (138M)* | | | | | | | |
| SEDD Uniform[†] | 105.51 | 41.10 | 82.62 | 57.29 | 82.64 | 55.89 | 50.86 |
| UDLM[†] | 112.82 | 39.42 | 77.59 | 53.57 | 80.96 | 50.98 | 44.08 |
| Duo[†] | **89.35** | **33.57** | 73.86 | 49.78 | 67.81 | 44.48 | 40.39 |
| Duo$^{++}$ ($k = 2$) | 94.96 | 34.05 | **73.80** | **48.67** | 67.14 | **43.98** | **38.93** |
| Duo$^{++}$ ($k = 3$) | 91.94 | 34.65 | 74.16 | 49.89 | **66.89** | 44.87 | 40.42 |
| Duo$^{++}$ ($k = 5$) | 94.46 | 34.52 | 74.91 | 50.93 | 68.72 | 46.79 | 41.04 |

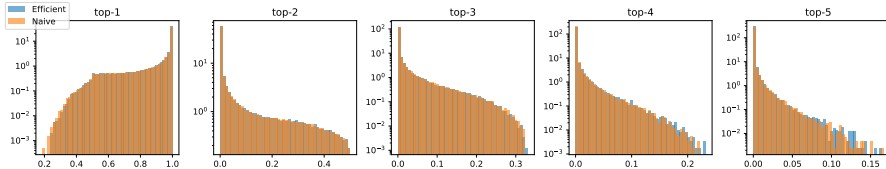

Figure 7: Marginal distributions of the top-5 entries using a tokenizer with 100k tokens, inverse temperature 100, and log signal-to-noise ratio −2. The histograms of the efficient and naive implementation match closely.

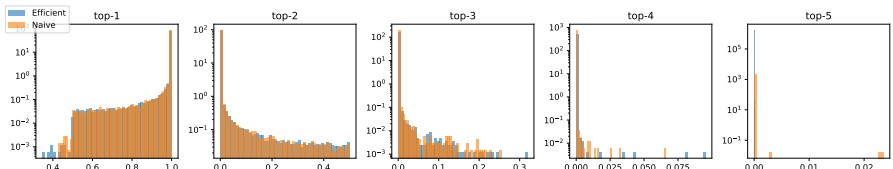

Figure 8: Marginal distributions of the top-5 entries using a tokenizer with 100k tokens, inverse temperature 1000, and log signal-to-noise ratio −1. The histograms of the efficient and naive implementation match closely.

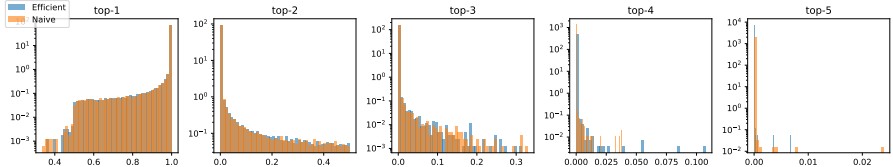

Figure 9: Marginal distributions of the top-5 entries using a tokenizer with 100k tokens, inverse temperature 1000, and log signal-to-noise ratio −2. The histograms of the efficient and naive implementation match closely.

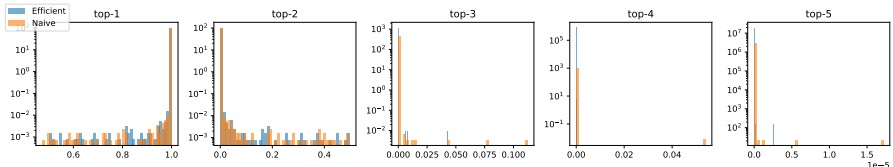

Figure 10: Marginal distributions of the top-5 entries using a tokenizer with 100k tokens, inverse temperature 1000, and log signal-to-noise ratio −4. The histograms of the efficient and naive implementation match closely.

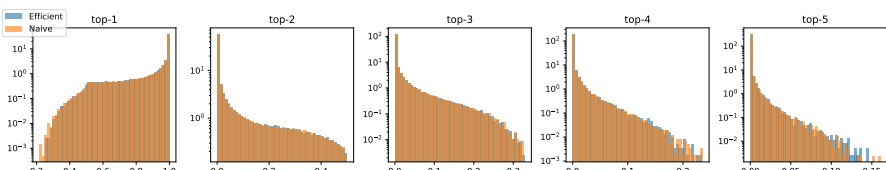

Figure 11: Marginal distributions of the top-5 entries using the GPT-2 tokenizer, inverse temperature 100, and log signal-to-noise ratio −2. The histograms of the efficient and naive implementation match closely.

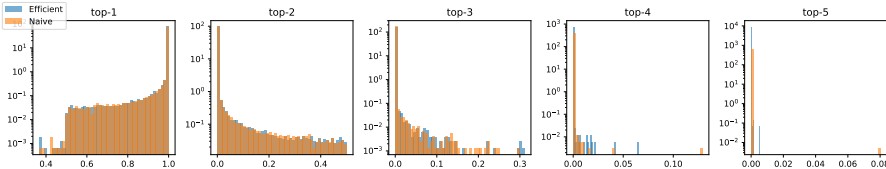

Figure 12: Marginal distributions of the top-5 entries using the GPT-2 tokenizer, inverse temperature 1000, and log signal-to-noise ratio −1. The histograms of the efficient and naive implementation match closely.

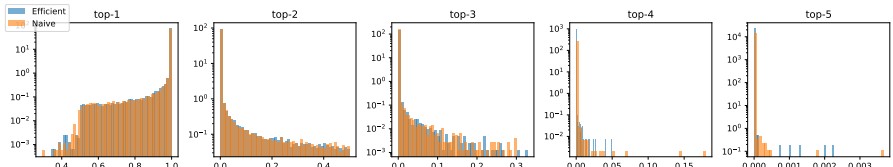

Figure 13: Marginal distributions of the top-5 entries using the GPT-2 tokenizer, inverse temperature 1000, and log signal-to-noise ratio −2. The histograms of the efficient and naive implementation match closely.

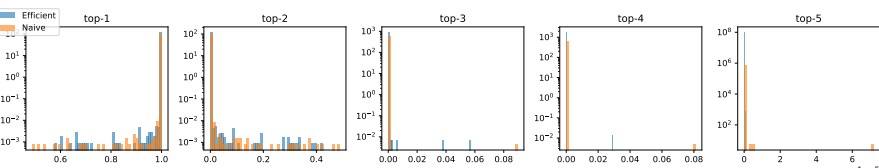

Figure 14: Marginal distributions of the top-5 entries using the GPT-2 tokenizer, inverse temperature 1000, and log signal-to-noise ratio −4. The histograms of the efficient and naive implementation match closely.

Table 6: FID on CIFAR-10 with ancestral sampling. We train and sample with the log-linear and cosine scheduler. MDLM performs best with time-conditioning while Duo does not. We sample with discrete classifier-free guidance ([Schiff et al., 2025](#)) with strength 1, and greedy predictions on the last step.

| Scheduler | Time | PPL ↓ | FID ↓ (Cosine) | | | | FID ↓ (Log-linear) | | | |
|---|---|---|---|---|---|---|---|---|---|---|
| | | | 64 | 256 | 1024 | 2048 | 64 | 256 | 1024 | 2048 |
| *MDLM* | | | | | | | | | | |
| Cosine | ✗ | 8.86 | 42.60 | 27.71 | 24.90 | 24.56 | **107.62** | 40.81 | 27.65 | 25.73 |
| Cosine | ✓ | **8.72** | **41.89** | **27.03** | **24.67** | **24.24** | 114.56 | **40.60** | **27.08** | **25.50** |
| Log-linear | ✗ | 8.76 | 43.95 | 29.01 | 26.11 | 25.67 | 111.77 | 42.15 | 28.85 | 26.89 |
| Log-linear | ✓ | 8.75 | 49.36 | 32.10 | 28.76 | 28.21 | 122.70 | 41.79 | 27.89 | 26.02 |
| *MDLM (nucleus p=0.9)* | | | | | | | | | | |
| Cosine | ✓ | **8.72** | 34.81 | 44.04 | 47.84 | 48.37 | 41.73 | 33.33 | 43.12 | 45.98 |
| *MDLM (no greedy)* | | | | | | | | | | |
| Cosine | ✓ | **8.72** | 42.14 | 27.19 | 24.47 | 24.46 | 114.55 | 40.92 | 27.13 | 25.60 |
| *Duo* | | | | | | | | | | |
| Cosine | ✗ | **10.27** | 32.37 | 27.28 | 26.38 | 26.02 | 33.93 | 27.93 | 26.51 | 26.03 |
| Cosine | ✓ | 10.32 | 33.74 | 27.98 | 26.81 | 26.96 | 36.23 | 28.77 | 27.08 | 26.79 |
| Log-linear | ✗ | 10.49 | **31.78** | **27.03** | **26.00** | **25.75** | **33.44** | **27.46** | **26.08** | **25.87** |
| Log-linear | ✓ | 10.45 | 34.05 | 27.74 | 26.58 | 26.37 | 36.46 | 28.49 | 26.60 | 26.22 |
| *Duo (nucleus p=0.9)* | | | | | | | | | | |
| Log-linear | ✗ | 10.49 | 23.13 | 22.21 | 22.58 | 22.49 | 24.24 | 22.41 | 22.35 | 22.54 |
| *Duo (no greedy)* | | | | | | | | | | |
| Log-linear | ✗ | 10.49 | 33.03 | 27.43 | 26.16 | 25.96 | 34.81 | 27.76 | 26.30 | 26.06 |

Table 7: FID on CIFAR-10 with ancestral sampling and a finer grid. We pick the variant with the best FID from Table 6.

| Algo | Train | Sample | p | FID ↓ | | | | | | | |
|---|---|---|---|---|---|---|---|---|---|---|---|
| | | | | 32 | 64 | 128 | 256 | 512 | 1024 | 2048 | 4096 |
| Duo | log-lin | log-lin | 1.0 | 42.71 | 33.44 | 29.18 | 27.46 | 26.62 | 26.08 | 25.87 | 25.79 |
| Duo | log-lin | log-lin | 0.9 | 28.53 | 24.24 | 22.89 | 22.41 | 22.56 | 22.35 | 22.54 | 22.41 |
| Duo | log-lin | cos | 1.0 | 39.65 | 31.78 | 28.55 | 27.03 | 26.03 | 25.89 | 25.75 | 25.63 |
| Duo | log-lin | cos | 0.9 | 25.96 | 23.13 | 22.68 | 22.21 | 22.26 | 22.58 | 22.49 | 22.49 |
| MDLM | cos | log-lin | 1.0 | 212.95 | 114.56 | 62.86 | 40.60 | 31.05 | 27.08 | 25.50 | 24.73 |
| MDLM | cos | log-lin | 0.9 | 84.85 | 41.73 | 31.28 | 33.33 | 38.49 | 43.12 | 45.98 | 55.37 |
| MDLM | cos | cos | 1.0 | 73.82 | 41.89 | 36.21 | 27.03 | 25.63 | 24.67 | 24.24 | 23.93 |
| MDLM | cos | cos | 0.9 | 58.31 | 34.81 | 37.91 | 44.04 | 45.32 | 47.84 | 48.37 | 49.23 |

Table 8: FID on CIFAR-10 with ReMDM (best checkpoints, as shown in Table 7). We sample with/without nucleus sampling, and with the 3 schedules of Wang et al. (2025) (cap, loop, rescale). For the loop schedule, we use $t_{\text{on}} = 0.55$, $t_{\text{off}} = 0.05$, $\alpha_{\text{on}} = 0.9$, following ReMDM. Sampling experiments are executed in the original codebase of Wang et al. (2025).

| | \multicolumn{8}{c}{Number of steps} | | | | | | | |
| --- | --- | --- | --- | --- | --- | --- | --- | --- |
| | 32 | 64 | 128 | 256 | 512 | 1024 | 2048 | 4096 |
| *ReMDM cap (p=1.0)* | | | | | | | | |
| $\eta = 0.005$ | **215.67** | **116.24** | **63.37** | **40.82** | **31.40** | **27.28** | **24.97** | **24.78** |
| $\eta = 0.010$ | 218.41 | 118.25 | 64.50 | 41.77 | 32.40 | 28.68 | 27.91 | 33.68 |
| $\eta = 0.020$ | 224.20 | 122.61 | 66.95 | 44.54 | 36.26 | 35.39 | 46.01 | 92.48 |
| $\eta = 0.050$ | 242.25 | 143.21 | 84.41 | 64.10 | 73.89 | 132.13 | 210.60 | 203.14 |
| *ReMDM loop (p=1.0)* | | | | | | | | |
| $\eta = 0.01$ | **307.56** | **234.55** | **138.56** | **80.50** | **55.86** | **47.05** | **45.44** | **50.44** |
| $\eta = 0.02$ | 307.81 | 237.28 | 142.21 | 83.68 | 59.96 | 53.88 | 60.50 | 87.54 |
| $\eta = 0.04$ | 308.24 | 242.70 | 152.28 | 94.63 | 76.93 | 88.53 | 135.05 | 196.58 |
| $\eta = 0.06$ | 308.88 | 248.76 | 165.79 | 114.92 | 113.26 | 157.92 | 223.70 | 237.16 |
| *ReMDM rescale (p=1.0)* | | | | | | | | |
| $\eta = 0.01$ | **216.92** | **116.73** | **63.56** | **40.65** | **30.86** | **26.03** | **23.77** | **23.71** |
| $\eta = 0.02$ | 221.21 | 119.79 | 65.08 | 42.02 | 32.29 | 28.11 | 28.66 | 39.39 |
| $\eta = 0.04$ | 229.72 | 127.94 | 70.89 | 46.98 | 38.74 | 41.23 | 67.44 | 130.05 |
| $\eta = 0.05$ | 234.35 | 133.08 | 75.02 | 50.92 | 45.01 | 57.03 | 107.13 | 164.44 |
| *ReMDM cap (p=0.9)* | | | | | | | | |
| $\eta = 0.005$ | 88.08 | 40.02 | **27.31** | **29.43** | 36.50 | 45.10 | 57.08 | 73.40 |
| $\eta = 0.010$ | 87.68 | 39.55 | 27.35 | 31.24 | 41.22 | 54.55 | 71.65 | 93.06 |
| $\eta = 0.020$ | 85.95 | 38.46 | 27.80 | 35.01 | 50.50 | 69.60 | 91.49 | 118.87 |
| $\eta = 0.050$ | **81.91** | **35.56** | 29.39 | 46.90 | 70.24 | 95.24 | 125.60 | 163.32 |
| *ReMDM loop (p=0.9)* | | | | | | | | |
| $\eta = 0.01$ | 209.24 | 100.01 | 47.27 | 29.44 | **27.55** | **30.50** | **34.21** | **37.56** |
| $\eta = 0.02$ | 208.36 | 99.29 | 47.12 | 29.38 | 27.74 | 31.17 | 35.42 | 39.52 |
| $\eta = 0.04$ | 206.51 | 98.18 | 46.87 | 29.28 | 28.09 | 32.12 | 37.19 | 42.45 |
| $\eta = 0.06$ | **204.83** | **97.24** | **46.72** | **29.19** | 28.30 | 32.77 | 38.47 | 44.64 |
| *ReMDM rescale (p=0.9)* | | | | | | | | |
| $\eta = 0.01$ | 87.31 | 39.51 | **27.25** | **30.74** | **40.22** | **53.30** | **70.24** | **91.79** |
| $\eta = 0.02$ | 85.94 | 38.45 | 27.45 | 34.13 | 49.00 | 67.89 | 90.61 | 118.10 |
| $\eta = 0.04$ | 83.47 | 36.44 | 28.29 | 41.76 | 63.40 | 87.03 | 115.60 | 153.03 |
| $\eta = 0.05$ | **82.26** | **35.69** | 28.99 | 44.69 | 68.80 | 94.07 | 125.42 | 165.62 |
| *ReMDM* | | | | | | | | |
| Best ($p = 1.0$) | 215.67 | 116.24 | 63.37 | 40.65 | 30.86 | **26.03** | **23.77** | **23.71** |
| Best ($p = 0.9$) | **81.91** | **81.91** | **27.25** | **29.19** | **27.55** | 30.50 | 34.21 | 37.56 |
| *MDLM* | | | | | | | | |
| Ancestral (p=1.0) | 212.95 | 114.56 | 62.86 | 40.60 | **31.05** | **27.08** | **25.50** | **24.73** |
| Ancestral (p=0.9) | **84.85** | **41.73** | **31.28** | **33.33** | 38.49 | 43.12 | 45.98 | 55.37 |

Table 9: FID on CIFAR-10 with $\Psi$-samplers, where $\Psi$-samplers are activated for steps with $t \in [t_{\text{off}}, t_{\text{on}}]$, when $\kappa_t$ is kept constant (according to the $\kappa$ column, 1 otherwise). We use the same checkpoints as in Table 7. Using a cosine sampling schedule and light noise injection ($\kappa$ close to 1) generally perform best. The CIFAR-10 curves in Fig. 1 show the best FID per number of steps.

| Algo | $\kappa$ | Train | Sample | $t_{\text{on}}$ | $t_{\text{off}}$ | FID ↓ | | | | | | | |
|------|------|-------|--------|------|------|------|------|------|------|------|------|------|------|
| | | | | | | 32 | 64 | 128 | 256 | 512 | 1024 | 2048 | 4096 |
| Duo | 0.02 | log-lin | cos | 0.2 | 0.15 | 40.64 | 33.06 | 30.36 | 29.85 | 31.31 | 34.36 | 39.06 | 38.38 |
| Duo | 0.02 | log-lin | cos | 0.5 | 0.45 | 41.81 | 33.67 | 29.50 | 26.55 | 24.83 | 25.12 | 31.63 | 51.83 |
| Duo | 0.02 | log-lin | cos | 0.8 | 0.7 | 43.99 | 37.41 | 35.68 | 38.88 | 46.76 | 59.68 | 75.46 | 91.73 |
| Duo | 0.5 | log-lin | cos | 0.2 | 0.1 | 39.95 | 32.14 | 28.86 | 27.18 | 26.57 | 26.46 | 27.29 | 28.35 |
| Duo | 0.5 | log-lin | cos | 0.6 | 0.4 | 39.54 | **29.40** | 23.46 | 20.77 | 23.72 | 38.42 | 72.97 | 105.75 |
| Duo | 0.5 | log-lin | cos | 0.9 | 0.65 | 43.00 | 34.68 | 31.85 | 34.73 | 45.68 | 64.97 | 88.07 | 107.36 |
| Duo | 0.95 | log-lin | cos | 0.5 | 0.1 | 39.30 | 30.58 | 26.15 | 23.46 | 20.93 | 18.48 | 16.38 | **15.05** |
| Duo | 0.95 | log-lin | cos | 0.6 | 0.1 | 39.19 | 30.15 | 25.14 | 21.54 | 18.64 | **16.70** | **16.30** | 18.99 |
| Duo | 0.95 | log-lin | cos | 0.9 | 0.3 | **39.04** | 29.88 | 24.72 | 20.90 | 19.20 | 21.09 | 30.00 | 51.43 |
| Duo | 0.95 | log-lin | cos | 0.9 | 0.4 | 39.21 | 30.29 | 25.26 | 21.57 | 19.92 | 21.50 | 30.03 | 50.88 |
| Duo | 0.98 | log-lin | cos | 1.0 | 0.05 | 39.31 | 30.97 | 26.39 | 23.13 | 20.56 | 18.80 | 19.46 | 25.83 |
| Duo | 0.98 | log-lin | cos | 1.0 | 0.1 | 39.31 | 30.99 | 26.40 | 23.14 | 20.58 | 18.83 | 19.48 | 25.82 |
| Duo | 0.99 | log-lin | cos | 1.0 | 0.05 | 39.34 | 31.56 | 27.46 | 24.73 | 22.35 | 20.07 | 18.50 | 19.39 |
| Duo | 0.99 | log-lin | cos | 1.0 | 0.1 | 39.35 | 31.57 | 27.46 | 24.73 | 22.37 | 20.09 | 18.51 | 19.41 |
| Duo | 0.02 | log-lin | log-lin | 0.2 | 0.15 | 42.25 | 33.71 | 29.84 | 27.95 | 27.64 | 27.56 | 29.35 | 31.02 |
| Duo | 0.02 | log-lin | log-lin | 0.5 | 0.45 | 43.86 | 36.29 | 33.35 | 33.24 | 34.74 | 36.97 | 36.77 | 37.30 |
| Duo | 0.02 | log-lin | log-lin | 0.8 | 0.7 | 43.95 | 33.75 | 28.32 | 27.78 | 37.12 | 69.66 | 113.05 | 132.86 |
| Duo | 0.5 | log-lin | log-lin | 0.2 | 0.1 | 42.10 | 33.40 | 29.19 | 27.14 | 26.22 | 25.52 | 25.10 | 24.71 |
| Duo | 0.5 | log-lin | log-lin | 0.6 | 0.4 | 42.44 | 33.68 | 29.15 | 25.93 | 24.16 | 22.44 | 21.00 | 27.97 |
| Duo | 0.5 | log-lin | log-lin | 0.9 | 0.65 | 42.87 | 31.04 | 26.37 | 31.86 | 61.36 | 121.64 | 155.77 | 151.48 |
| Duo | 0.95 | log-lin | log-lin | 0.5 | 0.1 | 41.74 | 32.97 | 28.57 | 26.05 | 24.62 | 23.13 | 21.81 | 20.16 |
| Duo | 0.95 | log-lin | log-lin | 0.6 | 0.1 | 41.46 | 32.47 | 27.74 | 24.97 | 22.94 | 20.83 | 18.87 | 16.82 |
| Duo | 0.95 | log-lin | log-lin | 0.9 | 0.3 | 41.10 | 30.55 | 24.54 | **20.50** | **17.97** | 18.04 | 22.14 | 35.43 |
| Duo | 0.95 | log-lin | log-lin | 0.9 | 0.4 | 41.18 | 30.58 | 24.71 | 20.59 | 18.08 | 18.02 | 22.07 | 35.44 |
| Duo | 0.98 | log-lin | log-lin | 1.0 | 0.05 | 41.80 | 31.96 | 26.83 | 23.17 | 20.10 | 18.12 | 18.38 | 22.89 |
| Duo | 0.98 | log-lin | log-lin | 1.0 | 0.1 | 41.81 | 31.98 | 26.85 | 23.17 | 20.12 | 18.15 | 18.40 | 22.94 |
| Duo | 0.99 | log-lin | log-lin | 1.0 | 0.05 | 41.99 | 32.63 | 27.74 | 24.67 | 22.13 | 19.72 | 17.93 | 18.25 |
| Duo | 0.99 | log-lin | log-lin | 1.0 | 0.1 | 41.99 | 32.63 | 27.75 | 24.67 | 22.13 | 19.75 | 17.95 | 18.28 |
| MDLM | 0.02 | cos | cos | 0.2 | 0.15 | 75.63 | 49.18 | 45.02 | 54.67 | 83.47 | 181.18 | 280.42 | 297.52 |
| MDLM | 0.02 | cos | cos | 0.5 | 0.45 | 117.57 | 89.53 | 111.75 | 200.49 | 283.55 | 310.51 | 314.98 | 313.93 |
| MDLM | 0.02 | cos | cos | 0.8 | 0.7 | 172.24 | 197.61 | 232.36 | 262.87 | 269.22 | 267.86 | 264.57 | 259.88 |
| MDLM | 0.5 | cos | cos | 0.2 | 0.1 | 73.13 | 46.10 | 38.47 | 39.71 | 48.49 | 75.27 | 173.09 | 266.36 |
| MDLM | 0.5 | cos | cos | 0.6 | 0.4 | 134.11 | 114.88 | 144.25 | 217.74 | 268.03 | 274.83 | 270.53 | 256.03 |
| MDLM | 0.5 | cos | cos | 0.9 | 0.65 | 151.90 | 131.04 | 147.67 | 177.75 | 198.33 | 201.97 | 193.77 | 184.76 |
| MDLM | 0.95 | cos | cos | 0.5 | 0.1 | 73.03 | 44.15 | 33.68 | 30.50 | 29.93 | 31.50 | 35.72 | 51.53 |
| MDLM | 0.95 | cos | cos | 0.6 | 0.1 | 74.57 | 45.00 | 34.07 | 30.32 | 29.16 | 31.03 | 37.46 | 64.74 |
| MDLM | 0.95 | cos | cos | 0.9 | 0.3 | 79.25 | 47.02 | 33.97 | 27.84 | 24.24 | 23.43 | 26.96 | 42.58 |
| MDLM | 0.95 | cos | cos | 0.9 | 0.4 | 78.18 | 46.36 | 33.06 | 26.69 | **22.67** | 20.91 | 21.90 | 28.82 |
| MDLM | 0.98 | cos | cos | 1.0 | 0.05 | 74.05 | 43.85 | 32.32 | 26.69 | 23.22 | 20.81 | 19.41 | 20.20 |
| MDLM | 0.98 | cos | cos | 1.0 | 0.1 | 74.05 | 43.85 | 32.31 | 26.65 | 23.17 | **20.76** | 19.26 | 19.98 |
| MDLM | 0.99 | cos | cos | 1.0 | 0.05 | 72.39 | **42.87** | 31.79 | 26.65 | 23.72 | 21.07 | 19.24 | 17.94 |
| MDLM | 0.99 | cos | cos | 1.0 | 0.1 | **72.38** | **42.87** | **31.78** | **26.64** | 23.69 | 21.04 | **19.19** | **17.86** |
| MDLM | 0.02 | cos | log-lin | 0.2 | 0.15 | 217.56 | 118.08 | 68.02 | 51.76 | 55.02 | 78.21 | 171.72 | 275.25 |
| MDLM | 0.02 | cos | log-lin | 0.5 | 0.45 | 247.31 | 157.61 | 124.97 | 162.92 | 256.01 | 298.74 | 305.05 | 310.28 |
| MDLM | 0.02 | cos | log-lin | 0.8 | 0.7 | 298.96 | 294.71 | 298.95 | 312.49 | 317.03 | 312.60 | 308.42 | 302.37 |
| MDLM | 0.5 | cos | log-lin | 0.2 | 0.1 | 216.08 | 116.99 | 65.73 | 45.72 | 41.32 | 45.95 | 68.60 | 152.77 |
| MDLM | 0.5 | cos | log-lin | 0.6 | 0.4 | 266.16 | 195.76 | 171.73 | 212.68 | 273.48 | 281.96 | 272.45 | 260.26 |
| MDLM | 0.5 | cos | log-lin | 0.9 | 0.65 | 296.08 | 268.98 | 265.73 | 278.38 | 281.68 | 275.20 | 265.49 | 247.21 |
| MDLM | 0.95 | cos | log-lin | 0.5 | 0.1 | 216.90 | 117.05 | 64.76 | 43.50 | 36.06 | 34.84 | 37.06 | 44.92 |
| MDLM | 0.95 | cos | log-lin | 0.6 | 0.1 | 218.58 | 118.21 | 65.33 | 44.32 | 37.14 | 36.09 | 39.42 | 55.34 |
| MDLM | 0.95 | cos | log-lin | 0.9 | 0.3 | 225.19 | 124.03 | 67.82 | 44.06 | 35.20 | 33.97 | 42.48 | 80.34 |
| MDLM | 0.95 | cos | log-lin | 0.9 | 0.4 | 223.84 | 123.04 | 67.19 | 43.29 | 33.85 | 32.00 | 37.23 | 63.89 |
| MDLM | 0.98 | cos | log-lin | 1.0 | 0.05 | 218.15 | 118.08 | 63.97 | 40.97 | 30.67 | 25.69 | 23.64 | 25.40 |
| MDLM | 0.98 | cos | log-lin | 1.0 | 0.1 | 218.14 | 118.09 | 63.96 | 40.96 | 30.65 | 25.64 | 23.57 | 25.29 |
| MDLM | 0.99 | cos | log-lin | 1.0 | 0.05 | 215.41 | 116.02 | 63.30 | 40.42 | 30.43 | 25.37 | 22.45 | 20.77 |
| MDLM | 0.99 | cos | log-lin | 1.0 | 0.1 | 215.40 | 116.03 | 63.27 | 40.41 | 30.43 | 25.35 | 22.42 | 20.71 |

Table 10: Inception Score on CIFAR-10 with $\Psi$-samplers, where $\Psi$-samplers are activated for steps with $t \in [t_{\text{off}}, t_{\text{on}}]$, when $\kappa_t$ is kept constant (according to the $\kappa$ column, 1 otherwise). We use the same checkpoints as in Table 7. The CIFAR-10 curves in Fig. 6 show the best Inception Score per number of steps.

| Algo | $\kappa$ | Train | Sample | $t_{\text{on}}$ | $t_{\text{off}}$ | Inception Score ↑ | | | | | | | |
|------|------|-------|--------|------|------|------|------|------|------|------|------|------|------|
| | | | | | | 32 | 64 | 128 | 256 | 512 | 1024 | 2048 | 4096 |
| Duo | 0.02 | log-lin | cos | 0.2 | 0.15 | 7.02 | 7.25 | 7.35 | 7.48 | 7.52 | 7.47 | 7.38 | 7.63 |
| Duo | 0.02 | log-lin | cos | 0.5 | 0.45 | 7.09 | 7.44 | 7.64 | 8.04 | 8.32 | **8.59** | 8.57 | 7.94 |
| Duo | 0.02 | log-lin | cos | 0.8 | 0.7 | 6.84 | 6.99 | 7.00 | 6.91 | 6.64 | 6.16 | 5.67 | 5.19 |
| Duo | 0.5 | log-lin | cos | 0.2 | 0.1 | 6.96 | 7.21 | 7.28 | 7.39 | 7.45 | 7.48 | 7.56 | 7.73 |
| Duo | 0.5 | log-lin | cos | 0.6 | 0.4 | **7.31** | **7.73** | **8.14** | **8.51** | **8.46** | 7.91 | 6.40 | 5.39 |
| Duo | 0.5 | log-lin | cos | 0.9 | 0.65 | 6.87 | 7.10 | 7.22 | 7.11 | 6.72 | 5.97 | 5.23 | 4.67 |
| Duo | 0.95 | log-lin | cos | 0.5 | 0.1 | 6.98 | 7.26 | 7.45 | 7.53 | 7.67 | 7.89 | 8.06 | 8.29 |
| Duo | 0.95 | log-lin | cos | 0.6 | 0.1 | 7.00 | 7.31 | 7.45 | 7.70 | 7.91 | 8.17 | 8.34 | 8.46 |
| Duo | 0.95 | log-lin | cos | 0.9 | 0.3 | 7.08 | 7.37 | 7.54 | 7.84 | 8.01 | 8.07 | 7.72 | 6.84 |
| Duo | 0.95 | log-lin | cos | 0.9 | 0.4 | 7.04 | 7.31 | 7.50 | 7.78 | 7.92 | 8.08 | 7.78 | 6.89 |
| Duo | 0.98 | log-lin | cos | 1.0 | 0.05 | 7.00 | 7.25 | 7.40 | 7.55 | 7.73 | 7.97 | 8.10 | 7.91 |
| Duo | 0.98 | log-lin | cos | 1.0 | 0.1 | 6.99 | 7.25 | 7.40 | 7.55 | 7.74 | 7.97 | 8.09 | 7.91 |
| Duo | 0.99 | log-lin | cos | 1.0 | 0.05 | 6.98 | 7.22 | 7.37 | 7.45 | 7.58 | 7.77 | 7.96 | 8.08 |
| Duo | 0.99 | log-lin | cos | 1.0 | 0.1 | 6.98 | 7.22 | 7.37 | 7.46 | 7.58 | 7.77 | 7.96 | 8.10 |
| Duo | 0.02 | log-lin | log-lin | 0.2 | 0.15 | 6.82 | 7.09 | 7.22 | 7.30 | 7.36 | 7.44 | 7.46 | 7.43 |
| Duo | 0.02 | log-lin | log-lin | 0.5 | 0.45 | 6.95 | 7.28 | 7.45 | 7.64 | 7.67 | 7.70 | 8.06 | 8.68 |
| Duo | 0.02 | log-lin | log-lin | 0.8 | 0.7 | 7.00 | 7.54 | 8.02 | 8.18 | 7.89 | 6.46 | 5.03 | 4.55 |
| Duo | 0.5 | log-lin | log-lin | 0.2 | 0.1 | 6.81 | 7.04 | 7.20 | 7.26 | 7.29 | 7.36 | 7.47 | 7.50 |
| Duo | 0.5 | log-lin | log-lin | 0.6 | 0.4 | 7.04 | 7.45 | 7.73 | 7.93 | 8.20 | 8.51 | **9.00** | **9.50** |
| Duo | 0.5 | log-lin | log-lin | 0.9 | 0.65 | 7.05 | 7.61 | 7.97 | 7.74 | 6.45 | 4.46 | 3.77 | 4.07 |
| Duo | 0.95 | log-lin | log-lin | 0.5 | 0.1 | 6.80 | 7.10 | 7.25 | 7.31 | 7.35 | 7.43 | 7.55 | 7.63 |
| Duo | 0.95 | log-lin | log-lin | 0.6 | 0.1 | 6.85 | 7.12 | 7.28 | 7.40 | 7.46 | 7.66 | 7.81 | 7.97 |
| Duo | 0.95 | log-lin | log-lin | 0.9 | 0.3 | 6.89 | 7.27 | 7.58 | 7.78 | 8.10 | 8.22 | 8.20 | 7.67 |
| Duo | 0.95 | log-lin | log-lin | 0.9 | 0.4 | 6.89 | 7.25 | 7.58 | 7.80 | 8.05 | 8.25 | 8.26 | 7.69 |
| Duo | 0.98 | log-lin | log-lin | 1.0 | 0.05 | 6.85 | 7.19 | 7.36 | 7.49 | 7.72 | 7.96 | 8.05 | 8.03 |
| Duo | 0.98 | log-lin | log-lin | 1.0 | 0.1 | 6.85 | 7.20 | 7.38 | 7.49 | 7.72 | 7.96 | 8.04 | 8.02 |
| Duo | 0.99 | log-lin | log-lin | 1.0 | 0.05 | 6.81 | 7.13 | 7.32 | 7.45 | 7.61 | 7.71 | 7.98 | 8.12 |
| Duo | 0.99 | log-lin | log-lin | 1.0 | 0.1 | 6.81 | 7.14 | 7.32 | 7.45 | 7.62 | 7.70 | 7.99 | 8.13 |
| MDLM | 0.02 | cos | cos | 0.2 | 0.15 | 5.56 | 6.61 | 6.90 | 6.75 | 5.52 | 2.68 | 1.57 | 1.56 |
| MDLM | 0.02 | cos | cos | 0.5 | 0.45 | 4.22 | 5.11 | 4.36 | 2.44 | 1.61 | 1.41 | 1.45 | 1.56 |
| MDLM | 0.02 | cos | cos | 0.8 | 0.7 | 3.12 | 2.82 | 2.41 | 2.03 | 1.96 | 1.97 | 2.02 | 2.09 |
| MDLM | 0.5 | cos | cos | 0.2 | 0.1 | 5.63 | 6.63 | 7.00 | 7.09 | 6.90 | 5.68 | 2.78 | 1.73 |
| MDLM | 0.5 | cos | cos | 0.6 | 0.4 | 3.83 | 4.32 | 3.55 | 2.35 | 1.85 | 2.14 | 2.51 | 2.99 |
| MDLM | 0.5 | cos | cos | 0.9 | 0.65 | 3.62 | 4.18 | 3.95 | 3.47 | 3.18 | 3.15 | 3.37 | 3.75 |
| MDLM | 0.95 | cos | cos | 0.5 | 0.1 | **5.66** | 6.68 | 7.13 | 7.29 | 7.44 | 7.41 | 7.44 | 6.77 |
| MDLM | 0.95 | cos | cos | 0.6 | 0.1 | 5.59 | 6.70 | 7.21 | 7.41 | 7.52 | 7.57 | 7.45 | 6.33 |
| MDLM | 0.95 | cos | cos | 0.9 | 0.3 | 5.43 | 6.68 | **7.25** | 7.63 | 7.90 | **8.15** | 8.18 | 7.58 |
| MDLM | 0.95 | cos | cos | 0.9 | 0.4 | 5.45 | 6.66 | **7.25** | **7.64** | **7.93** | 8.14 | **8.30** | 8.18 |
| MDLM | 0.98 | cos | cos | 1.0 | 0.05 | 5.57 | 6.71 | 7.22 | 7.45 | 7.71 | 7.93 | 8.14 | 8.30 |
| MDLM | 0.98 | cos | cos | 1.0 | 0.1 | 5.57 | 6.72 | 7.22 | 7.46 | 7.72 | 7.93 | 8.15 | **8.31** |
| MDLM | 0.99 | cos | cos | 1.0 | 0.05 | 5.60 | **6.73** | 7.18 | 7.39 | 7.53 | 7.81 | 7.97 | 8.12 |
| MDLM | 0.99 | cos | cos | 1.0 | 0.1 | 5.60 | **6.73** | 7.19 | 7.39 | 7.53 | 7.81 | 7.97 | 8.14 |
| MDLM | 0.02 | cos | log-lin | 0.2 | 0.15 | 2.63 | 4.59 | 5.86 | 6.46 | 6.45 | 5.59 | 2.78 | 1.67 |
| MDLM | 0.02 | cos | log-lin | 0.5 | 0.45 | 2.21 | 3.56 | 4.08 | 3.06 | 1.78 | 1.43 | 1.38 | 1.36 |
| MDLM | 0.02 | cos | log-lin | 0.8 | 0.7 | 1.65 | 1.63 | 1.55 | 1.43 | 1.42 | 1.60 | 1.80 | 1.96 |
| MDLM | 0.5 | cos | log-lin | 0.2 | 0.1 | 2.66 | 4.60 | 5.91 | 6.58 | 6.81 | 6.76 | 5.77 | 3.15 |
| MDLM | 0.5 | cos | log-lin | 0.6 | 0.4 | 1.97 | 2.78 | 2.99 | 2.27 | 1.62 | 1.58 | 1.92 | 2.33 |
| MDLM | 0.5 | cos | log-lin | 0.9 | 0.65 | 1.69 | 1.91 | 1.90 | 1.79 | 1.91 | 2.35 | 2.87 | 3.29 |
| MDLM | 0.95 | cos | log-lin | 0.5 | 0.1 | 2.65 | 4.60 | 5.95 | 6.64 | 6.93 | 7.03 | 7.07 | 6.78 |
| MDLM | 0.95 | cos | log-lin | 0.6 | 0.1 | 2.62 | 4.55 | 5.94 | 6.66 | 6.98 | 7.16 | 7.10 | 6.47 |
| MDLM | 0.95 | cos | log-lin | 0.9 | 0.3 | 2.51 | 4.45 | 5.93 | 6.84 | 7.35 | 7.61 | 7.31 | 5.83 |
| MDLM | 0.95 | cos | log-lin | 0.9 | 0.4 | 2.54 | 4.46 | 5.94 | 6.85 | 7.40 | 7.69 | 7.59 | 6.60 |
| MDLM | 0.98 | cos | log-lin | 1.0 | 0.05 | 2.62 | 4.56 | 6.04 | 6.85 | 7.31 | 7.66 | 7.79 | 8.01 |
| MDLM | 0.98 | cos | log-lin | 1.0 | 0.1 | 2.62 | 4.56 | 6.03 | 6.85 | 7.32 | 7.67 | 7.81 | 8.03 |
| MDLM | 0.99 | cos | log-lin | 1.0 | 0.05 | 2.68 | 4.64 | 6.02 | 6.84 | 7.21 | 7.47 | 7.70 | 7.93 |
| MDLM | 0.99 | cos | log-lin | 1.0 | 0.1 | 2.68 | 4.64 | 6.02 | 6.84 | 7.21 | 7.47 | 7.70 | 7.93 |

Table 11: FID on CIFAR-10 using $\Psi$-samplers whose $\kappa_t$ schedulers are equivalent to ReMDM. We use no nucleus sampling, no temperature scaling, and $\text{cfg} = 1$. As expected, with the log-linear scheduler, we reach a similar FID as when using the ReMDM codebase (Table 8). However, note that by using a log-linear scheduler, using a constant $\kappa_t = 0.99$, we reach a better FID than with the original ReMDM scheduler.

| Algo | Train | Sample | FID ↓ | | | | | | | |
|------|-------|--------|-------|----|-----|-----|-----|------|------|------|
| | | | 32 | 64 | 128 | 256 | 512 | 1024 | 2048 | 4096 |
| *Duo with the ReMDM rescale schedule* | | | | | | | | | | |
| Duo | log-lin | cos | **39.64** | **32.03** | **28.49** | **26.95** | **26.16** | **25.71** | 25.25 | **25.02** |
| Duo | log-lin | log-lin | 42.27 | 33.58 | 29.49 | 27.36 | 26.33 | 25.86 | **25.07** | 25.21 |
| *ReMDM Rescale ($\eta = 0.01$)* | | | | | | | | | | |
| MDLM | cos | cos | **70.64** | **41.94** | **31.60** | **27.31** | **25.27** | **24.61** | **23.41** | **23.25** |
| MDLM | cos | log-lin | 213.22 | 114.24 | 62.51 | 40.51 | 30.28 | 26.21 | 23.61 | 23.40 |
| *ReMDM Cap ($\eta = 0.005$)* | | | | | | | | | | |
| MDLM | cos | log-lin | 215.75 | 115.77 | 63.20 | 41.25 | 31.60 | 27.30 | 25.16 | 24.79 |
| *ReMDM Loop ($t_{on} = 0.55, t_{off} = 0.05, \alpha_{on} = 0.9, \eta = 0.01$)* | | | | | | | | | | |
| MDLM | cos | log-lin | 305.30 | 224.84 | 120.58 | 66.39 | 45.70 | 39.06 | 41.44 | 52.71 |

Table 12: Generative Perplexity (Gen. PPL) and Unigram Entropy on OpenWebText (Gokaslan & Cohen, 2019) with ancestral sampling (no nucleus, no temperature scaling). We train using the log-linear noise scheduler, and sampling with the cosine scheduler is slightly better. We stick to the log-linear schedule for sampling in further experiments, to follow prior work, and since the cosine schedule only marginally reduces the Gen. PPL.

| Algo | Dist. | p | Sched. | Gen. PPL (↓) | | | | | | | |
|------|-------|---|--------|------|------|------|------|------|------|------|------|
| | | | | 32 | 64 | 128 | 256 | 512 | 1024 | 2048 | 4096 |
| Duo | ✗ | 1.0 | cos | 87.23 (5.54) | 79.94 (5.55) | 75.87 (5.53) | 73.95 (5.54) | 72.13 (5.54) | 71.41 (5.53) | 72.29 (5.53) | 70.77 (5.52) |
| Duo | ✗ | 1.0 | log-lin | 96.76 (5.57) | 86.01 (5.56) | 79.97 (5.55) | 78.46 (5.53) | 76.93 (5.54) | 75.02 (5.53) | 75.65 (5.52) | 75.39 (5.52) |
| Duo | ✗ | 0.9 | cos | 42.42 (5.36) | 39.26 (5.37) | 37.62 (5.35) | 36.52 (5.35) | 35.21 (5.34) | 35.37 (5.34) | 35.39 (5.34) | 34.91 (5.33) |
| Duo | ✗ | 0.9 | log-lin | 44.24 (5.40) | 40.08 (5.40) | 37.93 (5.39) | 36.66 (5.37) | 35.77 (5.37) | 34.79 (5.35) | 34.93 (5.35) | 34.75 (5.35) |
| Duo | ✓ | 1.0 | cos | 67.04 (5.47) | 61.09 (5.45) | 59.65 (5.42) | 57.76 (5.42) | 57.90 (5.42) | 56.81 (5.43) | 56.39 (5.41) | 57.32 (5.42) |
| Duo | ✓ | 1.0 | log-lin | 68.35 (5.54) | 62.92 (5.54) | 59.82 (5.50) | 58.77 (5.46) | 58.32 (5.46) | 57.82 (5.45) | 55.39 (5.43) | 55.89 (5.42) |
| Duo | ✓ | 0.9 | cos | 34.20 (5.31) | 31.79 (5.29) | 31.09 (5.25) | 30.05 (5.25) | 29.82 (5.26) | 29.68 (5.27) | 29.52 (5.24) | 29.73 (5.23) |
| Duo | ✓ | 0.9 | log-lin | 35.92 (5.41) | 32.98 (5.40) | 31.49 (5.36) | 30.32 (5.31) | 30.06 (5.29) | 30.00 (5.28) | 28.90 (5.25) | 29.19 (5.25) |
| MDLM | ✗ | 1.0 | cos | 168.66 (5.68) | 131.55 (5.66) | 115.74 (5.64) | 111.72 (5.63) | 106.63 (5.63) | 104.56 (5.62) | 103.12 (5.62) | 104.73 (5.62) |
| MDLM | ✗ | 1.0 | log-lin | 194.09 (5.74) | 141.67 (5.69) | 120.95 (5.67) | 111.85 (5.65) | 107.89 (5.64) | 105.64 (5.64) | 105.40 (5.63) | 105.03 (5.62) |
| MDLM | ✗ | 0.9 | cos | 58.33 (5.39) | 46.71 (5.36) | 40.66 (5.32) | 39.43 (5.33) | 37.64 (5.32) | 37.39 (5.33) | 36.98 (5.31) | 36.87 (5.31) |
| MDLM | ✗ | 0.9 | log-lin | 70.34 (5.49) | 51.14 (5.43) | 43.60 (5.39) | 40.01 (5.37) | 39.02 (5.35) | 37.91 (5.34) | 37.59 (5.32) | 36.76 (5.31) |
| MDLM | ✓ | 1.0 | cos | 63.04 (5.45) | 52.72 (5.43) | 47.83 (5.41) | 45.94 (5.42) | 44.67 (5.41) | 44.60 (5.41) | 44.50 (5.41) | 44.42 (5.41) |
| MDLM | ✓ | 1.0 | log-lin | 68.61 (5.48) | 55.26 (5.45) | 49.51 (5.44) | 46.13 (5.42) | 45.61 (5.42) | 44.87 (5.42) | 44.53 (5.41) | 44.38 (5.42) |
| MDLM | ✓ | 0.9 | cos | 31.47 (5.21) | 26.52 (5.19) | 24.14 (5.18) | 23.49 (5.17) | 22.93 (5.17) | 22.64 (5.17) | 22.38 (5.16) | 22.49 (5.17) |
| MDLM | ✓ | 0.9 | log-lin | 34.85 (5.26) | 28.21 (5.23) | 25.27 (5.21) | 24.01 (5.19) | 23.25 (5.18) | 22.75 (5.17) | 22.73 (5.17) | 22.46 (5.16) |

Table 13: Generative Perplexity (Gen. PPL) and Unigram Entropy on OpenWebText (Gokaslan & Cohen, 2019) with $\Psi$-samplers using $\kappa_t$ schedules matching ReMDM (log-linear step size) and **non-distilled models** (as in Table 12). We experiment with nucleus sampling, following Wang et al. (2025). The rescale schedule is most effective to improve the Gen. PPL while retaining the unigram entropy. The lightblue rows are the ones plotted in Fig. 1 (left).

| Algo | Eta | Nucleus P | Gen. PPL (↓) | | | | | | | |
|---|---|---|---|---|---|---|---|---|---|---|
| | | | 32 | 64 | 128 | 256 | 512 | 1024 | 2048 | 4096 |
| *Ancestral Sampling* | | | | | | | | | | |
| Duo | N.A | 1.0 | 96.76 (5.57) | 86.01 (5.56) | 79.97 (5.55) | 78.46 (5.53) | 76.93 (5.54) | 75.02 (5.53) | 75.65 (5.52) | 75.39 (5.52) |
| Duo | N.A | 0.95 | 56.65 (5.49) | 50.78 (5.48) | 48.68 (5.48) | 47.26 (5.46) | 45.42 (5.45) | 45.11 (5.44) | 45.12 (5.44) | 44.84 (5.44) |
| Duo | N.A | 0.9 | 44.24 (5.40) | 40.08 (5.40) | 37.93 (5.39) | 36.66 (5.37) | 35.77 (5.37) | 34.79 (5.35) | 34.93 (5.35) | 34.75 (5.35) |
| MDLM | N.A | 1.0 | 194.09 (5.74) | 141.67 (5.69) | 120.95 (5.67) | 111.85 (5.65) | 107.89 (5.64) | 105.64 (5.64) | 105.40 (5.63) | 105.03 (5.62) |
| MDLM | N.A | 0.95 | 106.28 (5.61) | 77.06 (5.55) | 68.34 (5.53) | 63.19 (5.51) | 58.80 (5.49) | 56.94 (5.48) | 57.54 (5.47) | 56.44 (5.46) |
| MDLM | N.A | 0.9 | 70.34 (5.49) | 51.14 (5.43) | 43.60 (5.39) | 40.01 (5.37) | 39.02 (5.35) | 37.91 (5.34) | 37.59 (5.32) | 36.76 (5.31) |
| *Cap Schedule* | | | | | | | | | | |
| Duo | 0.005 | 1.0 | 88.78 (5.58) | 77.12 (5.57) | 72.05 (5.56) | 66.44 (5.54) | 61.63 (5.53) | 57.14 (5.51) | 52.49 (5.51) | 45.64 (5.45) |
| Duo | 0.01 | 1.0 | 86.89 (5.58) | 75.23 (5.56) | 68.98 (5.55) | 63.66 (5.54) | 57.34 (5.52) | 52.06 (5.50) | 46.04 (5.46) | 39.48 (5.39) |
| Duo | 0.005 | 0.95 | 55.56 (5.49) | 48.74 (5.47) | 44.93 (5.46) | 40.53 (5.43) | 36.26 (5.41) | 30.85 (5.37) | 25.66 (5.32) | 20.22 (5.22) |
| Duo | 0.01 | 0.95 | 54.07 (5.48) | 46.27 (5.46) | 41.93 (5.45) | 36.60 (5.41) | 30.98 (5.37) | 25.53 (5.31) | 20.10 (5.23) | 15.19 (5.07) |
| Duo | 0.005 | 0.9 | 44.06 (5.41) | 38.38 (5.39) | 34.84 (5.37) | 30.95 (5.33) | 27.37 (5.30) | 22.78 (5.24) | 18.66 (5.16) | 14.33 (5.03) |
| Duo | 0.01 | 0.9 | 43.05 (5.40) | 36.75 (5.38) | 32.27 (5.35) | 27.83 (5.30) | 23.38 (5.26) | 18.74 (5.17) | 14.40 (5.06) | 10.88 (4.87) |
| MDLM | 0.005 | 1.0 | 195.83 (5.74) | 142.25 (5.70) | 121.99 (5.68) | 113.94 (5.67) | 110.75 (5.66) | 112.78 (5.67) | 119.61 (5.69) | 131.85 (5.71) |
| MDLM | 0.01 | 1.0 | 198.02 (5.75) | 144.89 (5.70) | 125.25 (5.68) | 117.84 (5.68) | 116.62 (5.68) | 126.32 (5.71) | 143.96 (5.73) | 186.72 (5.76) |
| MDLM | 0.005 | 0.95 | 106.40 (5.61) | 74.97 (5.54) | 63.15 (5.52) | 55.82 (5.49) | 50.31 (5.47) | 43.78 (5.44) | 37.04 (5.40) | 30.46 (5.34) |
| MDLM | 0.01 | 0.95 | 105.45 (5.61) | 73.92 (5.54) | 61.41 (5.51) | 52.81 (5.48) | 46.03 (5.45) | 38.85 (5.42) | 31.30 (5.34) | 24.31 (5.23) |
| MDLM | 0.005 | 0.9 | 69.20 (5.49) | 49.59 (5.42) | 41.08 (5.38) | 35.19 (5.34) | 31.49 (5.31) | 26.33 (5.26) | 21.16 (5.18) | 15.87 (5.04) |
| MDLM | 0.01 | 0.9 | 68.57 (5.48) | 48.30 (5.42) | 38.80 (5.37) | 32.38 (5.32) | 27.66 (5.28) | 21.57 (5.18) | 16.26 (5.05) | 11.67 (4.79) |
| *Rescale Schedule* | | | | | | | | | | |
| Duo | 0.01 | 1.0 | 89.63 (5.58) | 79.80 (5.57) | 76.11 (5.56) | 73.43 (5.55) | 70.66 (5.54) | 70.46 (5.53) | 69.20 (5.54) | 68.25 (5.53) |
| Duo | 0.02 | 1.0 | 89.55 (5.58) | 79.44 (5.57) | 75.98 (5.56) | 72.99 (5.54) | 69.85 (5.54) | 68.39 (5.53) | 66.60 (5.53) | 63.70 (5.52) |
| Duo | 0.01 | 0.95 | 56.68 (5.49) | 50.80 (5.48) | 48.38 (5.47) | 46.91 (5.46) | 45.24 (5.45) | 44.64 (5.44) | 44.11 (5.44) | 43.49 (5.43) |
| Duo | 0.02 | 0.95 | 56.68 (5.49) | 50.66 (5.48) | 48.09 (5.47) | 46.19 (5.46) | 44.17 (5.44) | 42.71 (5.43) | 41.47 (5.43) | 38.06 (5.40) |
| Duo | 0.01 | 0.9 | 45.03 (5.41) | 40.02 (5.40) | 38.17 (5.39) | 36.60 (5.36) | 35.25 (5.35) | 34.35 (5.34) | 34.27 (5.35) | 33.07 (5.33) |
| Duo | 0.02 | 0.9 | 45.04 (5.41) | 40.00 (5.40) | 38.05 (5.39) | 36.15 (5.36) | 34.74 (5.35) | 33.13 (5.33) | 31.79 (5.32) | 29.08 (5.30) |
| Duo | 0.03 | 0.9 | 44.87 (5.41) | 40.05 (5.40) | 37.61 (5.39) | 35.26 (5.36) | 33.35 (5.34) | 31.17 (5.32) | 28.90 (5.31) | 24.93 (5.26) |
| Duo | 0.04 | 0.9 | 44.43 (5.41) | 39.67 (5.39) | 37.21 (5.38) | 34.75 (5.35) | 32.47 (5.34) | 29.30 (5.31) | 26.15 (5.28) | 22.05 (5.22) |
| Duo | 0.05 | 0.9 | 44.52 (5.41) | 39.49 (5.40) | 36.41 (5.38) | 33.68 (5.35) | 31.06 (5.34) | 26.94 (5.28) | 23.61 (5.25) | 19.21 (5.17) |
| MDLM | 0.01 | 1.0 | 194.29 (5.74) | 141.40 (5.69) | 121.04 (5.67) | 112.95 (5.65) | 107.80 (5.64) | 105.58 (5.64) | 105.69 (5.63) | 105.64 (5.63) |
| MDLM | 0.02 | 1.0 | 194.54 (5.74) | 140.81 (5.69) | 120.86 (5.67) | 112.64 (5.65) | 108.26 (5.64) | 105.65 (5.64) | 104.47 (5.63) | 105.61 (5.64) |
| MDLM | 0.01 | 0.95 | 106.43 (5.61) | 76.89 (5.55) | 65.42 (5.52) | 61.07 (5.50) | 58.77 (5.49) | 56.34 (5.47) | 56.29 (5.47) | 54.42 (5.45) |
| MDLM | 0.02 | 0.95 | 105.92 (5.60) | 76.23 (5.55) | 65.43 (5.52) | 60.80 (5.50) | 57.32 (5.49) | 54.94 (5.47) | 53.92 (5.46) | 50.57 (5.45) |
| MDLM | 0.01 | 0.9 | 70.45 (5.49) | 51.33 (5.43) | 43.59 (5.39) | 40.14 (5.36) | 38.68 (5.35) | 37.64 (5.34) | 36.48 (5.32) | 35.10 (5.31) |
| MDLM | 0.02 | 0.9 | 70.31 (5.49) | 51.06 (5.43) | 43.51 (5.39) | 39.61 (5.36) | 37.88 (5.35) | 36.28 (5.33) | 34.53 (5.31) | 31.62 (5.29) |
| MDLM | 0.03 | 0.9 | 69.89 (5.49) | 50.76 (5.42) | 43.23 (5.39) | 38.86 (5.36) | 36.77 (5.34) | 34.62 (5.32) | 31.44 (5.29) | 27.19 (5.25) |
| MDLM | 0.04 | 0.9 | 69.54 (5.49) | 50.30 (5.42) | 42.84 (5.39) | 38.02 (5.35) | 35.73 (5.33) | 32.44 (5.31) | 28.55 (5.27) | 23.72 (5.21) |
| MDLM | 0.05 | 0.9 | 69.44 (5.48) | 50.15 (5.42) | 42.39 (5.38) | 37.27 (5.35) | 34.10 (5.33) | 30.29 (5.30) | 26.03 (5.25) | 20.85 (5.16) |
| *Loop Schedule* | | | | | | | | | | |
| Duo | 0.01 | 1.0 | 108.15 (5.58) | 83.10 (5.58) | 71.16 (5.56) | 66.15 (5.55) | 60.49 (5.55) | 56.35 (5.53) | 53.06 (5.51) | 48.93 (5.48) |
| Duo | 0.02 | 1.0 | 103.48 (5.58) | 79.75 (5.58) | 67.99 (5.56) | 63.05 (5.55) | 56.92 (5.54) | 52.69 (5.51) | 48.63 (5.47) | 43.28 (5.37) |
| Duo | 0.01 | 0.95 | 65.29 (5.49) | 51.36 (5.48) | 43.27 (5.46) | 37.64 (5.43) | 32.04 (5.40) | 26.97 (5.35) | 22.94 (5.30) | 18.40 (5.20) |
| Duo | 0.02 | 0.95 | 61.61 (5.48) | 47.46 (5.47) | 38.78 (5.44) | 32.69 (5.40) | 27.26 (5.36) | 22.35 (5.29) | 18.43 (5.22) | 14.31 (5.06) |
| Duo | 0.01 | 0.9 | 52.12 (5.40) | 40.27 (5.39) | 33.71 (5.37) | 28.73 (5.33) | 24.47 (5.29) | 20.32 (5.23) | 17.01 (5.16) | 13.61 (5.05) |
| Duo | 0.02 | 0.9 | 49.08 (5.40) | 37.00 (5.38) | 30.08 (5.34) | 24.88 (5.29) | 20.59 (5.24) | 16.69 (5.16) | 13.61 (5.06) | 10.77 (4.92) |
| MDLM | 0.01 | 1.0 | 340.32 (5.81) | 192.48 (5.74) | 140.70 (5.70) | 127.32 (5.70) | 119.34 (5.69) | 127.63 (5.70) | 149.13 (5.73) | 198.48 (5.77) |
| MDLM | 0.02 | 1.0 | 338.82 (5.82) | 193.71 (5.75) | 144.92 (5.72) | 140.73 (5.72) | 136.30 (5.71) | 162.47 (5.75) | 246.89 (5.81) | 354.65 (5.78) |
| MDLM | 0.01 | 0.95 | 182.65 (5.67) | 101.56 (5.61) | 71.76 (5.56) | 58.43 (5.52) | 51.33 (5.50) | 45.27 (5.47) | 39.08 (5.43) | 33.48 (5.38) |
| MDLM | 0.02 | 0.95 | 177.31 (5.67) | 97.61 (5.61) | 68.49 (5.55) | 55.21 (5.51) | 47.71 (5.49) | 41.64 (5.45) | 34.91 (5.40) | 29.63 (5.33) |
| MDLM | 0.01 | 0.9 | 117.28 (5.55) | 65.24 (5.48) | 46.91 (5.43) | 37.62 (5.38) | 31.93 (5.34) | 27.80 (5.31) | 23.38 (5.25) | 19.78 (5.20) |
| MDLM | 0.02 | 0.9 | 112.21 (5.55) | 61.93 (5.48) | 43.89 (5.42) | 34.69 (5.37) | 28.99 (5.33) | 24.58 (5.29) | 20.09 (5.20) | 16.68 (5.13) |

Table 14: Generative Perplexity (Gen. PPL) and Unigram Entropy on OpenWebText (Gokaslan & Cohen, 2019) with $\Psi$-samplers using $\kappa_t$ schedules matching ReMDM (log-linear step size) and **distilled models** (as in Table 12). We experiment with nucleus sampling, following Wang et al. (2025).

| Algo | Eta | Nucleus P | Gen. PPLL ($\downarrow$) | | | | | | | |
|---|---|---|---|---|---|---|---|---|---|---|
| | | | 32 | 64 | 128 | 256 | 512 | 1024 | 2048 | 4096 |
| *Ancestral Sampling* | | | | | | | | | | |
| Duo | N.A | 1.0 | 68.35 (5.54) | 62.92 (5.54) | 59.82 (5.50) | 58.77 (5.46) | 58.32 (5.46) | 57.82 (5.45) | 55.39 (5.43) | 55.89 (5.42) |
| Duo | N.A | 0.95 | 44.94 (5.47) | 41.78 (5.46) | 40.32 (5.43) | 38.93 (5.39) | 38.69 (5.37) | 38.45 (5.36) | 36.92 (5.33) | 37.26 (5.33) |
| Duo | N.A | 0.9 | 35.92 (5.41) | 32.98 (5.40) | 31.49 (5.36) | 30.32 (5.31) | 30.06 (5.29) | 30.00 (5.28) | 28.90 (5.25) | 29.19 (5.25) |
| MDLM | N.A | 1.0 | 68.61 (5.48) | 55.26 (5.45) | 49.51 (5.44) | 46.13 (5.42) | 45.61 (5.42) | 44.87 (5.42) | 44.53 (5.41) | 44.38 (5.42) |
| MDLM | N.A | 0.95 | 46.07 (5.37) | 36.55 (5.33) | 32.91 (5.31) | 30.96 (5.30) | 30.26 (5.29) | 29.73 (5.29) | 29.54 (5.28) | 29.53 (5.28) |
| MDLM | N.A | 0.9 | 34.85 (5.26) | 28.21 (5.23) | 25.27 (5.21) | 24.31 (5.19) | 23.25 (5.18) | 22.75 (5.17) | 22.73 (5.17) | 22.46 (5.16) |
| *Cap Schedule* | | | | | | | | | | |
| Duo | 0.005 | 1.0 | 66.13 (5.54) | 58.49 (5.52) | 53.61 (5.48) | 47.85 (5.42) | 41.59 (5.39) | 34.05 (5.34) | 25.67 (5.22) | 19.25 (5.11) |
| Duo | 0.01 | 1.0 | 64.22 (5.53) | 55.84 (5.51) | 49.90 (5.48) | 40.95 (5.39) | 33.90 (5.34) | 26.29 (5.24) | 19.34 (5.11) | 14.31 (4.96) |
| Duo | 0.005 | 0.95 | 43.68 (5.47) | 38.77 (5.45) | 35.55 (5.40) | 31.36 (5.33) | 26.74 (5.28) | 21.84 (5.22) | 16.22 (5.08) | 12.00 (4.94) |
| Duo | 0.01 | 0.95 | 42.34 (5.46) | 37.14 (5.44) | 32.39 (5.38) | 27.25 (5.30) | 21.84 (5.22) | 16.74 (5.10) | 11.70 (4.92) | 8.68 (4.72) |
| Duo | 0.005 | 0.9 | 34.80 (5.40) | 30.95 (5.38) | 28.15 (5.34) | 24.47 (5.26) | 21.25 (5.18) | 17.02 (5.12) | 12.86 (4.99) | 9.48 (4.81) |
| Duo | 0.01 | 0.9 | 33.91 (5.40) | 29.27 (5.37) | 25.28 (5.31) | 21.40 (5.21) | 17.36 (5.13) | 13.22 (5.00) | 9.55 (4.82) | 6.92 (4.56) |
| MDLM | 0.005 | 1.0 | 67.27 (5.48) | 52.34 (5.45) | 44.38 (5.42) | 38.14 (5.40) | 32.35 (5.37) | 26.37 (5.34) | 20.64 (5.27) | 15.80 (5.19) |
| MDLM | 0.01 | 1.0 | 65.29 (5.47) | 49.78 (5.44) | 41.29 (5.40) | 33.39 (5.38) | 27.16 (5.34) | 21.04 (5.28) | 16.13 (5.19) | 12.16 (5.08) |
| MDLM | 0.005 | 0.95 | 44.71 (5.36) | 34.56 (5.32) | 29.42 (5.30) | 25.28 (5.27) | 21.55 (5.23) | 17.39 (5.18) | 13.63 (5.09) | 10.47 (4.98) |
| MDLM | 0.01 | 0.95 | 43.20 (5.36) | 32.84 (5.32) | 26.90 (5.29) | 22.19 (5.24) | 17.80 (5.19) | 13.93 (5.11) | 10.61 (4.98) | 7.68 (4.76) |
| MDLM | 0.005 | 0.9 | 33.81 (5.26) | 26.71 (5.22) | 22.81 (5.19) | 19.65 (5.16) | 16.67 (5.11) | 13.79 (5.06) | 10.74 (4.94) | 8.10 (4.78) |
| MDLM | 0.01 | 0.9 | 32.94 (5.25) | 25.51 (5.22) | 20.89 (5.18) | 17.19 (5.13) | 13.91 (5.05) | 10.91 (4.95) | 8.15 (4.78) | 5.93 (4.54) |
| *Rescale Schedule* | | | | | | | | | | |
| Duo | 0.01 | 1.0 | 68.33 (5.54) | 62.77 (5.53) | 59.65 (5.50) | 57.89 (5.46) | 57.43 (5.45) | 56.18 (5.44) | 53.13 (5.42) | 51.93 (5.41) |
| Duo | 0.02 | 1.0 | 68.18 (5.54) | 62.24 (5.53) | 59.07 (5.50) | 56.96 (5.46) | 55.73 (5.44) | 53.31 (5.43) | 48.20 (5.40) | 44.51 (5.38) |
| Duo | 0.01 | 0.95 | 45.04 (5.47) | 41.74 (5.46) | 39.99 (5.43) | 38.80 (5.38) | 38.10 (5.37) | 37.51 (5.36) | 35.43 (5.33) | 34.71 (5.32) |
| Duo | 0.02 | 0.95 | 44.89 (5.47) | 41.33 (5.46) | 39.81 (5.43) | 38.09 (5.38) | 36.79 (5.36) | 35.47 (5.35) | 31.97 (5.31) | 29.25 (5.28) |
| Duo | 0.01 | 0.9 | 35.91 (5.41) | 33.05 (5.40) | 31.55 (5.36) | 30.39 (5.31) | 29.94 (5.29) | 29.70 (5.28) | 27.73 (5.25) | 27.43 (5.24) |
| Duo | 0.02 | 0.9 | 35.81 (5.41) | 32.77 (5.40) | 31.17 (5.36) | 29.70 (5.30) | 28.70 (5.28) | 27.70 (5.26) | 25.31 (5.22) | 22.83 (5.19) |
| MDLM | 0.01 | 1.0 | 68.66 (5.48) | 55.16 (5.45) | 49.71 (5.43) | 45.88 (5.42) | 45.11 (5.42) | 43.79 (5.41) | 42.55 (5.40) | 40.90 (5.40) |
| MDLM | 0.02 | 1.0 | 68.73 (5.48) | 54.85 (5.45) | 48.12 (5.43) | 45.35 (5.42) | 44.10 (5.42) | 41.48 (5.41) | 38.76 (5.39) | 34.66 (5.38) |
| MDLM | 0.01 | 0.95 | 46.01 (5.37) | 36.58 (5.33) | 32.80 (5.31) | 30.65 (5.30) | 29.92 (5.29) | 29.18 (5.28) | 28.34 (5.28) | 27.38 (5.27) |
| MDLM | 0.02 | 0.95 | 45.92 (5.37) | 36.45 (5.33) | 32.49 (5.31) | 30.25 (5.29) | 29.01 (5.28) | 27.68 (5.28) | 25.75 (5.26) | 22.95 (5.24) |
| MDLM | 0.01 | 0.9 | 34.83 (5.26) | 28.15 (5.23) | 25.24 (5.21) | 23.73 (5.19) | 23.03 (5.18) | 22.36 (5.17) | 21.75 (5.17) | 20.93 (5.15) |
| MDLM | 0.02 | 0.9 | 34.83 (5.26) | 28.17 (5.23) | 24.97 (5.21) | 23.34 (5.19) | 22.34 (5.17) | 21.33 (5.17) | 19.88 (5.15) | 17.75 (5.12) |
| *Loop Schedule* | | | | | | | | | | |
| Duo | 0.01 | 1.0 | 80.39 (5.55) | 61.64 (5.54) | 52.51 (5.52) | 47.30 (5.48) | 40.27 (5.44) | 34.27 (5.40) | 27.28 (5.32) | 21.97 (5.26) |
| Duo | 0.02 | 1.0 | 75.97 (5.55) | 57.36 (5.53) | 47.47 (5.52) | 41.33 (5.47) | 34.18 (5.41) | 28.72 (5.36) | 22.16 (5.26) | 17.67 (5.18) |
| Duo | 0.01 | 0.95 | 51.76 (5.48) | 40.91 (5.47) | 34.68 (5.44) | 30.83 (5.39) | 25.86 (5.34) | 21.31 (5.27) | 17.15 (5.18) | 13.69 (5.10) |
| Duo | 0.02 | 0.95 | 48.78 (5.48) | 37.61 (5.46) | 30.95 (5.43) | 26.55 (5.36) | 21.60 (5.30) | 17.64 (5.22) | 13.84 (5.11) | 11.15 (5.02) |
| Duo | 0.01 | 0.9 | 41.15 (5.42) | 32.51 (5.40) | 27.96 (5.38) | 24.49 (5.32) | 20.52 (5.25) | 17.17 (5.19) | 13.90 (5.10) | 11.44 (5.02) |
| Duo | 0.02 | 0.9 | 38.73 (5.42) | 30.04 (5.40) | 24.99 (5.37) | 21.24 (5.29) | 17.40 (5.21) | 14.25 (5.13) | 11.51 (5.02) | 9.51 (4.94) |
| MDLM | 0.01 | 1.0 | 99.76 (5.51) | 62.76 (5.48) | 47.50 (5.45) | 39.07 (5.43) | 32.85 (5.41) | 28.01 (5.38) | 23.18 (5.34) | 19.32 (5.29) |
| MDLM | 0.02 | 1.0 | 93.99 (5.51) | 58.00 (5.48) | 43.00 (5.45) | 33.84 (5.42) | 28.60 (5.39) | 24.13 (5.36) | 19.81 (5.30) | 16.32 (5.24) |
| MDLM | 0.01 | 0.95 | 65.09 (5.40) | 41.85 (5.37) | 31.76 (5.33) | 26.11 (5.30) | 22.21 (5.28) | 19.19 (5.24) | 16.12 (5.20) | 13.59 (5.15) |
| MDLM | 0.02 | 0.95 | 61.24 (5.40) | 38.68 (5.36) | 28.92 (5.33) | 23.21 (5.29) | 19.45 (5.26) | 16.60 (5.21) | 13.84 (5.16) | 11.73 (5.09) |
| MDLM | 0.01 | 0.9 | 48.86 (5.29) | 32.03 (5.26) | 24.51 (5.23) | 20.56 (5.20) | 17.79 (5.18) | 15.42 (5.14) | 13.19 (5.09) | 11.29 (5.04) |
| MDLM | 0.02 | 0.9 | 46.12 (5.29) | 29.77 (5.27) | 22.52 (5.22) | 18.46 (5.19) | 15.86 (5.16) | 13.57 (5.11) | 11.54 (5.05) | 9.85 (4.98) |

