# OpenReview forum: "The Diffusion Duality, Chapter II: $\Psi$-Samplers"
_ICLR.cc/2026/Conference — ICLR 2026 Poster_

### Official Review · Reviewer_78X8 · 2025-10-19

**Soundness:** 3
**Presentation:** 3
**Contribution:** 2
**Rating:** 6
**Confidence:** 4

**Summary:**

The paper introduces a family of Predictor-Corrector (PC) samplers for discrete diffusion models, referred to as $\Psi$-samplers, which extend and generalize existing PC methods to work efficiently with arbitrary noise processes. The authors also propose a scalable and memory-efficient curriculum learning strategy for training these models.

**Strengths:**

1. The paper presents $\Psi$-samplers by generalizing the PC approach to discrete diffusion models, providing a unified framework that extends prior work.

2. The provided experiment shows that this memory-efficient curriculum method is likely effective, while empirical improvements are achieved in both language modeling and image generation tasks.

**Weaknesses:**

1. The $\Psi$-sampler introduces non-Markovian posteriors, but the theoretical discussion heavily relies on Gaussian diffusion analogies, and the impact on sample quality, variance, and convergence remains underexplored.


2. The method is highly sensitive to hyperparameters like $\kappa_t$ and noise schedules, and the evaluation primarily focuses on OpenWebText and CIFAR-10.

3. The paper claims that generation quality improves with more sampling steps, but this holds true for language modeling tasks, not image generation.

**Questions:**

1. Why is the Predictor-Corrector approach effective in improving sampling efficiency and generative quality in discrete diffusion models? Does the paper provide a detailed theoretical foundation explaining why this approach works, particularly regarding the non-Markovian dynamics introduced by the $\Psi$-samplers?

2. How might the proposed Predictor-Corrector $\Psi$-sampler relate to techniques like [1], [2], and [3], which have recently been proposed to improve generative quality by directly optimizing conditional variance/entropy during sampling?

3. Does the $\Psi$-sampler consider resolving, or at least attempting to address, the significant numerical instability issue arising from the schedules in diffusion models as $\sigma_t$ approaches 0?

[1]. Li, S., et al., EVODiff: Entropy-aware Variance Optimized Diffusion Inference, NeurIPS 2025.

[2]. Ifriqi, T. B.,  et al., Entropy Rectifying Guidance for Diffusion and Flow Models, NeurIPS 2025.

[3]. Stancevic, D., et al.,  Entropic Time Schedulers for Generative Diffusion Models, NeurIPS 2025.

---

> ### Author Response · Authors · 2025-11-19
> **Response (1 / 2) to the reviewer 78X8**
>
> We want to thank the reviewer for their constructive feedback. We address the reviewers comments and questions below.
>
> ## Concern 1: Theoretical Analysis on the effectiveness  of Predictor-Corrector Samplers
>
> We believe the reviewer is referring to `lines [295-297]`, where we motivate the effectiveness of the $\Psi$-sampler by drawing parallels to predictor-corrector (PC) samplers in Gaussian diffusion [1]. We now provide an intuitive explanation in `lines [305-313]`, summarized as follows.
>
> In practice, the denoising model may not perfectly model the clean data. Notice that the reverse posterior for the $\Psi$-sampler in `Eqn. 12`  includes an offset term  $(1 - \kappa_t)(1 - \alpha_s)\mathbf{\pi}$, which plays a crucial role in the effectiveness of the $\Psi$-sampler.
>
> - For MDMs where $\mathbf{\pi} = \mathbf{m}$,
>     - This offset allows a "clean" token to transition back to a masked state during reverse generation, unlike the ancestral sampler, which disallows remasking.
>     - Hence, incorrect tokens can be replaced by higher-quality ones.
> - For USDMs,
>     - If the denoiser assigns near-zero probability to the correct token, ancestral sampling would never recover it.
>     - In contrast, the $\Psi$-sampler, with $\mathbf{\pi} = \mathbf{1} / K$, ensures every token has a non-zero sampling probability, allowing the correct token a to reappear.
>     - Although this offset introduces the possibility of sampling incorrect tokens, the sampler’s marginals `Eqn. 1` match the intermediate marginals in `Eqn. 11`. Therefore, with sufficient Monte Carlo samples, the process is assured to converge toward the correct distribution.
>
> ---
>
> **References**
>
> [1] Score-based generative modeling through stochastic differential equation, Song et al., ICLR 2021

---

> ### Author Response · Authors · 2025-11-19
> **Response (2 / 2) to the reviewer 78X8**
>
> ## Concern 2: Sensitivity to hyperparameters
>
> Predictor-corrector (PC) samplers—both in Gaussian diffusion [4-7] and discrete diffusion [8-9]—introduce additional tunable hyperparameters. **Our approach follows this standard practice**: $\Psi$-samplers introduce only a single such parameter, $\kappa_t$.
>
> ## Comment 1: Relation of $\Psi$-samplers to samplers that account for the entropy or variance
>
> We thank the reviewer for pointing us to [1-3]. These works improve the estimate of the ancestral sampler $q_{s|t}(.| \mathbf{z}\_t, \mathbf{x}\_\theta)$ and are orthogonal to the $\Psi$-samplers. Essentially, these techniques can be used to improve $\Psi_{s|t} = \kappa_t q_{s|t}(.| \mathbf{z}\_t, \mathbf{x}\_\theta) + (1 - \kappa_s)q_{s}(.| \mathbf{x}\_\theta)$ by providing a more accurate estimate of $q_{s|t}$ .
>
> ## Comment 2: Any numerical instability during sampling at low noise levels?
>
> While Gaussian diffusion models often face **numerical instability** at low noise levels due to their continuous functional form, this issue **does not arise** in our case. $\Psi$-samplers operate in discrete space, and **their functional form is inherently stable across all noise levels.**
>
> ---
>
> **References**
>
>
> [1]. Li, S., et al., EVODiff: Entropy-aware Variance Optimized Diffusion Inference, NeurIPS 2025.
>
> [2]. Ifriqi, T. B., et al., Entropy Rectifying Guidance for Diffusion and Flow Models, NeurIPS 2025.
>
> [3]. Stancevic, D., et al., Entropic Time Schedulers for Generative Diffusion Models, NeurIPS 2025.
>
> [4] Fast Sampling of Diffusion Models with Exponential Integrator, Zhang et al., 2022
>
> [5] DPM-Solver: A Fast ODE Solver for Diffusion Probabilistic Model Sampling in Around 10 Steps, Lu et al., 2022
>
> [6] DPM-Solver++: Fast Solver for Guided Sampling of Diffusion Probabilistic Models, Lu et al., 2022
>
> [7] Denoising Diffusion Implicit Models, Song et al., 2020
>
> [8] Discrete Flow Matching, Gat et al., 2024
>
> [9] Remasking Discrete Diffusion Models with Inference-Time Scaling, Wang et al., 2024

---

### Official Review · Reviewer_XWfo · 2025-11-01

**Soundness:** 2
**Presentation:** 1
**Contribution:** 2
**Rating:** 6
**Confidence:** 4

**Summary:**

The paper propose predictor-correct samplers for discrete diffusion models. The authors:

1. First identity posteriors that sample from the marginals of a masked diffusion model.
2. Next, the authors propose posteriors, that are a mixture of distributions, a predictor for un-masking, and another distribution to re-noise.

Additionally, the authors also propose an efficient estimator for the curriculum learning objective of uniform state diffusion models. The authors note that:

1. There is a deterministic push-forward map between Gaussian and uniform state models. The denoiser can therefore be trained with continuous Gaussian noise.
2. However, for each token, the curriculum learning objective in Sahoo et al 2025a requires sampling several high-dimensional Gaussian vectors, which can be expensive for large vocabulary sizes.
3. The authors then propose a cheaper approximation that leads to reduced memory usage and faster training.
4. Computing the noise for each position requires $|\text{Vocab}|$ many Gaussians, however the authors rely on order-statistics to sample from the $k$ largest values of a Gaussian vector.

The authors then show that the superposition samplers combined with the diffusion duality model (Sahoo et al 2025a), a uniform state diffusion model, can match the performance of masked diffusion language models. The authors also show that using in-expensive approximations to compute the continual learning objective does not lead to a significant deterioration in performance.

**Strengths:**

The paper derives rigorous predictor-corrector schemes for both masked and uniform state diffusion models, as well as tractable approximations for the uniform state diffusion models.

**Weaknesses:**

See questions.

**Questions:**

1. To compute the test perplexity in table 2, how many $t$ samples do the authors use for each sample $x$.
2. Can the authors also demonstrate the predictor-corrector methods they propose with existing pre-trained diffusion language models such as LLaDA and Dream.

---

> ### Author Response · Authors · 2025-11-19
> **Response to Reviewer XWfo**
>
> ## Clarification 1: Number of MC samples to approximate perplexity
>
> We follow standard practice in the literature and use a single Monte Carlo (MC) sample, consistent with [1-3]. We found this setup stable across runs and used the same default random seed (1) as in those works.
>
> ## Clarification 2: Applicability to Llada and Dream
>
> **Yes, $\Psi$-samplers are applicable to Llada and Dream**. As proved in lines `[941-967]`, under the Masked Diffusion setting, $\Psi$-samplers reduce to ReMDM [2], the predictor-corrector method for Masked Diffusion Models. ReMDM has been successfully applied to Llada, **improving downstream performance on** **Countdown** (bidirectional reasoning) [4] and **TruthfulQA** (factual knowledge grasp) [5].
>
> ## Question to the reviewer: Regarding presentation
>
> > **Presentation:** 1: poor
> >
>
> We would appreciate specific feedback on which aspects of the presentation were unclear or could be improved. We are happy to address these points in the future revision.
>
> ---
> **References**
>
> [1] The Diffusion Duality, Sahoo et al, ICML 2025
>
> [2] Remasking Discrete Diffusion Models with Inference-Time Scaling, Wang et al., 2025
>
> [3] Simple and Effective Masked Diffusion Language Models
>
> [4] Jiacheng Ye, Jiahui Gao, Shansan Gong, Lin Zheng, Xin Jiang, Zhenguo Li, and Lingpeng
> Kong. Beyond autoregression: Discrete diffusion for complex reasoning and planning. arXiv
> preprint arXiv:2410.14157, 2024.
>
> [5] Stephanie Lin, Jacob Hilton, and Owain Evans. Truthfulqa: Measuring how models mimic
> human falsehoods. arXiv preprint arXiv:2109.07958, 2021.

---

### Official Review · Reviewer_FN73 · 2025-11-05

**Soundness:** 3
**Presentation:** 2
**Contribution:** 3
**Rating:** 8
**Confidence:** 3

**Summary:**

The authors tackle the problem of fast sampling in discrete diffusion models and introduce a family of predictor corrector samplers that generalize over prior methods and apply to arbitrary noising processes. The authors then combine this framework with the Uniform state discrete diffusion models and show that the method outperforms baselines on text and image generation tasks. The authors also propose a curriculum learning strategy which improves upon the training efficiency of existing discrete diffusion models.

**Strengths:**

1. I like Figure 1 which conveys the main results of the paper convincingly.
2. The paper is well written in the sense that the background section is well formulated, the main contributions of the paper are well supported by empirical results.
3. The idea of formulating non-markovian forward processes for discrete diffusion models is quite interesting given its numerous applications in the context of continuous diffusion models in the form of DDIM.

**Weaknesses:**

I dont have a lot of concerns around the proposed method but rather a few suggestions for improving the presentation of the paper.

**Presentation Issues**

1. Is there a reason for using the psi notation to denote distributions throughout the paper? We can probably get rid of notations and denote distributions using their standard notations like p(.) or q(.) like other works in the literature.

2. In general, a lot of intuition is missing around the sampler design in Section 3. It is not clear, why can we expect the non-markovian design of the forward process to yield more efficient samplers? Can the authors provide some intuition around the sampler design in this context? The formulation of the non-markovian diffusion forward processes in Section 3 reminds me of the DDIM formulation for continuous time diffusion models so maybe adding some intuition from the continuous time diffusion processes literature could be useful?

3. Similarly in Eq. 10 what is the intuition behind different terms in the psi-posterior? All of these intuitions can be clarified in the main text to improve readability and presentation. In the current form the section appears to be very dense in equations which hinders readability.

4. Can the authors provide some intuition of the Curriculum learning strategy proposed in Section 4?. The section seems very heavy on notation and it is hard to understand the high level idea on a first read.

5. The authors can also consider revising the title of the paper as it currently seems very opaque.

**Empirical Issues**

1. In Fig. 1 unlike the MDLM and Duo samplers, the performance of the MDLM + ReMDM baseline and the proposed method keeps improving. Do the authors have updated results on how these curves saturate? I guess the cost of sampling beyond 4k steps is probably too high from a practical standpoint but it could still be worth exploring it from a research standpoint.

2. Can the authors report the FID score comparisons in Fig. 3. Im curious how discrete diffusion models compare to continuous time models on image quality.

**Questions:**

See weaknesses

---

> ### Author Response · Authors · 2025-11-20
> **Response (1 / 2) to Reviewer FN73**
>
> hank the reviewer for their constructive feedback and for describing our work as well-written and that our experiments support well our claims. We address their questions below.
>
> ## Clarification 1: Intuition behind the $\Psi$-sampler and the $\kappa_t$ hyperparameter
>
> We now provide an intuitive explanation in `lines [305-313]`, summarized as follows.  In practice, the denoising model $\mathbf{x}_\theta$ may not perfectly model the clean data $\mathbf{x}$. Notice that the reverse posterior for the $\Psi$-sampler in `Eqn. 12`  includes an `offset` term  $(1 - \kappa_t)(1 - \alpha_s)\mathbf{\pi}$, which plays a crucial role in the effectiveness of the $\Psi$-sampler.
>
> - For MDMs where $\mathbf{\pi} = \mathbf{m}$,
>     - This `offset` allows a "clean" token to transition back to a masked state during reverse generation—unlike the ancestral sampler, which disallows remasking.
>     - Hence, incorrect tokens can be replaced by higher-quality ones.
> - For USDMs,
>     - If the denoiser assigns near-zero probability to the correct token, ancestral sampling would never recover it.
>     - In contrast, the $\Psi$-sampler, with $\mathbf{\pi} = \mathbf{1} / K$, ensures every token has a non-zero sampling probability, allowing the correct token a to reappear.
>     - Although this `offset` introduces the possibility of sampling incorrect tokens, the sampler’s marginals `Eqn. 1` match the intermediate marginals in `Eqn. 11`. Therefore, with sufficient Monte Carlo samples, the process is assured to converge toward the correct distribution.
>
> **Intuition behind $\kappa_t$ :** The `offset` term is controlled by $\kappa_t$, which regulates **exploration** during sampling.
>
> - In MDMs, smaller $\kappa_t$ values lead to more remasking and re-prediction of “clean” tokens.
> - In USDMs, smaller $\kappa_t$ increases transitions to alternative tokens.
>
> Hence, too small a $\kappa_t$ may increase sampling variance and destabilize generation, while **$\kappa_t = 1$** reduces the $\Psi$-sampler to the ancestral sampler.
>
> **Connection to DDIM** Indeed, $\Psi$-samplers are similar to DDIM samplers [4] in Gaussian diffusion, in the sense that both are non-Markovian. Exploring a deeper connection between the two is an interesting direction for future work.
>
> ---
>
> **References**
>
> [1] Score-based generative modeling through stochastic differential equation, Song et al., ICLR 2021
>
> [2] A Continuous Time Framework for Discrete Denoising Models, Campbell et al.,
>
> [3] Remasking Discrete Diffusion Models with Inference-Time Scaling, Wang et al., 2024
>
> [4] Denoising Diffusion Implicit Models, Song et al., 2022

---

> ### Author Response · Authors · 2025-11-20
> **Response (2 / 2) to Reviewer FN73**
>
> ## Clarification 2: Intuition on curriculum learning
>
> We have provided more context on curriculum learning in `lines [207-215, 230-237]`. We summarize it below:
>
> Sahoo et al. (2025) [6] introduce a training scheme for USDMs where the denoising transformer is trained to recover the clean sentence from the *Gaussian latents*—obtained by adding Gaussian noise to the one-hot representation of the clean sequence—instead of discrete latents. They engineer the model so that when the model operates on Gaussian latents, it receives a **superposition** of the embeddings of clean and noisy tokens, making the denoising task easier since the input retains partial information about the clean sequence.
>
> A hyperparameter, $\tau$, controls this mixture: $\tau > 0$ blends clean and noisy token embeddings, while $\tau = 0$ removes the mixture entirely and recovers the standard USDM training. Training begins with $\tau > 0$ (providing richer input signals) and gradually anneals to $\tau = 0$, forming a **curriculum learning** scheme that transitions from easier to harder denoising tasks.
>
> ## Clarification 3: The $\Psi$ Notation
>
> We chose the letter $\Psi$ to emphasize that our sampler is a superposition of two atomic/simple distributions: $q_t$ and $q_{s|t}$. The $q$	 notation is reserved for the standard forward and reverse process and hence we chose not to overload it futher. The inspiration for $\Psi$ comes from quantum mechanics (wave particle duality, symbol for the wave function).
>
> ## Clarification 4: What happens when sampling with more than 4k steps?
>
> Sampling with more than 4k steps was prohibitively expensive. Since we are a small academic lab, we chose to allocate our compute budget to explore a wider range of $\kappa_t$ schedule (`Tabs. 6 - 10` in the paper) and proper tuning of the baselines (e.g. ReMDM).
>
> ## Clarification 5: FID score comparison in Fig. 3
>
> **Note:** In the original manuscript, `Figs. 1, 3` show the FID and *Inception Score* (IS) when sampling with temperature T=0.8. After submission, we observed that using T=1 improves the FID in the high NFE regime. Therefore, we updated `Figs. 1, 3` to report the FID and IS without temperature scaling. Similar to text experiments, the FID and IS continue to improve with increasing sampling steps. The original plots are shown in the Appendix (Fig. 5) for reference.
>
> > Can the authors report the FID score comparisons in Fig. 3.
> >
>
> `Fig. 3` shows the best *Inception Score* obtained for different samplers and the associated FID is shown in `Fig. 1`.
>
> > Im curious how discrete diffusion models compare to continuous time models on image quality.
> >
>
> Below, we compare the FID and IS of **Duo** and **MDLM** with **EDM** [8], a Gaussian diffusion model by *Karras et al.*, 2022. EDM achieves lower FID due to several enhancements—optimized score-network preconditioning, loss weighting with log-normal noise, and non-leaking data augmentations [9]. We also hypothesize that Gaussian diffusion is inherently better suited for image generation than discrete diffusion models.
>
> | **Model** | **cond. FID ($\downarrow$)** | **cond. IS ($\uparrow$)** | **# Parameters** |
> | --- | --- | --- | --- |
> | MDLM | 24.67 | 7.42 | 35M |
> | Duo | 25.89 | 7.33 | 35M |
> | MDLM + ReMDM | 21.04 | 8.15 | 35M |
> | Duo + $\Psi$-samplers | 18.48 | 8.59 | 35M |
> | EDM **(Gaussian Diffusion)** | 1.84 | 9.95 | 55M |
>
> ## Comment: Title
>
> Thanks for the feedback. We’ll consider updating in the future version.
>
> ---
> **References**
>
> [1] Denoising Diffusion Implicit Models, Song et al., ICLR 2021
>
> [2] Remasking Discrete Diffusion Models with Inference-Time Scaling, Wang et al., 2025
>
> [3] Score-based generative modeling through stochastic differential equation, Song et al, 2021
>
> [4] Scheduled Sampling for Sequence Prediction with Recurrent Neural Networks, Bengio et al., 2015
>
> [5] Sequence Level Training with Recurrent Neural Networks, Ranzato et al., 2015
>
> [6] The Diffusion Duality, Sahoo et al, ICML 2025
>
> [7] Curriculum Learning, Bengio et al, ICML, 2009
>
> [8] Elucidating the Design Space of Diffusion-Based Generative Models, Karras et al, NeurIPS 2022
>
> [9] Training generative adversarial networks with limited data, Karras et al., Neurips 2020.

---

### Official Review · Reviewer_NGPT · 2025-11-06

**Soundness:** 3
**Presentation:** 3
**Contribution:** 3
**Rating:** 8
**Confidence:** 4

**Summary:**

This work proposes two improvements for uniform discrete diffusion models. The main contribution is a new family of predictor–corrector samplers (called psi-samplers) based on non-Markovian “psi-posteriors.”

In more detail, they first define an x-conditional version that is a convex combination of the forward process and x-conditioned reverse posterior. The true reverse posterior then turns out to be a convex combination of q_{s|t} and q_{0|t} for 0 < s < t < 1. The reverse posteriors preserve the same marginals as standard diffusion while allowing adjustable stochasticity during sampling. This leads to empirical gains over ancestral sampling for uniform-state diffusion models, particularly in text and image generation tasks. Unlike standard samplers that plateau with number of steps, Psi-samplers are shown to continue improving with more steps (perhaps could be interpretted as a type of Gibbs sampling in the regime where steps > dims).

A second contribution includes a practical efficiency optimization for a "curriculum learning" step from prior work, used during training. By approximating the vocabulary-wide averaging operation with a sparse top-k estimate, the authors reduce training time and memory usage while maintaining comparable perplexity and downstream accuracy. Altogether the combination of the train-time efficiency gain plus the new samplers give efficiency, perplexity, and sample quality gains on a number of tasks relative to a number of baselines.

The sampling idea is novel and presented cleanly, while the curriculum result is an engineering improvement, still meaningful, but not quite clearly defined or motivated besides the literal algorithm of the proposed computation.

Overall, a solid technical contribution with good empirical validation. I view the sampling advance as the core insight and the training speedup as a perhaps useful but secondary contribution, at least given the amount of explanation/exploration of the underlying method being optimized.

**Strengths:**

- clear motivation for why sampling needs improvement in discrete diffusion: standard samplers can "over-commit" and cannot "self-correct" without further hacks, sometimes including adversarial training or other approximations.

- the psi-sampler formulation is general and recovers previous a few predictor–corrector methods as special cases (though please careful to not say "all cases in the literature", you never know with so many papers coming out daily, I suggest "that the authors are aware of")

- broad experimental coverage: language modeling, QA, and image generation for tasks + ablations on the \kappa_t hyperparameter and time-scheduling of where to use the psi-posterior.

- consistent empirical setups with appropriate baselines (Duo, MDLM, ReMDM) drawing on the Sahoo, Lou, etc... works.

- demonstrates measurable improvement in both FID/inception (images) and (generative GPT2) perplexity (on text) at fewer sampling steps, with nice gains in performance as NFE increases.

**Weaknesses:**

- The curriculum section assumes prior familiarity with Sahoo 2025a and gives little intuition for why that weighted-average operation helps training. Since you are devoting nearly a whole section to this method, I ask that you at least give a few sentences on how exactly the "curriculum" technique during training uses this average computation that you are approximating. Otherwise, the speedup could be relegated to an appendix section (I think "curriculum" is used 23 times in main text without being defined).

- experiments, while complete, lack interpretation, or at least attempts at providing intuition: results are shown, but, e.g., there is no discussion of why low-noise $\kappa_t$ regimes work best or what happens mechanistically when $\kappa_t$ is too small or too large

- the connection between psi-samplers and the NELBO s not explored, which might provide extra insight on what's going on.

- somewhat minor: when describing t_off, t_on, etc... it's possible that t is overloaded. Are you describing a function, t_for_model_input(t_real)? If so could you distinguish in symbols? It was a bit hard to read phrases like "t is constant in [t_off, t_on]".

**Questions:**

- there is some emphasis on the non-markovianness of the samplers. It would be curious to see where exactly this non-markovianness really matters, for example by seeing which conditioning values or time steps are correlated.

- could $\kappa_t$ be learned or adapted dynamically during sampling rather than chosen?

- the paper says psi-samplers improve with increasing NFEs while ancestral ones plateau. Why exactly does extra noise continue to help after many steps? If you run it for a while (steps >> dims/length), does it ever settle? Or if not, is there some relationship to Gibbs sampling?

- It seems like the psi posterios could be used for purely continuous diffusion models too. Has something like this been done in that setting? What would be things to watch out for?  More generally, is there any such trick in other areas of sampling that motivates this approach?

- when $\kappa_t < 1$, the sampler is said to be “noisier.” How is this related to the variance schedule $\alpha_t$? Is there any sense in which $\kappa_t$ values index an implied different set of $\alpha_t$'s?

- why does the largest gains from psi-posteriosr occur at low noise levels and $\kappa_t$ close to 1? Is this due to the denoiser’s inductive (architectural) bias? how the mixture interacts with the diffusion schedule? some special property of the sequence of psi posteriors over time? something else?

- could the authors visualize or quantify how psi-sampling trajectories differ qualitatively from ancestral ones?

---

> ### Author Response · Authors · 2025-11-20
> **Response (1 / 3) to Reviewer NGPT**
>
> We thank the reviewer for their constructive feedback and for describing our work as a solid technical contribution with strong empirical validation. We address their comments and questions below.
>
> ## Clarification 1: Intuition on curriculum learning
>
> We have provided more context on curriculum learning in `lines [207-215, 230-237]`. We summarize it below:
>
> Sahoo et al. (2025) [1] introduce a training scheme for USDMs where the denoising transformer is trained to recover the clean sentence from the *Gaussian latents*—obtained by adding Gaussian noise to the one-hot representation of the clean sequence—instead of discrete latents. They engineer the model so that when the model operates on Gaussian latents, it receives a **superposition** of the embeddings of clean and noisy tokens, making the denoising task easier since the input retains partial information about the clean sequence.
>
> A hyperparameter, $\tau$, controls this mixture: $\tau > 0$ blends clean and noisy token embeddings, while $\tau = 0$ removes the mixture entirely and recovers the standard USDM training. Training begins with $\tau > 0$ (providing richer input signals) and gradually anneals to $\tau = 0$, forming a **curriculum learning** scheme that transitions from easier to harder denoising tasks.
>
> ---
>
> **Reference**
>
> [1] The Diffusion Duality, Sahoo et al., ICML 2025

---

> ### Author Response · Authors · 2025-11-20
> **Response (2 / 3) to the reviewer NGPT**
>
> ## Clarification 2: Intuition behind the $\Psi$-sampler— $\kappa_t$ hyperparameter
>
> We now provide an intuitive explanation in `lines [305-313]`, summarized as follows.  In practice, the denoising model $\mathbf{x}_\theta$ may not perfectly model the clean data $\mathbf{x}$. Notice that the reverse posterior for the $\Psi$-sampler in `Eqn. 12`  includes an `offset` term  $(1 - \kappa_t)(1 - \alpha_s)\mathbf{\pi}$, which plays a crucial role in the effectiveness of the $\Psi$-sampler.
>
> - For Masked Diffusion Models where $\mathbf{\pi} = \mathbf{m}$,
>     - This `offset` allows a "clean" token to transition back to a masked state during reverse generation—unlike the ancestral sampler, which disallows remasking.
>     - Hence, incorrect tokens can be replaced by higher-quality ones.
> - For Uniform State Diffusion Models,
>     - If the denoiser assigns near-zero probability to the correct token, ancestral sampling would never recover it.
>     - In contrast, the $\Psi$-sampler, with $\mathbf{\pi} = \mathbf{1} / K$, ensures every token has a non-zero sampling probability, allowing the correct token a to reappear.
>     - Although this `offset` introduces the possibility of sampling incorrect tokens, the sampler’s marginals `Eqn. 1` match the intermediate marginals in `Eqn. 11`. Therefore, with sufficient Monte Carlo samples, the process is assured to converge toward the correct distribution. **This motivates why the sample quality keeps on improving as we increase the number of sampling steps**
>
> > the paper says psi-samplers improve with increasing NFEs while ancestral ones plateau. Why exactly does extra noise continue to help after many steps?
> >
>
> Please refer to our argument above where we motivate why increasing the sampling steps improve the sample quality.
>
> > Does the performance settle?
> >
>
> Sampling with more than 4k steps was prohibitively expensive. Since we are a small academic lab, we chose to allocate our compute budget to explore a wider range of $\kappa_t$ schedule (`Tabs. 6 - 10` in the paper) and proper tuning of the baselines (e.g. ReMDM [1]).
>
> > why does the largest gains from psi-posteriosr occur at low noise levels and $\kappa_t$
> >
>
> The `offset` term is controlled by $\kappa_t$, which regulates **exploration** during sampling.
>
> - In MDMs, smaller $\kappa_t$ values lead to more remasking and re-prediction of “clean” tokens.
> - In USDMs, smaller $\kappa_t$ increases transitions to alternative tokens.
>
> **Hence, too small a $\kappa_t$ may increase sampling variance and destabilize generation; hence, $\kappa_t$ is kept close to 1**. Also, note that **$\kappa_t = 1$** reduces the $\Psi$-sampler to the ancestral sampler.
>
> > why does the largest gains from psi-posteriosr occur at low noise levels
> >
>
> The goal of a predictor–corrector sampler is to **correct accumulated errors** during sampling. Thus, correction steps ($\kappa_t < 1$) are most effective in the **later denoising stages** ($t \to 0$). Empirically, we observe the same trend—**$\Psi$-samplers perform best at lower noise levels** (`Tabs. 9, 12`).
>
> > connection between psi-samplers and the NELBO s not explored, which might provide extra insight on what's going on.
> >
>
> As described in `Lines [297-298]` the $\Psi$-posteriors admit a well principled NELBO (presented in `Sec. A3`); however, it isn't central to the theme of the paper which focuses on sampling.
>
> > could $\kappa_t$ be learned or adapted dynamically during sampling rather than chosen
> >
>
> One can potentially, define $\kappa_t$ as a learnable parameter and try learning it by minimimizing the NELBO induced by $\Psi$-samplers. We leave this for future exploration.
>
> > $\kappa_t < 1$, the sampler is said to be “noisier.” How is this related to the variance schedule $\alpha_t$?
> >
>
> We have provided an intuitive explanation above as to how $\kappa_t$ affects the variance of the sampler. $\kappa_t$ and $\alpha_t$ are varied independent of each other.
>
> > It seems like the psi posteriors could be used for purely continuous diffusion models too.
> >
>
> $\Psi$-posteriors are a linear combination of the forward and reverse posteriors, $\Psi_{s|t} = \kappa_t q_{s|t} + (1 - \kappa_t) q_s$, making them compatible with **Gaussian diffusion models** as well. To our knowledge, this connection has not yet been explored and is an interesting direction for future work.
>
> ---
>
> **References**
>
> [1] Remasking Discrete Diffusion Models with Inference-Time Scaling, Wang et al. NeurIPS 2025

---

> ### Author Response · Authors · 2025-11-20
> **Response (3 / 3) to the reviewer NGPT**
>
> ## Clarification: On the time notation
>
> > when describing t_off, t_on, etc... it's possible that t is overloaded. Are you describing a function, t_for_model_input(t_real)? If so could you distinguish in symbols? It was a bit hard to read phrases like "t is constant in [t_off, t_on]".
> >
>
> We agree the notation can be confusing and follows the baseline [1]. In our setup, the **diffusion timestep t is a function of the Number of Function Evaluations (NFEs)**. In standard ancestral sampling, timestep $t$ is decreased by a step size $\Delta$ to produce a sample at $s = t - \Delta$.
>
> In one of the ablations for the predictor–corrector samplers called “loop-$t$”, we instead repeatedly sample from $\Psi_{s|t}$ with $s = t$ for specified number of times. As evident, we incur NFEs in this phase but don’t necessarily change $t$.
>
> Nomenclature summary:
>
> - The variable $t: 1 \rightarrow 0$ : the time index,
> - $t_\text{on}, t_\text{off} \in [0, 1]$ : the hyperparameters with $t_\text{on} > t_\text{off}$
>     - $\kappa_t < 1$ for $t \in [t_\text{off}, t_\text{on}]$ — The predictor corrector phase
>     - $\kappa_t = 1$ otherwise— ancestral phase.
>
> We will clarify this notation in the next revision.
>
> ---
>
> **References**
>
> [1] Remasking Discrete Diffusion Models with Inference-Time Scaling, Wang et al. NeurIPS 2025

---

### Author Response · Authors · 2025-12-04

We sincerely thank all reviewers for their thoughtful feedback. We are grateful to the ACs for their work as well. As the discussion period is ending soon, we would like to summarize our contributions.

---

## The Diffusion Language Modeling Landscape

Diffusion language models have transitioned from a niche research topic to a serious alternative to autoregressive models. In the past year alone, Google  (`Gemini Diffusion`), ByteDance (`Seed Diffusion` [1]), and Inception Labs (`Mercury Coder` [2]) have announced diffusion models rivaling state-of-the-art LLMs, with impressive speed-quality tradeoffs. Notably, these models [1-2] are built on the MDLM [3-5] and BD3-LMs [6] frameworks. Similarly, academic research has focused almost exclusively on **Masked Diffusion Models (MDMs)** [3-8]**.**

In this work, **we challenge this stance** and argue that Uniform-State Diffusion Models (USDMs) [9-10] deserve significantly more attention, particularly because of their **native self-correction ability**. Unlike MDMs, whose mistakes cannot be revised, **USDMs can natively correct errors throughout the sampling process.**

### Our Contributions

Our work provides (1) a theoretical unification and simplification of prior Predictor-Corrector samplers, (2) significantly improved results on text and images, and (3) practical training speedups for USDMs. Specifically,

1. **$\Psi$-Samplers:** We introduce a family of Predictor-Corrector (PC) samplers that simplify and generalize prior PC methods to arbitrary noise priors. With $\Psi$-Samplers, **USDMs outperform MDMs across all inference budgets**, on both text and images. Crucially, $\Psi$-Samplers allow **USDMs to generate significantly higher quality samples than MDMs as compute resources increase**, while ancestral sampling plateaus.
2. **Efficient Training**: We propose an efficient algorithm to **reduces the end-to-end training time by 25% for state-of-the-art USDMs**, and peak memory usage by 33%.

USDMs are commonly seen as beneficial only at low inference budgets. **Our results challenge this belief: USDMs scale better than MDMs when both use PC samplers**. Combined with their native self-correction ability, this makes USDMs a promising yet under-explored paradigm.

---

[1] Seed et al., 2025 “Seed Diffusion: A Large-Scale Diffusion Language Model with High-Speed Inference”.

[2] Khanna et al., 2025 “Mercury: Ultra-Fast Language Models Based on Diffusion”.

[3] Shi et al., 2024 “Simplified and Generalized Masked Diffusion for Discrete Data”

[4] Sahoo et al., 2024 “Simple and Effective Masked Diffusion Language Models”

[5] Ou et al., 2024 “Your Absorbing Discrete Diffusion Secretly Models the Conditional Distributions of Clean Data”

[6] Arriola et al., 2025, "Block Diffusion: Interpolating Between Autoregressive and Diffusion Language Models”

[7] Gat et al., 2024 “Discrete Flow Matching”

[8] Wang et al., 2025 “Remasking Discrete Diffusion Models with Inference-Time Scaling”

[9] Sahoo et al., 2025 “The Diffusion Duality”

[10] Schiff et al., 2025 “Simple Guidance Mechanisms for Discrete Diffusion Models”.

---

### Meta-Review · Area_Chair_aREP · 2026-01-05

**Summary:**

This paper proposes a unified predictor–corrector sampling framework for discrete diffusion models via non-Markovian Ψ-posteriors/Ψ-samplers, which generalize prior remasking/PC samplers and extend naturally to uniform-state diffusion. In addition, it introduces a scalable curriculum approximation for the Gaussian-relaxation phase used in Duo-style training, reducing memory and training cost while maintaining likelihood and downstream performance. Across reviewers, there is broad agreement that the Ψ-sampler contribution is the core insight and is empirically well supported on both language modeling (e.g., OpenWebText/LM1B and downstream MCQ) and image generation (CIFAR-10), with improvements over strong baselines and better scaling with increased NFEs where ancestral sampling plateaus.

**Reviewer Concerns:**

The main concerns raised are largely about presentation and intuition: several reviewers found Sections 3–4 equation-dense and requested clearer motivation for the non-Markovian construction, the role of κ (and schedules), and a higher-level explanation of the curriculum mechanism. The rebuttal addresses many of these points by adding intuition for the offset term in the Ψ-sampler posterior (enabling error correction/remasking and nonzero token probability while preserving marginals), clarifying curriculum learning, and responding to questions about perplexity estimation and applicability to pretrained masked diffusion models via the reduction to ReMDM. Some open questions remain (e.g., behavior at extremely large NFEs, broader dataset coverage, and deeper theoretical analysis of variance/convergence), but these are not required for acceptance.

**Reviewer Scores:**

Reviewer 78X8 is likely to raise the score given the raised issued are adequately handled.
 Reviewer XWfo not sure.
 Reviewer FN73 and Reviewer NGPT are likely to maintain their high scores.

---

### Decision · Program_Chairs · 2026-01-26

Accept (Poster)